# Rho-Kinase Inhibitors for the Treatment of Refractory Diabetic Macular Oedema

**DOI:** 10.3390/cells10071683

**Published:** 2021-07-03

**Authors:** Milagros Mateos-Olivares, Luis García-Onrubia, Fco. Javier Valentín-Bravo, Rogelio González-Sarmiento, Maribel Lopez-Galvez, J. Carlos Pastor, Ricardo Usategui-Martín, Salvador Pastor-Idoate

**Affiliations:** 1Department of Ophthalmology, Hospital Clínico Universitario de Valladolid, 47003 Valladolid, Spain; mmateoso@saludcastillayleon.es (M.M.-O.); Luis.GarciaOnrubia@gstt.nhs.uk (L.G.-O.); fvalentin@saludcastillayleon.es (F.J.V.-B.); mlopezgal@saludcastillayleon.es (M.L.-G.); pastor@ioba.med.uva.es (J.C.P.); 2Department of Ophthalmology, St Thomas’ Hospital, London SE1 7EH, UK; 3Area of Infectious, Inflammatory and Metabolic Disease, Institute of Biomedical Research of Salamanca (IBSAL), 37007 Salamanca, Spain; gonzalez@usal.es; 4Institute of Molecular and Cellular Biology of Cancer (IBMCC), University of Salamanca-CSIC, 37007 Salamanca, Spain; 5Retina Group, IOBA (Institute of Applied Ophthalmobiology), University of Valladolid, 47002 Valladolid, Spain; 6Cooperative Network for Research in Ophthalmology Oftared, National Institute of Health Carlos III, 28220 Madrid, Spain

**Keywords:** diabetic macular oedema, diabetic retinopathy, macular oedema, rho-associated kinases

## Abstract

Diabetic macular oedema (DMO) is one of the leading causes of vision loss associated with diabetic retinopathy (DR). New insights in managing this condition have changed the paradigm in its treatment, with intravitreal injections of antivascular endothelial growth factor (anti-VEGF) having become the standard therapy for DMO worldwide. However, there is no single standard therapy for all patients DMO refractory to anti-VEGF treatment; thus, further investigation is still needed. The key obstacles in developing suitable therapeutics for refractory DMO lie in its complex pathophysiology; therefore, there is an opportunity for further improvements in the progress and applications of new drugs. Previous studies have indicated that Rho-associated kinase (Rho-kinase/ROCK) is an essential molecule in the pathogenesis of DMO. This is why the Rho/ROCK signalling pathway has been proposed as a possible target for new treatments. The present review focuses on the recent progress on the possible role of ROCK and its therapeutic potential in DMO. A systematic literature search was performed, covering the years 1991 to 2021, using the following keywords: “rho-Associated Kinas-es”, “Diabetic Retinopathy”, “Macular Edema”, “Ripasudil”, “Fasudil” and “Netarsudil”. Better insight into the pathological role of Rho-kinase/ROCK may lead to the development of new strategies for refractory DMO treatment and prevention.

## 1. Introduction

Diabetic retinopathy (DR) is one of the feared chronic complications of Diabetes Mellitus (DM), a pandemic disease that affects approximately 463 million adults nowadays. Its prevalence has been rising in recent years, exemplifying its steady increase; by 2045, 700 million adults are expected to live with diabetes. A total of 40% of diabetic patients over 40 years old suffer from DR. [1]. 

Diabetic macular oedema (DMO) is associated with an abnormal increase in fluid volume in the macula, whether infiltrating retinal layers or collecting in the subretinal space. It is a significant cause of vision loss in those patients since it primarily affects central vision. DMO may occur in any disease stage and is more frequent in insulin-dependent type 2 DM [1]. 

The pathogenesis of DMO is multifactorial and complex, with a combination of pathological conditions related to retinal hypoxia, vascular permeability, angiogenesis, and inflammation processes leading to the development of the characteristic retinal alterations in patients with DMO [2,3,4,5].

Since the 1970s, focal/grid laser treatment has been considered the only available treatment for the management of this and it was accepted that in general, it only stabilised functional loss but did not improve it [6]. However, the breakthrough of intravitreal antivascular endothelial growth factor (VEGF) therapy has now replaced laser treatment as several trials have shown improved outcomes [7,8], which is nowadays the treatment of choice. However, anti-VEGF therapy has several limitations since it requires multiple visits and injections, increasing the burden placed on the healthcare system. Furthermore, DMO persists in more than 40% of patients even after numerous intravitreal injections according to the protocol I trial [4]. The efficacy of anti-VEGF treatment may be less pronounced in the clinical practice setting where undertreatment is very frequent [9,10,11] (Figure 1). Moreover, there are certain doubts about the long-term effect of the inhibition of a factor (VEGF) that is essential, among other things, to keep the choriocapillaris in good condition. 

## 2. Materials and Methods

A comprehensive review of the literature was performed through MEDLINE, PubMed, Web of Science, Scopus, and Embase electronic databases, covering the years 1991–2021. Potentially relevant articles were sought by using the search terms in combination as Medical Subject Headings (MeSH) terms and text words: “rho-Associated Kinases”, “Diabetic Retinopathy”, “Diabetic Macular Edema”, “Ripasudil”, “Fasudil” and “Netarsudil”. We also studied reviews, comments, and disquisitions on the pathology. In addition, we scanned the reference lists of the retrieved publications to identify additional relevant articles. The search was supplemented using the MEDLINE option ‘Related Articles’. No language restrictions were applied. The abstracts for each article were studied to ensure relevance and significance to the review. The identification of studies was illustrated using the PRISMA 2020 flow diagram for new systematic reviews (Appendix A). Two reviewers independently rated the quality of each study by assessing its methodology with the Mixed Methods Appraisal Tool (MMAT), designed for the appraisal of the stage of systematic mixed studies reviews (Appendix A) [12].

## 3. Pathophysiology of Diabetic Macular Oedema

These pathogenic mechanisms of DMO are multifactorial and complex, making its management challenging. Changes under DMO comprise metabolic disturbances that are both causes and consequences of retinal sight-threatening damage. This condition disturbs retinal tissues at all levels from the inner to outer layers [1]. The problem is that there are still gaps in understanding these underlying mechanisms in its development.

The abnormal accumulation of fluid in the macular area can occur intra-cellularly (cytotoxic), extracellularly (vasogenic), or in both manners. In patients with DMO, the two forms are cytotoxic at the beginning and vasogenic later on when the blood retinal-barrier (BRB) breakdown occurs, and fluids accumulate in the extracellular space, even accompanied by a neurosensory serous detachment [13,14,15]. 

### 3.1. Metabolic Changes

Chronic hyperglycaemia is considered the leading risk factor for DR and DMO [2,9]. The lack of glucose regulation produces a hyperglycaemia-induced metabolic state activating and overusing alternative metabolic pathways such as the polyol-pathway, which produces intracellular stress in osmotic cells due to the accumulation of sorbitol and protein-kinase-C-diacylglycerol (PKC-DAG) pathway, which is associated with cytotoxic oedema, vascular abnormalities, and endothelial permeability. Non-enzymatic glycation also leads to the accumulation of advanced glycation end-products (AGEs) with similar consequences [14,16,17,18]. Glutamate is also over-produced in DR, a neurotransmitter in the brain and retina whose excess is extremely cytotoxic [15,19,20].

All these disturbances result in an altered cellular redox state, with augmented free radicals and reactive oxidative species (ROS) representing an inflammatory stimulus [14,21,22,23]. This situation leads to the modification and conformation of endothelial cell junctions, the activation of the cellular injury pathways, the alteration of extracellular matrix components and increases in capillary basement membrane thickness state, impeding vascular physiological functioning and permeability, thus contributing to ischemic processes in DR [23,24].

### 3.2. Blood-Retinal Barrier Breakdown

Neuroretinal tissue is highly vulnerable, and therefore, the balance between molecules and fluid entry and exit is highly regulated. The principal regulatory mechanism for fluid access in the retina is BRBs, both inner and outer [2]. BRB breakdown is a complex process that is regulated by multiple factors and involves different mechanisms, and when it occurs at inner and/or outer BRB, excess fluid can accumulate, and this can result in macular oedema. Regarding fluid exit, the drainage system depends on a complex system of transport of ion/water channels located in the polarized retinal pigment epithelium (RPE) cells and the retinal Müller glial cells [2,25].

The inner BRB is composed of endothelial cells of retinal vessels sealed by tight junctions, where pericytes also seem to play an important role, and the astrocytes create a complex structure formed by the interaction of different scaffolding and transmembrane and signalling proteins [2,17,26]. 

In contrast, the outer BRB comprises the intercellular junction complex of the RPE, its basal membrane and, partially, Bruch’s membrane, which creates a barrier between the choroidal vasculature and the neuroretina. Choriocapillaris vessels nourish photoreceptors and external retinal layers, and unlike retinal vessels, are highly fenestrated, permeable and lack pericytes [27]. Of note, recent studies have also demonstrated the barrier properties of the outer limiting membrane [28].

As mentioned, the inner BRB is highly regulated by a complex neuro-glio-vascular system made by endothelial cells and their basal lamina, surrounded by pericytes, astrocytes, Müller cells and microglia, which interact with them, and other retinal components selectively controlling the molecular transport across this barrier [2,15,29].

BRB breakdown can result from a disruption of the tight junctions, by upregulation of substance transport across RPE cells or retinal vascular endothelial cells, by degenerative changes to the barrier-forming cells or to the regulatory cells (pericytes and glia cells) [30]. BRB breakdown can be related to localized structural defects, such as microaneurysm, or to a diffuse retinal vascular leakage in which diffusible factors can be involved. 

For example, an overproduction by retinal cells and an imbalance between VEGF and platelet-derived growth factor (PDGF), is partially responsible for altering BRBs integrity and permeability and for vasogenic oedema and cellular damage including endothelial and Müller cells, astrocytes and pericytes loss [17,26,31,32,33]. In DR, a chronic hyperglycemic environment ends up provoking an increase of inflammatory and proangiogenic factors and a metabolic disruption in both barriers [15].

### 3.3. Retinal Pigment Epithelium Damage

Right above the RPE and below the neural retina, the subretinal space is found, which is a potential space maintained by the RPE and where fluid buildup can occur because of DR. The specific protein composition and disposition in the outer BRB allows the RPE to accomplish its functions, including epithelial transport, immune privilege preservation and trophic factors supply for the retina [27,34,35,36,37].

Therefore, in the outer-BRB, proteins transporters, pumps and channels are purposefully expressed either on the apical or basal membrane cells [38,39,40]. This polarisation is maintained by its own tight, adherent and gap junctions combined with cytoskeletal and other specific signalling and effector cell proteins such as the Rho/Rac kinases [27,41].

Several studies have reported external retina damage due to DMO with altered photoreceptors shape and cellular mortality [42,43], RPE disturbances [44,45,46], retinal thickening due to vascular leakage [47], increased oxidative stress and leukocyte circulation [48,49], inflammatory factors [50] and an altered ion-flux across the RPE [16].

### 3.4. Hypoxia

Capillary abnormalities and obstruction in DR induce retinal hypoxia, which is also one of the hinges of DME pathogenesis [2,14]. In hypoxic conditions, hypoxia-inducible factor 1α (HIF-1α) promotes the expression of angiogenic factors, including VEGF. Therefore, HIF production promotes retinal vessels formation in an attempt to improve retinal oxygenation but contributing to pro and antiangiogenic factors disbalance and its deleterious consequences for retinal cells and BRBs [2,15,31,33]. VEGF and HIF-1α have shown an intravitreal concentration up to 10 times higher in diabetic patients than in nondiabetic ones [14,15].

### 3.5. Inflammation Role

New evidence supports the role of inflammation in the early changes found in DR and DMO [29,51,52]. The increased redox cellular state constitutes an inflammatory stimulus with the consequent production of cytokines (IL-6, IL-8, IL-1b, TNF-α), chemokines (CCL-2, CCL-5, CXCL-8, CXCL-10, CXCL-12), growth factors (VEGF, PDGF, HGF), and a pull factor for monocytes and leukocytes, which suppose a loss in retinal immune privilege [53,54]. In fact, recent investigations have suggested that endothelial injury in the vascular retina is due to glucose-induced cytokine release rather than a direct effect of high glucose on endothelial cells [55].

The upregulation of intercellular adhesion molecule 1 (ICAM-1) and vascular cell adhesion molecule 1 (VCAM-1) is a good case in point [3,11,56]. The potential ability of diabetic leukocytes to adhere to the vascular endothelium and generate toxic superoxide radicals has been shown. In addition, leukocytes of diabetic patients have been related to some abnormalities in the vascular system of DR, such as capillary nonperfusion, endothelial cell damage, or vascular leakage due to the disruption of the BRB [57,58]. Of note, the potential role of ICAM-1 has been shown in the development of retinal abnormalities, since the reduced expression of ICAM-1 in experimental models has significantly decreased the formation of acellular capillaries as well as prevented the loss of pericytes, which is a well-known early histopathological feature of DR [59].

### 3.6. Refractory Diabetic Macular Oedema

All of these inflammatory mechanisms independent of VEGF become increasingly relevant to chronic or refractory DMO where the effectiveness of anti-inflammatory drugs, such as intravitreal corticosteroids, has been proved to surpass antiangiogenic treatment [29,58,59,60,61].

If central macular thickness (CMT) is persistently augmented and VA does not improve despite antiangiogenic treatment, DMO is classified as refractory, which implies the perpetuation in time of damage mechanisms feeding themselves in a sustained inflammatory state [23,62]. In fact, the longer the DMO has been present, the poorer the antiangiogenic response is [63]. Retinal disorganisation, persistent large amounts of sub-macular fluid and large macular cystoid spaces in optical coherence tomography (OCT) are also predictors of a poorer visual prognosis and treatment response [63,64,65].

## 4. Therapeutic Options for DMO

Once DMO is established and sight-threatening, therapeutic approaches are limited. Intravitreal antiangiogenic agents and corticosteroids, laser therapy and surgery, are the available options, alone or in combination [15,33,58,66]. Laser photocoagulation and pars plana vitrectomy surgery are suitable for certain selected patients when maximal medical therapy is insufficient or when there is a significant tractional component in DMO, respectively [15,33,66].

### 4.1. Anti-VEGF

Nowadays, anti-VEGF agents are the first-line therapy in DMO treatment, which include ranibizumab (Lucentis^®^; Genentech, San Francisco, CA, USA/Novartis Ophthalmics, Basel, Switzerland), bevacizumab (Avastin^®^; Genentech, San Francisco, CA, USA) and aflibercept (Eylea^®^; Regeneron, Tarrytown, NY, USA), the most widespread intravitreal therapies for this condition. Their effectiveness with different protocols has been widely studied in comparative studies and separately in RCTs [8,60,67,68,69,70]. For instance, protocol T from the Diabetic Retinopathy Clinical Research Network (DRCRN) established an administration pattern for an intravitreal drug every four weeks until cure or improvement of best-corrected visual acuity (BCVA) and structural optical coherence tomography (OCT) thickness parameters or as a defect, until a worsening in some of them. Whether the treatment is successful or not, from the 24-week visit on, a lack of response to the last two injections means treatment should be discontinued. In addition, intravitreal treatment must still be combined with focal or grid laser treatment when it is considered necessary [69]. When this protocol was followed for follow-up over two years, a mean clinical improvement in DMO patients has been demonstrated [67,70,71].

This approach implies repeated intravitreal administration of antiangiogenic agents for an extended period, which means a well-known burden for patients and the possibility of significant side effects. In addition, some studies have suggested that sustained anti-VEGF suppression could have undesirable consequences, such as capillary dropout, retinal thinning, ganglion-cell damage and a certain predisposition to retinal atrophy [72,73]. Moreover, intravitreal injections could produce an angio-fibrotic switch because of the low VEGF levels resulting in tractional complications because of fibrosis stimulation and epiretinal membrane formation [74].

### 4.2. Switching Strategy

Choosing between different anti-VEGF therapies to initiate DMO treatment requires the consideration of many factors, including cost and efficacy [66]. Switching strategy implementation must be taken into account in cases unresponsive to the initial regime with a monthly intravitreal antiangiogenic drug at three or six months follow-up when either a persistent CMT on OCT, a visual acuity under 20/40, or an improvement of VA of less than a line from baseline, is present [75].

This strategy is a simple approach to consider in cases of initially persistent DMO and has been shown to reduce CMT and improve visual acuity [76,77,78].

### 4.3. Corticosteroids and Combination Therapy

A non-negligible percentage of DMO patients do not respond to antiangiogenic treatment, although there is a lack of consensus regarding when to consider a DMO to be persistent, resistant or refractory [60,79,80]. For example, DRCRN and RISE/RIDE papers defined persistent DMO as those eyes under monthly intravitreal treatment for at least 6 months with a central subfield thickness (CST) measured by OCT of above 250 µm [8,60]. In contrast, other research groups considered refractory DMO to occur when there was no response to the last three monthly anti-VEGF injections, considering a lack of response to be a worsening of BCVA according to two early treatment diabetic retinopathy studies (ETDRS) or when there was a reduction of less than 10% or 50 µm in CST [61,79,81,82].

Whether one criterion or the other is considered, resistant or persistent DMO is still considered a relevant proportion of DMO patients. In protocol T from DRCRN, 40% of patients suffer from persistent DMO within 24 weeks of treatment [60,70], and 30–40% of patients did not improve visual function in three years of therapy in the RISE/RIDE study [13]. In this framework, from the 1970s, intravitreal corticosteroid treatment has been studied in DMO with positive outcomes due to the potential role of inflammation on its pathogenesis and because some of them seem to also have anti-VEGF properties [83,84,85]. Triamcinolone acetonide (TA) was the first corticosteroid used for this purpose [52].

Bearing in mind the inflammatory component in DMO, corticosteroids, either used alone or in combination with specific anti-VEGF therapy, are a current therapy that can help us individualise treatments [52,66,81]. Intravitreal dexamethasone implant (0.7 mg) (Ozurdex^®^, Allergan, Inc., Irvine, CA, USA) treatment has been demonstrated to produce an improvement in structural parameters, mainly CMT and CST, alone or in combination with anti-VEGF therapy [61,79,81,85,86]. Its posology is more convenient since it consists of a biodegradable solid polymer drug-delivery system whose efficacy has been demonstrated up to six months after its administration [84,87]. Fluocinolone acetonide (FA) (0.19 mg) is the content of another nonbiodegradable implant (Iluvien^®^, Alimera Sciences, Alpharetta, GA, USA) which can release the drug for up to three years, and FAME studies demonstrated its efficacy particularly in long-term chronic DMO [47,52].

However, a problem remains in that corticosteroids have not shown an improvement in visual acuity superior to VEGF, neither in naive nor in persistent DMO, since the multifactorial origin of DMO seems to require a multi-target treatment. Moreover, they are associated with substantial side effects, such as the development of cataracts or glaucoma, especially in predisposed patients [66,81,84,86,88].

### 4.4. Surgery

In DMO with vitreomacular adhesions, proliferation, or both, surgical options should be considered [66]. If mechanical factors are present, vitrectomy with or without epiretinal membrane removal could facilitate oedema resolution, oxygen diffusion to the retina and VEGF clearance [66,89]. In contrast, intravitreal drugs seem to last a shorter amount of time inside the vitreous cavity in vitrectomised eyes [90].

### 4.5. Healthcare Burden

As mentioned above, these therapies require a great number of injections and multiple visits with the use of sophisticated exploration equipment over a long-term period, with a considerable burden on health and socio-economic systems. Considering that approximately 21 million people suffer from DMO worldwide, with bilateral involvement in 33–47% of patients since DM is a systemic and chronic disease [83,91,92] and with the cost per dose of intravitreal treatment varying from around USD 50 for bevacizumab to USD 1950 for aflibercept or USD 1400 for Ozurdex^®^ in the United States [93,94,95], the economic impact on the health system is not negligible. Indeed, the DRCRN studied the cost-effectiveness of intravitreal antiangiogenic treatment in DMO according to their protocols. One-year costs per participant varied from around USD 4100 from bevacizumab to USD 26,100 from aflibercept [94].

Considering these facts and the administration protocols, DMO accounts for not only a substantial percentage of patients at ophthalmological retinal consultations but enormous care and economic load for any healthcare system, unaffordable for a single person [96]. It is an issue of grave concern, and its application in a sustainable manner, selecting the appropriate drugs and dosage, is quite popular nowadays [93,94,96,97,98,99]. Therefore, new treatment modalities exploring new pathways that could overcome the shortcomings of current therapies are required.

## 5. Rho Kinase and ROCK Pathways

The Rho subfamily belongs to a member of the small molecule G protein in the Ras superfamily and has GTPase activity. It acts as a molecular switch (from the inactive state (bound to GDP) to the activated state (bound to GTP), exerting various biological effects by binding to its downstream target effector molecule (Figure 2). Among them, Rho-associated kinases (ROCK) are the most important effector molecules of the serine-threonine family that participate in the downstream signalling of Rho GTP-binding proteins [100,101,102]. There are two homologous isomers in the cell: ROCK1 (ROCK-I, ROKβ) and ROCK2 (ROCK-II, ROKα). Both are involved in the regulation of cell morphology, polarity, and cytoskeletal remodeling by regulating actin and cell migration, e.g., ROCKs phosphorylate myosin light chain (MLC) and LIM kinases to change actin cytoskeleton [103,104,105].

The ROCK1 and ROCK2 genes are located in the 18q11 and 2p24 chromosome regions. In addition, ROCK2 has a splicing mutant (mROCK2), called the small ROCK, which appears to be a product of partial gene duplication.

ROCK2 and mROCK2 are mainly localized to the cytoplasm, but ROCK2 is also localized to the plasma membrane through the C-terminal region. ROCK2 is highly expressed in the brain and heart. However, ROCK1 can be concentrated in the plasma membrane of endothelium or in the center of the microtubule tissue of the moving cells, indicating its involvement in cell migration. ROCK1, is highly expressed in lung, liver, spleen, and kidney [106,107,108,109]. Both ROCK isomers are also present in ocular tissues and aberrant regulation of ROCK levels plays a role in the pathogenesis of corneal wound healing, glaucoma, diabetic retinopathy, and AMD [110,111,112].

### 5.1. Rock Signalin Pathway

ROCKs can be activated by the GTP-bound form of Rho and the activated form then phosphorylates downstream targets [110]. Binding of Rho-GTP opens the loop formation of the enzyme, and the activated form then phosphorylates downstream targets (Figure 2). Then, the activated ROCK can directly phosphorylate MLC, LIM Kinases, intermediate filament proteins, among other substrates, and in most cases, substrates are phosphorylated by their respective ROCK proteins. Taken together, Rho pathway activation leads to a concerted series of events resulting in increased actin-myosin contractility and cytoskeletal change. The C-terminus of the ROCK protein is a self-inhibiting region of kinase activity, including the RBD region and the PH region. In the inactive state, the RBD and PH regions of the C-terminus of the ROCK protein interact with the kinase domain to form a self-inhibiting loop (Figure 2) [106,107,108,109,110].

### 5.2. Relationship with Diabetic Retinopathy

Recently, RhoA and ROCK have been implicated in the pathogenesis of DR. The long-term glucose homeostasis imbalance, which leads to an accumulation of advanced glycation end products in the vessels. Under these inflammatory and hyperglycaemic diabetic conditions, guanine exchange factors (GEF) activate Rho, and subsequently activating ROCKs [105]. This abnormal expression of ROCK protein plays an important role in DMO pathogenesis, acting on retinal microvessels, leading to structural changes including vascular rigidity promoting leucocyte adhesion to the microvasculature by intercellular adhesion molecule-1 (ICAM-1) and contributing to VEGF-elicited microvascular hyperpermeability [113] (Figure 2). In addition, the activation of the Rho/ROCK signalling pathway regulates the NF-κB signalling pathway, which upregulates inflammatory genes, fibronectin and transcription factor AP-1 activation and accumulation of glomerular matrix proteins. Upregulation of Rho/ROCK pathway also results in the dephosphorylation of endothelial nitric oxide synthase (eNOS), which induces endothelial cell apoptosis and vasoconstriction [110,111,112,114].

Considering that increased activity of the Rho/ROCK pathway in diabetic patients contributes to exacerbates retinal vessel permeability, macular oedema, vascular occlusion and retinal ischaemia, several in vitro and in vivo studies have been conducted to assess the effect of Rho/ROCK pathway inhibition on diabetic retinopathy and DMO (Table 1 and Table 2).

### 5.3. Involvement of Rho/ROCK Pathway in the Pathogenesis of DR

As mentioned before, the pathogenesis of DMO is multifactorial and complex involving several mechanisms in which Rho/ROCK pathway is actively engaged. Recent studies have revealed that Rho and its target protein ROCK are implicated in important pathological pathways in DR and DMO, such as hyperglycaemia, RPE disturbances, vasoconstriction, endothelial impairment, leukostasis and inflammation, and vascularisation (Table 1 and Figure 3) [110,111,112,114,115].

#### 5.3.1. Hyperglycaemia

Hyperglycaemia is one of the most important stimuli for ROCK-1 activation, and the correlation between diabetes and microvascular endothelial cell dysfunction has been demonstrated in various studies before. e.g., on choroid-retinal endothelial cell lines, it has been shown by a meaningful increase in phospho-myosin phosphatase target protein-1 (p-MYPT-1) as a result of this overactivity in response to elevated glucose [116]. Rho activity is also involved in the pathogenesis of renal and aortic complications during diabetic states [117,118]. In addition, hyperglycaemia has been shown to increase the expression of monocyte chemoattractant protein-1 (MCP-1/CCL2) and VEGF by vascular endothelial cells and in the vitreous samples from patients with DR [119,120]. It has been reported that the inhibition of RhoA/ROCK1 may attenuate the hypertonicity of endothelial cell caused by high glucose microenvironment, partly block inflammation and prevented considerably the apoptosis of endothelial cells aroused by high glucose [111,112,115,116].

#### 5.3.2. Inflammation and Vascularisation

The levels of the proinflammatory mediators are increased in the ocular microenvironment of patients with DR, suggesting that persistent inflammation is critical for DR initiation and progression.

Several studies have documented that diabetes enhances the production of inflammatory mediators such as p-ERK ½, p-NF-κB, iNOS, VEGF and MCP-1/CCL2), and how Rho-kinases play a crucial role in the inflammatory signalling, proliferation, fibrosis, and apoptosis through mitogen-activated protein kinase (MAPK) p38MAP kinase and NF-κB activation. Activation of NF-κB induces the upregulation of leukocyte adhesion molecules and the production of proinflammatory cytokines and angiogenic factors [119].

As mentioned, inflammatory factors play a key role in DMO pathogenesis, and their own signalling pathways and expression are influenced by Rho kinases. In vitro assays have also been conducted to prove this relationship using human retinal Müller glial cells and microvascular endothelial cells, ex vivo retinal explants, bovine retinal endothelial cells and rat retinal cells [117,121,122,123,124]. Results of recent research indicate that ROCK-1 activation induces focal vascular constrictions, endoluminal blebbing, retinal hypoxia, and remodelling of RPE cells contributing to outer barrier breakdown [117,125]. The blebbing-induced closure reversed by ROCK inhibitors, could open a window for intervention in case of macular ischemia.

VEGF stimulates ROCK activity and rho/ROCK pathways, particularly those mediating VEGF-dependent angiogenic processes on proliferative and non-proliferative DR, i.e., Rho/ROCK pathways regulate VEGF expression induced by platelet-derived growth factor (PDGF)-BB in retinal vessel pericytes [121,122,124]. Rho-kinase signalling is also involved in the injury of inner retinal cells such as Müller cells in the case of hypoxia and oxidative stress [123].

#### 5.3.3. RPE Disturbances

Diabetic hyperglycaemia, TNF-α, and TGF-β activate and increase the ROCK pathway at endothelial and RPE cells and hyalocytes [116,126]. ROCK-1 proteins are usually cytoplasmic proteins in rat and human RPEs and endothelial cells, but in diabetic induced models, it has been demonstrated to translocate to the membrane accompanying ROCK-1 positive cell membrane bleb formations. By contrast, ROCK-2 localisation remained undisturbed with staining at the cytoplasmic membrane [125]. At the RPE level, this translocation implies a remodelling of the actin cytoskeleton and a modification and opening of tight cell junctions, so crucial for RPE functions and participating in the outer BRB breakdown [27,117,125]. In addition, relocation and inactivation of ROCK-1 and an improvement of RPE barrier function after fasudil (HA-1077) injection have been reported in vitro assays supporting Rho-pathway involvement in DMO pathogenesis [125].

#### 5.3.4. Vasoconstriction

Vasoconstriction is an important mechanism for retinal regulation and for proper functioning, which is damaged in DMO. Rat, porcine and human retinal models and in vitro and in vivo assays have shown Rho participation in this damage by multiple mechanisms. ROCK signalling is involved in retinal arteriolar and venular constriction due to vascular endothelin-1 (ET-1), a potent vasoconstrictor secreted by vascular endothelial cells [127,128]. In fact, expression of ROCK-1 was increased in mural cells, and there exist ROCK-1 positive blebs on the vessel wall of constricted areas from diabetic rats, narrowing retinal vessels lumen [125]. It has been shown promising results in hypoxia-induced retinal neovascularisation animal models just using topical ROCK inhibitors. This study showed that the topical ROCK inhibitor inhibited neovascularisation and could enhance normal retinal vascularisation, opening a new horizon for treatment of DR without affecting normal retinal vasculature or reducing the neuroretinal damage [121].

Its contribution also has been demonstrated in simvastatin-induced nitric-oxide-mediated dilatation of retinal arterioles [129]. The eNOS enzyme and its phosphorylation appear to be downregulated in retinal endothelial cells, and fasudil treatment also has been proved to have a positive effect in reversing the decreased expression of this enzyme, but the production of nitric oxide (NO) is required to be maintained for fasudil to work [126].

#### 5.3.5. Endothelial Impairment

ROCK-1 translocation and blebbing and an increased level of activated Rho in endothelial cells were also found in diabetic rat retinas producing endothelial impairment and the DMO [125,126]. Rat and human models have shown that under hyperglycaemic conditions and ROCK-1 activation, which occurred in the RPE, occludin, ZO-1 and claudin-5 protein levels are reduced, destabilising tight junctions in endothelial cells and causing inner-BRB disruption [9,116].

#### 5.3.6. Leukostasis

CD11b/CD18 is a beta-2 integrin, a leukocyte-specific transmembrane protein participating in neutrophils adhesion, communication and migration, and they bind other adhesion molecules such as VCAM-1, ICAM-1 or certain complement components [130]. They have been proven to be upregulated in retinal vessels from rat diabetic models since an augmented expression of ICAM-1, an augmented fluorescence of beta-2 integrins and a higher number of neutrophils bound to harmed endothelial cells with respect to controls were found in in vitro assays [126,130]. Rho pathway activation stimulates ICAM overexpression and facilitates leukocyte adhesion and leukostasis by producing ICAM clustering and activating anchoring proteins at endothelial cells by phosphorylation [125,126]. The adhesion and stasis of leukocytes have been shown to produce diabetic microvascular damage and endothelial dysfunction contributing to DMO pathogenesis [56]. Intravitreal fasudil, ripasudil (K-115) and AMA0428, ROCK inhibitors suppressed and reduced leukocyte migration, inhibited ICAM-1 expression in endothelial cells and decreased vascular leakage and neurodegeneration of diabetic retinal vessels, thereby improving retinal thickness and DMO [126,131,132,133].

The eNOS enzyme and its phosphorylation appear to be downregulated in diabetic retinal endothelial cells, and fasudil treatment has also proven to have a positive effect in reversing the decreased expression of this enzyme, but the production of nitric oxide (NO) is required to be maintained for fasudil to work [106,107,108,109,110,111,114].

## 6. From the Bench to the Bedside

Anti-VEGF therapy in DMO treatment has changed the management of this condition, improving the control of the disease. However, anatomical and visual results are not always as expected, and some cases remain refractory to this treatment. Therefore, the identification of adjuvant or alternative therapies is required, the modulation of Rho/ROCK pathway is a potential candidate, since the suppression of leukocyte adhesion and neutrophil-induced retinal endothelial cell damage has been shown with its inhibition, which could have beneficial effects in terms of anatomical and visual outcomes. A summary of studies in which ROCK inhibitors are used as a treatment was provided in Table 2.

Since its discovery in the second half of the 1990s, the modulation of rho kinase has been considered a potential therapeutic target to explore in a broad range of human diseases, being cardiovascular disease one of the main areas in which they have been studied; unfortunately, its inhibition also carries the risk of unacceptable systemic side effects. In this framework, the eye is a possible application area due to the limited number of side effects, being the conjunctival hyperemia the main one.

Of note, the protein kinase family inhibitors have been historically classified according to their mechanism of action based on the binding modes, leading to a type I−V classification [134]. Although, a thorough analysis of the different mechanism of actions goes beyond the scope of this review, is essential to take into consideration that the majority of kinase inhibitors are based on type I mode, in which the inhibitor targets the ATP binding site of the kinase in its active conformation [104,134,135]. Another practical mode of classification is how they impact the kinase parameters, their difference substrate, and their competitive or non-competitive profile [104]; being a matter of controversy, what is the best classification method.

Regarding the eye, the vast majority of rho-kinase inhibitors act on both, ROCK-1 and ROCK-2. Fasudil (HA-1077, Fasudil, Asahi Kasei Pharma Corporation, Tokyo, Japan), an isoquinoline derivate and a calcium antagonist, is approved for the treatment of cerebral vasospasms in Japan and China; its fluorinated analogue, with much more power and selectivity. Ripasudil (K-115, Glanatec^®^ Ophthalmic Solution 0.4%, Kowa Company, Ltd., Japan) and netarsudil (AR-13324, Rhopressa^®^ Ophthalmic Solution 0.02%, Aerie Pharmaceuticals, Inc., Irvine, CA, USA), an amino-isoquinoline amide, act on both kinases and are the best-known ones. The three of them have been approved for glaucoma treatment because of their ability to reduce intraocular pressure-lowering aqueous outflow resistance [105,136,137]. Only Belumosudil (KD-025) is a selective inhibitor of ROCK-2 (phase 2 clinical trials) being considered as a potential treatment for patients with some systemic inflammatory conditions [138]. AMA-0076, GSK-269962A, H-1152, Y-27632, RKI-1447, SAR-407899, SB-772077-B, Wf-536 andY-39983, are also double-acting ROCK inhibitor molecules [105,136].

It is crucial to take into consideration the mode of administration as most of the glaucoma preclinical and clinical studies are based on topical applied ophthalmic solutions, having been demonstrated their safety profile and hypotensive properties but with a high incidence of conjunctival hyperaemia and with limited intraocular bioavailability due to the short corneal residence time and their characteristic hydrophilic properties [137,139,140,141,142]. This is one reason under the high rate of studies based on intravitreal injections for pathologies involving the posterior segment of the eye, trying to increase the intraocular active-drug concentration [143,144].

The first report in the literature of the use of ROCK inhibitors for the treatment of refractive DMO came with Ahmadieh et al. [133], who combined the intravitreal injection of bevacizumab with fasudil in five eyes of five patients with persistent DMO, showing a significant improvement in BVCA and CMT after six weeks, without toxic effects on the retina. These results were in concordance with further studies, which showed the possible positive effect of intravitreal injections of fasudil in combination with bevacizumab in refractory cases [132]. Although both studies decided to use 0.025 mg of fasudil since previous animal studies demonstrated that there were no electrophysiological or morphological toxic effects in the retina with a concentration of 100 μM [145], the optimal dose of fasudil for its use intravitreally remains unknown, and further studies are needed to solve this question.

In addition, it has also been hypothesised that these drugs could be used not only for refractive cases but also for other cases of DMO. For instance, Ahmadieh et al. [146] recently published the outcomes of a prospective randomised clinical trial (NCT01823081). It showed the beneficial effects of intravitreal fasudil in combination with bevacizumab compared to bevacizumab monotherapy in patients with severe DMO. In this trial, a loading phase of three-monthly injections was administered. BCVA and CMT were evaluated, at 3 and 6 months, and the beneficial effects on BCVA and CMT were more prolonged in the group of patients treated with the combined therapy of bevacizumab and fasudil; the percentage of cases with an increase of 15 or more ETDRS letters was clinically more pronounced in the combined group at the 6-month follow-up.

In this framework, Minami et al. [147] decided to retrospectively evaluate the potential benefits of ripasudil, which is similar to fasudil with a more selective mechanism and has been approved for its use as a topical treatment in patients with glaucoma or ocular hypertension (OH) and in patients with glaucoma or OH and associated DMO. Their outcomes revealed that at the 1-month follow-up, the IOP and foveal thickness of patients treated with ripasudil decreased significantly, although the BCVA did not show any significant improvement. Of note, they were aware of the limitations of the study, as it was a retrospective study with a small number of patients and a short follow-up period, and the drug was administered topically, with implies a limited intraocular bioavailability. However, it should be considered for the development of further prospective clinical trials.

As previously explained, netarsudil, appears to be effective in lowering IOP, but with low intraocular bioavailability when administered topically [136]. Interestingly, netarsudil has particular physiochemical and pharmacological properties, as its more lipophilic profile, which makes it a suitable candidate for intravitreal administration. AR-13503, a prodrug of netarsudil with ROCK/PKC inhibition properties, is being investigated for its administration intravitreally in a biodegradable implant [148]. In vivo studies showed a safety profile and an improved bioavailability, being able to deliver adequate levels of AR-13503 to the posterior segment for a 5–6 month-time period [149].

Moreover, alone or in combination with anti-VEGF drugs, AR-13503 has shown an additive efficacy on antiangiogenic properties and RPE barrier functions protecting power on DR in vitro models [148,149]. These characteristics make AR-13503 implant a possible game-changing drug for the treatment of DME. The prodrug is ongoing a phase 1 safety testing clinical trial in DME and Neovascular Age-Related Macular Degeneration patients, which has not finished yet (NCT03835884). Since the studies in this review are focused on DMO, questions arise as to whether ROCK inhibitors could be of benefit in other types of macular oedema, either as unique therapy or in combination with other therapies. It is interesting to note that the potential use of the intravitreal injection of bevacizumab and fasudil in refractory cases of macular oedema secondary to retinal vein occlusion is under investigation, although the results have not been published yet (NCT: 03391219).

## 7. Concluding Remarks

DMO is the leading cause of visual impairment in diabetic patients. It has a multifactorial and complex pathogenesis, in which the rupture of the BRBs is the primary event, VEGF being the main molecule related to this process.

Currently, anti-VEGF therapies are the first-line treatment, having changed the prognosis and management of the condition. However, this therapy has shortcomings as it does not resolve all DMO, and it requires a considerable number of injections and follow-up appointments, placing a severe burden on the healthcare system. Although there are other therapies for these refractory cases, such as corticosteroids, laser or surgical treatments, up to 30% of patients are resistant to conventional treatment. Therefore, the development of new therapies which could improve the management of this condition is needed.

Understanding the role of the Rho/ROCK signalling pathway in the pathogenesis of DMO has increased interest in this field. Under the hypothesis that its modulation could benefit some patients with DMO, several studies have tried to unveil its potential role in treating this condition, showing favourable results. However, there is still some controversy regarding the best route of administration, the optimal dose, or what cases are more suitable for this treatment.

In conclusion, although some questions remain unsolved and further studies are needed, it seems that the modulation of the Rho/ROCK pathway could be game-changing in the management of patients with DMO refractory to conventional treatment.

## Figures and Tables

**Figure 1 cells-10-01683-f001:**
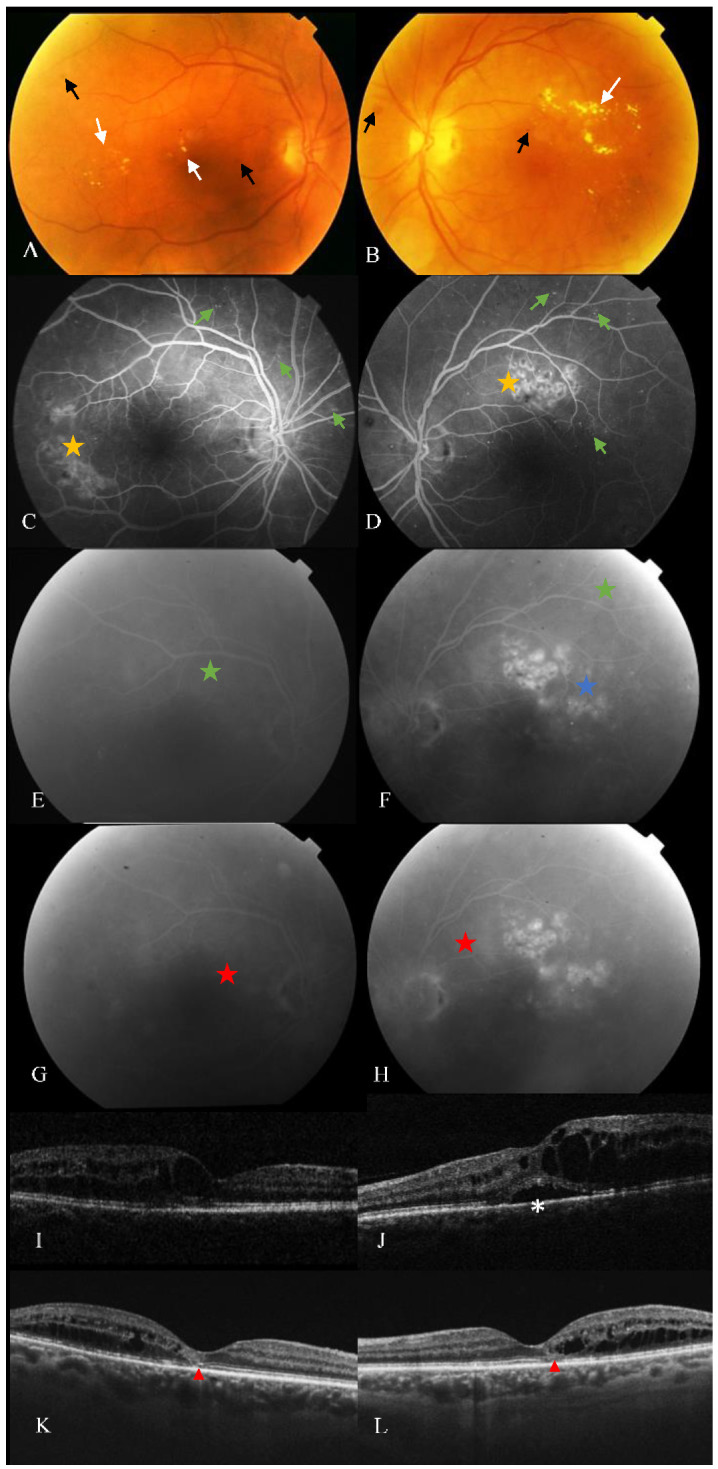
Retinal alterations despite sustained intravitreal therapy for DMO. Color fundus photography (**A**,**B**), fluorescein angiography (FA) (**C**,**H**), and spectral-domain optical coherence tomography (SD-OCT) (**I**,**L**) of a patient with nonproliferative diabetic retinopathy and refractory diabetic macular oedema (rDMO). (**A**,**B**) Hard exudates (top left and right, white arrows), microaneurysms and retinal hemorrhages (black arrows). (**C**,**D**) Early phase of FA showing multiple hyperfluorescent spots predominantly from areas with microaneurysmatic changes (green arrows) and hyperreflectivity of focal laser scars (yellow stars). (**E**–**H**) Late-phase of FA of the same patient showing diffuse leakage from microaneurysm (green stars), circinate areas (blue stars) towards and within the foveal avascular zone corresponding to rDMO (red stars). (**I**,**J**) SD-OCT displaying multiple intraretinal cysts and subfoveal neuroretinal detachment (white asterisk). (**K**,**L**) B-scans of the same patient, 14 years after multiple treatments, showing refractory cystoid macular oedema, multiple hyperreflective dots and disruption of external retinal layers (red arrowheads).

**Figure 2 cells-10-01683-f002:**
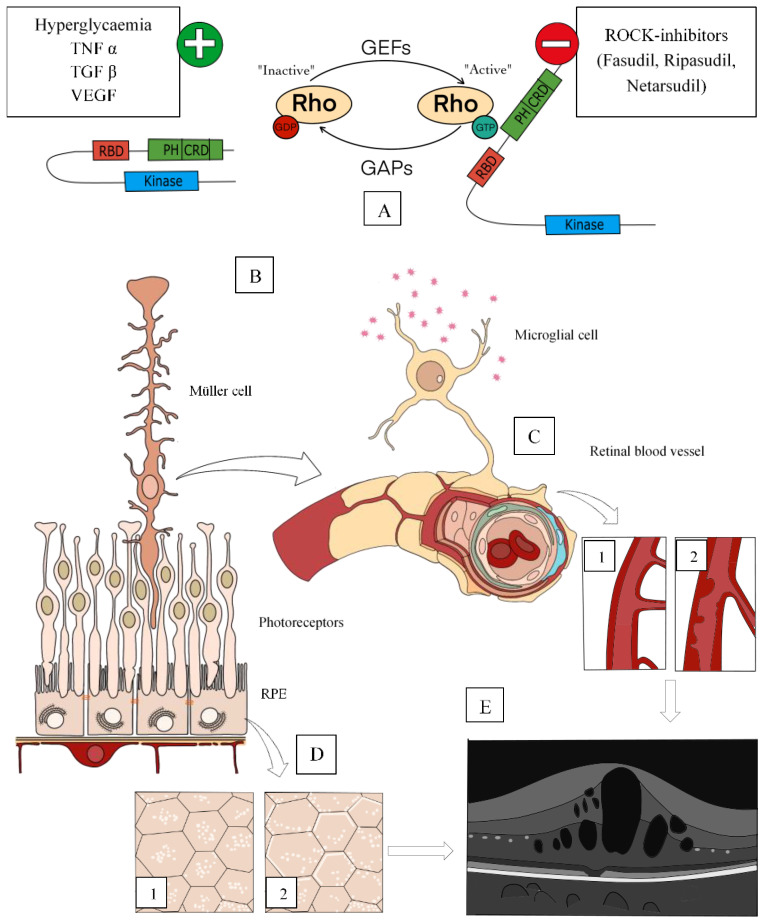
Rho activation and participation in DMO pathways. (**A**) In response to hyperglycaemia and in the presence of cytokines (TNF-α) and growth factors, such as VEGF and TGF-β, the activation of Rho by guanine exchange factors (GEF) is produced. GTPase-activating proteins (GAP) stimulate endogenous GTPase activity, facilitating GTP hydrolysis in GDP and inactivating Rho. (**B**) On the left side, a normal-structured external retina is represented, and on the right side, an anomalous neurovascular junction in DMO. Rho-kinase signalling pathways are upregulated in Müller glial cells in response to hypoxia and oxidative stress and take part in DMO pathogenesis. (**C**) Representation of vessels from a normal retina (**C1**) and a diabetic one (**C2**). within inner-retinal vessels, rho pathways are involved in retinal vasoconstriction, endoluminal blebbing, endothelial impairment and leukostasis in response to an increased inflammation state and ICAM-1 and VCAM-1 hyperproduction in DMO. (**D**) At RPE level, ROCK participates in bleb formation and cellular remodulation and produces polarization and cell adhesion changes in response to hyperglycaemia modifying outer-BRB. In normal RPE cells, ROCK-1 is cytoplasmic (**D1**), but in diabetic conditions, it is recruited to the membrane (**D2**). (**E**) Representation of an OCT image showing a cystoid DMO with hyperreflective dots as a result of the mentioned patho-physiological abnormalities. Rho kinase and ROCK pathways. The ROCK molecular structure consists of three parts, the N-terminal kinase region, that catalyzes the phosphorylation/dephosphorylation of a series of downstream substrates; a coiled-coil region, in which the Rho-binding domain (RBD) regulates the activation signal of Rho transduction; and a C-terminal region terminal, which contains a pleckstrin homology (PH) and cysteine-rich domain (CRD). This region contains a self-inhibiting region that interacts with the kinase domain to inhibit ROCK activity [106,107,108,109].

**Figure 3 cells-10-01683-f003:**
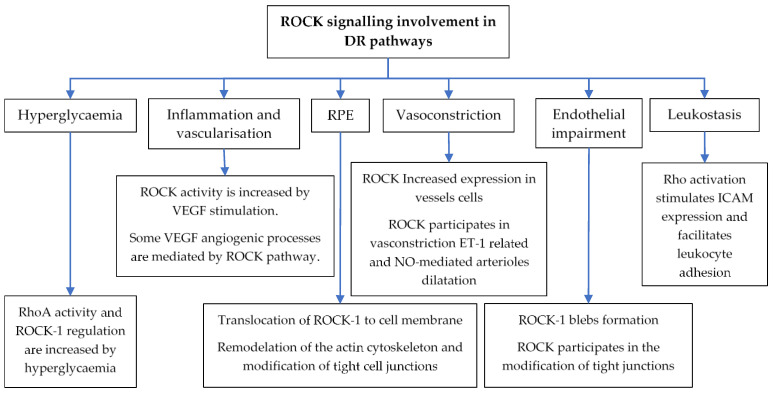
ROCKs involvement in the mechanisms underlying DMO according to the results of preclinical studies. Together with inflammation, elevated glucose, which is a thorn in DR, constitutes a strong stimulus for Rho pathways activation and ROCK-1 upregulation. Moreover, VEGF overproduction because of DR, also increases ROCK activity, particularly those pathways in charge of VEGF-mediated angiogenic processes, getting caught in a vicious circle. At RPE level, ROCK-1 cytoplasmic proteins translocate to the cell membrane and participate in actin cytoskeleton and tight cell junctions remodulation. At a vascular level, the calibre of the retinal vessels is also regulated by ROCKs related pathways. In addition, they contribute to endothelial impairment by ROCK-1 blebs formation and tight junction modifications, as happened at RPE. Finally, these kinases stimulate ICAM expression, which facilitate leukocyte adhesion and contributes to leukostasis. Thus, ROCKs take part in leukostasis, inflammation and vascularisation processes and participate in endothelial and RPE impairment, perpetuating DMO pathogenesis.

**Table 1 cells-10-01683-t001:** ROCK signalling involvement in diabetic retinopathy pathway.

Author, Year	Type of Study	Model	Outcomes
Hyperglycaemia			
(Lu et al., 2014)	In vitro	Choroid-retinal endothelial cell line	High level of glucose increased RhoA activity
(Mohammad et al., 2018)	In vitro	Human retinal Müller glial cells	High glucose-induced upregulation of ROCK-1 in human retinal Müller glial cells
(Yao et al., 2018)	In vitro	High-glucose-induced human retinal endothelial cells	Rho/ROCK signalling pathway is involved in the hyperglycaemia-induced microvascular endothelial dysfunction
Inflammation and vascularisation			
(Mohammad et al., 2018)	In vitro	Human retinal Müller glial cells	High glucose-induced upregulation of p-ERK ½, p-NF-κB, iNOS, VEGF and MCP-1/CCL2
(Yamaguchi et al., 2016)	In vitro	Human retinal microvascular endothelial cells	Increase in ROCK activity by VEGF stimulation
(Bryan et al., 2010)	Ex vivo/In vitro	Ex vivo retinal explants Bovine retinal endothelial cells	Rho/ROCK pathway mediates different VEGF-mediated angiogenesis processes
RPE			
(Rothschild et al., 2017)	In vivo	GK rat model	Diabetic conditions modified the subcellular distribution of ROCK-1, which was located along the membrane, while ROCK-2 localisation was undisturbed
		Human retina tissue of diabetic patients	Diabetic conditions modified the subcellular distribution of ROCK-1, which was located along the membrane, while ROCK-2 localisation was undisturbed
Vasoconstriction			
(Rothschild et al., 2017)	In vivo	GK rat model	Increased expression of ROCK-1 in mural cells and in the vessel wall of constricted areas
(Chen et al., 2018)		Porcine models	ROCK signalling is involved in the venular constriction due to ET-1
(Rosa 2009)	In vitro	Human retina	ET-1 provokes vasoconstriction of retinal arterioles via ROCK-signalling pathway
(Nagaoka et al.,, n.d.)	In vitro	Porcine retinal arterioles	Rho/ROCK pathway takes part in the simvastatin-induced NO-mediated dilatation of retinal arterioles
Endothelial impairment			
(Rothschild et al., 2017)	In vivo	GK rat model	ROCK-1 expression in endothelial cell blebs.
		Human retina tissue of diabetic patients	
(Lu et al., 2014)			Under hyperglycemic conditions, ROCK-1 contributes to endothelial dysfunction, through the damage of tight junctions
(Arita et al., 2009)	In vitro	Diabetic rat model	Increased levels of activated Rho in diabetic rat retinas
Leukostasis			
(Arita et al., 2009)		Diabetic rat model	CD11b and CD18 fluorescence in neutrophils of diabetic rats were higher, while CD11a fluorescence intensity did not differ
			ICAM expression was significantly higher in diabetic retinas

GK—Goto-Kakizaki, ET-1—Endothelin-1; CD—cluster of differentiation; MCP-1—Monocyte Chemoattractant Protein-1.

**Table 2 cells-10-01683-t002:** ROCK inhibitors as a therapeutical approach in patients with DMO. Summary of clinical studies.

Author, Year	Type of Study	No. of Patients (Eyes)		Inclusion Criteria	Therapy Evaluated	Parameters Assessed	Outcomes
		Case	Control				
(Ahmadieh et al., 2013)	Prospective pilot study	5 (5)	-	Persistent DMO, previously treated with: Macular laser photocoagulation Intravitreal bevacizumab injections	Intravitreal fasudil injection (0.025 mg/0.05 mL) combined with Intravitreal bevacizumab injection (1.25 mg/0.05 mL)	BCVA CMT	No toxic retinal effect was found BCVA improved from 0.82 to 0.34 logMAR (*p* = 0.04) and CMT improved from 409 to 314 µm (*p* = 0.04) avg. at week 6
(Nourinia et al., 2013)	Prospective pilot study	15 (15)	-	Refractory DMO, previously treated with: Intravitreal bevacizumab injections	Intravitreal fasudil injection (0.025 mg/0.05 mL) combined with Intravitreal bevacizumab injection (1.25 mg/0.05 mL)	BCVA CMT	BCVA improved from 0.84 to 0.49 log-MAR (*p* = 0.003) and CMT improved from 448 to 347 µm (*p* = 0.001) avg. at week 4
(Ahmadieh et al., 2019)	Prospective randomised clinical trial	22 (22)	22 (22)	Severe DMO	Intravitreal fasudil injection (50 µM/L) combined with Intravitreal bevacizumab injection (1.25 mg/0.05 mL)	BCVA CMT	BCVA changed in 12 ETDRS letters more at month 6 in combined group vs. bevacizumab group (*p* < 0.001). CMT was also significantly more pronounced at month 6 in combined group (*p* < 0.001)
(Minami et al., 2019)	Retrospective study	10 (12)	10 (14)	Patients under ripasudil treatment for PAG with DMO	Ripasudil eyedrops	BCVA FT IOP	Decreased IOP (2.7 mmHg less; *p* = 0.008) and FT (44 µm less; *p* = 0.01) after one month of treatment No significant changes were found in the BCVA

BCVA—Best Corrected Visual Acuity; CMT—Central Macular Thickness; PAG—Primary Angle Glaucoma; IOP—Intraocular Pressure; FT—Foveal Thickness; DMO—Diabetic Macular Oedema; avg.—on average.

## Data Availability

Not applicable.

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
