# Peer review of "Rho-Kinase Inhibitors for the Treatment of Refractory Diabetic Macular Oedema"

_cells, 2021, doi:10.3390/cells10071683_

Round 1
Reviewer 1 Report
The review describes the recent progress on the role of Rho/ROCK signaling pathway as possible target for the treatment of diabetic macular edema.
The article is a useful brief summary of the latest results in this context. The manuscript is well written. The citations are adequate. However, some basic processes are superficially explained.
Considering that the section 3 is related to the pathophysiology of diabetic macular oedema, the title of some subsections appears inadequate. In particular, change the title of section 3.2. Blood-retinal barrier in 3.2. Blood-retinal barrier breakdown. Moreover, the authors have to better describe the event underlying the blood-retinal barrier breakdown.
Please change the title of section 3.3. Retinal pigment epithelium in Retinal pigment epithelium damage.
In the section 3., the authors should add a paragraph regarding the hypoxic event, considering that beside inflammation, hypoxia is a key player in the progression of this pathology.
The authors should better deepen the paragraph regarding the use of ROCK inhibitors in the treatment of DMO.
Author Response
To whom may concern,
First and foremost, thank you for taking the time to assess our manuscript. We addressed all the concerns you raised and appreciated your patience. Our responses to your comments are detailed below, point by point and in the order presented. Line and page numbers used to locate all the revisions has been made according to the original manuscript uploaded.
Review Report Form 1
The review describes the recent progress on the role of Rho/ROCK signaling pathway as possible target for the treatment of diabetic macular edema.
The article is a useful brief summary of the latest results in this context. The manuscript is well written. The citations are adequate. However, some basic processes are superficially explained.
- Considering that the section 3 is related to the pathophysiology of diabetic macular oedema, the title of some subsections appears inadequate. In particular, change the title of section 3.2. Blood-retinal barrier in 3.2. Blood-retinal barrier breakdown. Moreover, the authors have to better describe the event underlying the blood-retinal barrier breakdown.
- We agree with your opinion, and we believe we should update section 3. The subsection title was modified as suggested (page 4, line 113), and section 3.2., about blood-retinal barrier breakdown, was described in greater depth. When such modifications are made, section 3.2 read as follows (changes regarding the main manuscript have been highlighted with green colour):
3.2. Blood-retinal barrier breakdown
“Neuroretinal tissue is highly vulnerable, and therefore, the balance between molecules and fluid entry and exit is highly regulated. The principal regulatory mechanism for fluid access in the retina is BRBs, both inner and outer [1]. BRB breakdown is a complex process that is regulated by multiple factors and involves different mechanisms, and when it occurs at inner and/or outer BRB, excess fluid can accumulate, and this can result in macular oedema. Regarding fluid exit, the drainage system depends on a complex system of transport of ion/water channels located in the polarized retinal pigment epithelium (RPE) cells and the retinal Müller glial cells [1,2].
The inner BRB is composed of endothelial cells of retinal vessels sealed by tight junctions, where pericytes also seem to play an important role and the astrocytes create a complex structure formed by the interaction of different scaffolding and transmembrane and signalling proteins [1,3,4].
In contrast, the outer BRB comprises the intercellular junction complex of the RPE, its basal membrane and, partially, Bruch’s membrane, which creates a barrier between the choroidal vasculature and the neuroretina. Choriocapillaris vessels nourish photoreceptors and external retinal layers, and unlike retinal vessels, are highly fenestrated, permeable and lack pericytes [5]. Of note, recent studies have also demonstrated the barrier properties of the outer limiting membrane [6].
As mentioned, the inner BRB is highly regulated by a complex neuro-glio-vascular system made by endothelial cells and their basal lamina, surrounded by pericytes, astrocytes, Müller cells and microglia, which interact with them, and other retinal components selectively controlling the molecular transport across this barrier [1,7,8].
BRB breakdown can result from a disruption of the tight junctions, by upregulation of substance transport across RPE cells or retinal vascular endothelial cells, by degenerative changes to the barrier-forming cells or to the regulatory cells (pericytes and glia cells) [9]. BRB breakdown can be related to localized structural defects, such as microaneurysm, or to a diffuse retinal vascular leakage in which diffusible factors can be involved.
For example, an overproduction by retinal cells and an imbalance between VEGF and platelet-derived growth factor (PDGF), is partially responsible for altering BRBs integrity and permeability and for vasogenic oedema and cellular damage, including endothelial and Müller cells, astrocytes and pericytes loss [3,4,10–12]. In DR, a chronic hyperglycemic environment ends up provoking an increase of inflammatory and proangiogenic factors and a metabolic disruption in both barriers [7].”
- Please change the title of section 3.3. Retinal pigment epithelium in Retinal pigment epithelium damage.
- We have made the suggested change on page 5, line 136, from the original manuscript since that title appears more accurate. It would show as follows:
“3.3. Retinal pigment epithelium damage”.
- In the section 3., the authors should add a paragraph regarding the hypoxic event, considering that besides inflammation, hypoxia is a key player in the progression of this pathology.
- We thank the reviewer for pointing this out, we have reviewed pathophysiology section and we believe that it is necessary to complete it with the hypoxic event. To solve it we have created a new subsection just after point 3.3. (Retinal pigment epithelium damage), in line 135 page 5:
“3.4. Hypoxia
Capillary abnormalities and obstruction in DR induce retinal hypoxia, which is also one of the hinges in DME pathogenesis [1,13]. In hypoxic conditions, hypoxia-inducible factor 1α (HIF-1α) promote the expression of angiogenic factors, including VEGF. Therefore, HIF production promotes retinal vessels formation in an attempt to improve retinal oxygenation but contributing to pro and antiangiogenic factors disbalance and its deleterious consequences for retinal cells and BRBs [1,7,10,12]. VEGF and HIF-1α have shown an intravitreal concentration up to 10 times higher in diabetic patients than in nondiabetic ones [7,13].”
- The authors should better deepen the paragraph regarding the use of ROCK inhibitors in the treatment of DMO.
- Regarding this suggestion, we have decided to completely update section 5 and 6, regarding ROCK pathways and ROCK-inhibitors implementation adding a few paragraphs, dividing section 5 into subsections, modifying figure 2 and changing the order of the tables. In addition, Figure 3, a summary diagram of point 5.3. (Involvement of Rho/ROCK pathway in the pathogenesis of DR) was attached.
- Such modifications would be made on pages 8-13, lines 280-376 for section 5, and pages 13-15, lines 385-428 for section 6 of the text of the original main manuscript. Table 1 was reorganised, changing the order of the rows, and in table 2 the quantitative results of the studies included were added. Table 2 should appear in section 5, just after table 1 (page 11) and the new figure, the summary diagram, at the beginning of section 5 (“(…)” symbol was used to shorten the non-modified text from the original manuscript):
“5. Rho kinase and ROCK pathways
The Rho subfamily belongs to a member of the small molecule G protein in the Ras superfamily and has GTPase activity. It acts as a molecular switch (from inactive state (bound to GDP) to activated state (bound to GTP), exerting various biological effects by binding to its downstream target effector molecule (Figure 2). Among them, Rho-associated kinases (ROCK) are the most important effector molecules of the serine-threonine family that participate in the downstream signaling of Rho GTP-binding proteins [14–16]. There are two homologous isomers in the cell: ROCK1 (ROCK-I, ROKβ) and ROCK2 (ROCK-II, ROKα). Both are involved in the regulation of cell morphology, polarity, and cytoskeletal remodeling by regulating actin and cell migration, e.g., ROCKs phosphorylate myosin light chain (MLC) and LIM kinases to change actin cytoskeleton [17–19].
Figure 2. Rho activation and participation in DMO pathways. (A) In response to hyperglycaemia and in the presence of cytokines (TNF-α) and growing factors, such as VEGF and TGF-β, the activation of Rho by guanine exchange factors (GEF) is produced. GTPase-activating proteins (GAP) stimulate endogenous GTPase activity, facilitating GTP hydrolysis in GDP and inactivating Rho. (B) On the left side a normal-structured external retina is represented and on the right side an anomalous neurovascular junction in DMO. Rho-kinase signalling pathways are upregulated in Müller glial cells in response to hypoxia and oxidative stress and take part in DMO pathogenesis. (C) Representation of vessels from a normal retina (C,1) and a diabetic one (C,2). within inner-retinal vessels, rho pathways are involved in retinal vasoconstriction, endoluminal blebbing, endothelial impairment and leukostasis in response to an increased inflammation state and ICAM-1 and VCAM-1 hyperproduction in DMO. (D) At RPE level, ROCK participates in bleb formation and cellular remodelation and produces polarization and cell adhesion changes in response to hyperglycaemia modifying outer-BRB. In normal RPE cells ROCK-1 is cytoplasmic (D,1) but in diabetic conditions is recruited at the membrane (D,2). (E) Representation of an OCT image showing a cystoid DMO with hyperreflective dots as a result of the mentioned patho-physiological abnormalities. Rho kinase and ROCK pathways.
The ROCK molecular structure consists of three parts, the N-terminal kinase region, that catalyzes the phosphorylation/dephosphorylation of a series of downstream substrates; a coiled-coil region, in which the Rho-binding domain (RBD) regulates the activation signal of Rho transduction; and a C-terminal region terminal, which contains a pleckstrin homology (PH) and cysteine rich domain (CRD). This region contains a self-inhibiting region which interacts with the kinase domain to inhibit ROCK activity [20–23].
The ROCK1 and ROCK2 genes are located in the 18q11 and 2p24 chromosome regions. In addition, ROCK2 has a splicing mutant (mROCK2), called the small ROCK, which appears to be a product of partial gene duplication.
ROCK2 and mROCK2 are mainly localized to the cytoplasm, but ROCK2 is also localized to the plasma membrane through the C-terminal region. ROCK2 is highly expressed in brain and heart. However, ROCK1 can be concentrated in the plasma membrane of endothelium or in the center of the microtubule tissue of the moving cells, indicating its involvement in cell migration. ROCK1, is highly expressed in lung, liver, spleen, and kidney [20–23]. Both ROCK isomers are also present in ocular tissues and aberrant regulation of ROCK levels plays a role in the pathogenesis of corneal wound healing, glaucoma, diabetic retinopathy, and AMD [24–26].
5.1. Rock signalin pathway
ROCKs can be activated by the GTP-bound form of Rho and the activated form then phosphorylates downstream targets [24]. Binding of Rho-GTP opens the loop formation of the enzyme, and the activated form then phosphorylates downstream targets (Figure 2). Then, the activated ROCK can directly phosphorylate MLC, LIM Kinases, intermediate filament proteins, among other substrates, and in most cases, substrates are phosphorylated by their respective ROCK proteins. Taken together, Rho pathway activation leads to a concerted series of events resulting in increased actin-myosin contractility and cytoskeletal change. The C-terminus of the ROCK protein is a self-inhibiting region of kinase activity, including the RBD region and the PH region. In the inactive state, the RBD and PH regions of the C-terminus of the ROCK protein interact with the kinase domain to form a self-inhibiting loop (Figure 2) [20–24].
5.2. Relationship with diabetic retinopathy
Recently, RhoA and ROCK has been implicated in the pathogenesis of DR. The long-term glucose homeostasis imbalance, which leads to an accumulation of advanced glycation end products in the vessels. Under these inflammatory and hyperglycaemic diabetic conditions, guanine exchange factors (GEF) activate Rho, and subsequently activating ROCKs [19]. This abnormal expression of ROCK protein plays an important role in DMO pathogenesis, acting on retinal microvessels, leading to structural changes including vascular rigidity promoting leucocyte adhesion to the microvasculature by intercellular adhesion molecule-1 (ICAM-1) and contributing to VEGF-elicited microvascular hyperpermeability [27] (Figure 2). In addition, the activation of the Rho/ROCK signaling pathway regulates the NF-κB signaling pathway, which upregulates inflammatory genes, fibronectin and transcription factor AP-1 activation and accumulation of glomerular matrix proteins. Upregulation of Rho/ROCK pathway also results in the dephosphorylation of endothelial nitric oxide synthase (eNOS), which induces endothelial cell apoptosis and vasoconstriction [24–26,28].
Considering that increased activity of the Rho/ROCK pathway in diabetic patients contributes to exacerbates retinal vessel permeability, macular oedema, vascular occlusion and retinal ischaemia, several in vitro and in vivo studies have been conducted to assess the effect of Rho/ROCK pathway inhibition on diabetic retinopathy and DMO (Table 1 and Table 2).
Table 1. ROCK signalling involvement in diabetic retinopathy pathway.
|
Author, year |
Type of Study |
Model |
Outcomes |
|
|
Hyperglycaemia |
||||
|
(Lu et al. 2014) |
In vitro |
Choroid-retinal endothelial cell line |
High level of glucose increased RhoA activity |
|
|
(Mohammad et al. 2018) |
In vitro |
Human retinal Müller glial cells |
High glucose-induced upregulation of ROCK-1 in human retinal Müller glial cells |
|
|
(Yao et al. 2018) |
In vitro |
High-glucose-induced human retinal endothelial cells |
Rho/ROCK signalling pathway is involved in the hyperglycaemia-induced microvascular endothelial dysfunction |
|
|
Inflammation and vascularisation |
||||
|
(Mohammad et al. 2018) |
In vitro |
Human retinal Müller glial cells |
High glucose-induced upregulation of p-ERK ½, p-NF-κB, iNOS, VEGF and MCP-1/ CCL2 |
|
|
(Yamaguchi et al. 2016) |
In vitro |
Human retinal microvascular endothelial cells |
Increase in ROCK activity by VEGF stimulation |
|
|
(Bryan et al. 2010) |
Ex vivo/In vitro |
Ex vivo retinal explants Bovine retinal endothelial cells |
Rho/ROCK pathway mediates different VEGF-mediated angiogenesis processes |
|
|
RPE |
||||
|
(Rothschild et al. 2017)
|
In vivo
|
GK rat model |
Diabetic conditions modified the subcellular distribution of ROCK-1, which was located along the membrane, while ROCK-2 localisation was undisturbed |
|
|
Human retina tissue of diabetic patients |
Diabetic conditions modified the subcellular distribution of ROCK-1, which was located along the membrane, while ROCK-2 localisation was undisturbed |
|||
|
Vasoconstriction |
||||
|
(Rothschild et al. 2017) |
In vivo |
GK rat model |
Increased expression of ROCK-1 in mural cells and in the vessel wall of constricted areas |
|
|
(Chen et al. 2018) |
|
Porcine models |
ROCK signalling is involved in the venular constriction due to ET-1 |
|
|
(Rosa 2009) |
In vitro |
Human retina |
ET-1 provokes vasoconstriction of retinal arterioles via ROCK-signalling pathway |
|
|
(Nagaoka et al., n.d.) |
In vitro |
Porcine retinal arterioles |
Rho/ROCK pathway takes part in the simvastatin-induced NO-mediated dilatation of retinal arterioles |
|
|
Endothelial impairment |
||||
|
(Rothschild et al. 2017) |
In vivo
|
GK rat model |
ROCK-1 expression in endothelial cell blebs. |
|
|
Human retina tissue of diabetic patients |
||||
|
(Lu et al. 2014) |
|
|
Under hyperglycemic conditions, ROCK-1 contributes to endothelial dysfunction, through the damage of tight junctions |
|
|
(Arita et al. 2009) |
In vitro |
Diabetic rat model |
Increased levels of activated Rho in diabetic rat retinas |
|
|
Leukostasis |
||||
|
(Arita et al. 2009)
|
|
Diabetic rat model
|
CD11b and CD18 fluorescence in neutrophils of diabetic rats were higher, while CD11a fluorescence intensity did not differ |
|
|
ICAM expression was significantly higher in diabetic retinas |
||||
GK—Goto-Kakizaki, ET-1—Endothelin-1; CD—cluster of differentiation; MCP-1—Monocyte Chemoattractant Protein-1
Table 2. ROCK inhibitors as a therapeutical approachin patients with DMO. Summary of clinical studies.
|
Author, year |
Type of study |
No. of patients (eyes) |
Inclusion criteria |
Therapy evaluated |
Parameters assessed |
Outcomes |
|
|
Case |
Control |
||||||
|
(Ahmadieh et al. 2013) |
Prospective pilot study |
5 (5) |
- |
Persistent DMO, previously treated with: · Macular laser photocoagulation · Intravitreal bevacizumab injections |
Intravitreal fasudil injection (0.025mg/0.05mL) combined with Intravitreal bevacizumab injection (1.25mg/0,05mL) |
BCVA CMT |
· No toxic retinal effect was found · BCVA improved from 0.82 to 0.34 logMAR (p = 0.04) and CMT improved from 409 to 314µm ( p = 0.04) avg. at week 6 |
|
(Nourinia et al. 2013) |
Prospective pilot study |
15 (15) |
- |
Refractory DMO, previously treated with: · Intravitreal bevacizumab injections |
Intravitreal fasudil injection (0.025mg/0.05mL) combined with Intravitreal bevacizumab injection (1.25mg/0,05mL) |
BCVA CMT |
· BCVA improved from 0.84 to 0.49 log-MAR (p = 0.003) and CMT improved from 448 to 347 µm (p = 0.001) avg. at week 4 |
|
(Ahmadieh et al. 2019) |
Prospective randomised clinical trial |
22 (22) |
22 (22) |
Severe DMO |
Intravitreal fasudil injection (50 µM/L) combined with Intravitreal bevacizumab injection (1.25mg/0,05mL) |
BCVA CMT |
· BCVA changed in 12 ETDRS letters more at month 6 in combined group vs. bevacizumab group (p<0.001). CMT was also significantly more pronounced at month 6 in combined group (p<0.001) |
|
(Minami et al. 2019) |
Retrospective study |
10 (12) |
10 (14) |
Patients under ripasudil treatment for PAG with DMO |
Ripasudil eyedrops |
BCVA FT IOP |
· Decreased IOP (2.7 mmHg less; p = 0.008) and FT(44 µm less; p = 0.01) after one month of treatment · No significant changes were found in the BCVA |
BCVA— Best Corrected Visual Acuity; CMT— Central Macular Thickness; PAG— Primary Angle Glaucoma; IOP— Intraocular Pressure; FT— Foveal Thickness; DMO— Diabetic Macular Oedema; avg. — on average
5.3. Involvement of Rho/ROCK pathway in the pathogenesis of DR
As mentioned before, the pathogenesis of DMO is multifactorial and complex involving several mechanisms in which Rho/ROCK pathway is actively engaged. Recent studies have revealed that Rho and its target protein ROCK are implicated in important pathological pathways in DR and DMO such as hyperglycaemia, RPE disturbances, vasoconstriction, endothelial impairment, leukostasis and inflammation, and vascularization (Table 1 and Figure 3) [24–26,28,29].
Figure 3. ROCKs involvement in the mechanisms underlying DMO according to the results of preclinical studies. Together with inflammation, elevated glucose, which is a thorn in DR, constitutes a strong stimulus for Rho pathways activation and ROCK-1 upregulation. Moreover, VEGF overproduction because of DR, also increases ROCK activity, particularly those pathways in charge of VEGF-mediated angiogenic processes, getting caught in a vicious circle. At RPE level, ROCK-1 cytoplasmic proteins translocate to cell membrane and participate in actin cytoskeleton and tight cell junctions remodelation. At a vascular level, the calibre of the retinal vessels is also regulated by ROCKs related pathways. In addition, they contribute to endothelial impairment by ROCK-1 blebs formation and tight junction modifications, as happened at RPE. Finally, these kinases stimulate ICAM expression, which facilitate leukocyte adhesion and contributes to leukostasis. Thus, ROCKs take part in leukostasis, inflammation and vascularisation processes and participate in endothelial and RPE impairment, perpetuating DMO pathogenesis.
5.3.1. Hyperglycaemia
Hyperglycaemia is one of the most important stimuli for ROCK-1 activation, and the correlation between diabetes and microvascular endothelial cell dysfunction has been demonstrated in various studies before. e.g., on choroid-retinal endothelial cell lines, it has been shown by a meaningful increase in phospho-myosin phosphatase target protein-1 (p-MYPT-1) as a result of this overactivity in response to elevated glucose [30]. Rho activity is also involved in the pathogenesis of renal and aortic complications during diabetic states [31,32]. In addition, hyperglycaemia has been shown to increase the expression of monocyte chemoattractant protein-1 (MCP-1/CCL2) and VEGF by vascular endothelial cells and in the vitreous samples from patients with DR [33,34]. It has been reported that the inhibition of RhoA/ROCK1 may attenuate the hypertonicity of endothelial cell caused by high glucose microenvironment, partly block inflammation and prevented considerably the apoptosis of endothelial cells aroused by high glucose [25,26,29,30].
5.3.2. Inflammation and vascularization
The levels of the proinflammatory mediators are increased in the ocular microenvironment of patients with DR, suggesting that persistent inflammation is critical for DR initiation and progression.
Several studies have documented that diabetes enhances the production of inflammatory mediators such as p-ERK ½, p-NF-κB, iNOS, VEGF and MCP-1/ CCL2), and how Rho-kinases play a crucial role in the inflammatory signalling, proliferation, fibrosis, and apoptosis through mitogen-activated protein kinase (MAPK) p38MAP kinase and NF-κB activation. Activation of NF-κB induces the upregulation of leukocyte adhesion molecules and the production of proinflammatory cytokines and angiogenic factors [33].
As mentioned, inflammatory factors play a key role in DMO pathogenesis, and their own signalling pathways and expression are influenced by Rho kinases. In vitro assays has also been conducted to prove this relationship using human retinal Müller glial cells and microvascular endothelial cells, ex vivo retinal explants, bovine retinal endothelial cells and rat retinal cells [35–39]. Results of recent research indicate that ROCK-1 activation induces focal vascular constrictions, endoluminal blebbing, retinal hypoxia, and remodelling of RPE cells contributing to outer barrier breakdown [31,40]. The blebbing-induced closure reversed by ROCK inhibitors, could open a window for intervention in case of macular ischemia.
VEGF stimulates ROCK activity and rho/ROCK pathways, particularly those mediating VEGF-dependent angiogenic processes on proliferative and non-proliferative DR, i.e. Rho/ROCK pathways regulate VEGF expression induced by platelet-derived growth factor (PDGF)-BB in retinal vessel pericytes [36,37,39]. Rho kinase signaling is also involved in the injury of inner retinal cells such as Müller cells in the case of hypoxia and oxidative stress [38].
5.3.3. RPE disturbances
Diabetic hyperglycaemia, TNF-α, and TGF-β activate and increase the ROCK pathway at endothelial and RPE cells and hyalocytes [30,41]. ROCK-1 proteins are usually cytoplasmic proteins in rat and human RPEs and endothelial cells, but in diabetic induced models, it has been demonstrated to translocate to the membrane accompanying ROCK-1 positive cell membrane bleb formations…
(…)
5.3.4. Vasoconstriction
Vasoconstriction is an important mechanism for retinal regulation and for proper functioning, which is damaged in DMO. Rat, porcine and human retinal models and in vitro and in vivo assays have shown Rho participation in this damage by multiple mechanisms. ROCK signalling is involved in retinal arteriolar and venular constriction due to vascular endothelin-1 (ET-1), a potent vasoconstrictor secreted by vascular endothelial cells [42,43]. In fact, expression of ROCK-1 was increased in mural cells, and there exist ROCK-1 positive blebs on the vessel wall of constricted areas from diabetic rats, narrowing retinal vessels lumen [40]. It has been shown promising results in hypoxia-induced retinal neovascularization animal models just using topical ROCK inhibitors. This study showed that the topical ROCK inhibitor inhibited neovascularisation and could enhance normal retinal vascularisation, opening a new horizon for treatment of DR without affecting normal retinal vasculature or reducing the neuroretinal damage [44].
(…)
5.3.5. Endothelial impairmet
ROCK-1 translocation and blebbing and an increased level of activated Rho in endothelial cells were also found in diabetic rat retinas producing endothelial impairment and the DMO…
(…)
5.3.6. Leukostasis
CD11b/CD18 is a beta-2 integrin, a leukocyte-specific transmembrane protein participating in neutrophils adhesion, communication and migration…
(…)
Intravitreal fasudil, ripasudil (K-115) and AMA0428, ROCK inhibitors suppressed and reduced leukocyte migration, inhibited ICAM-1 expression in endothelial cells and decreased vascular leakage and neurodegeneration of diabetic retinal vessels, thereby improving retinal thickness and DMO [41,45–47].
The eNOS enzyme and its phosphorylation appear to be downregulated in diabetic retinal endothelial cells, and fasudil treatment has also proven to have a positive effect in reversing the decreased expression of this enzyme, but the production of nitric oxide (NO) is required to be maintained for fasudil to work [20–25,28].
- From the bench to the bedside
Anti-VEGF therapy in DMO treatment has changed the management of this condition, improving the control of the disease. However, anatomical and visual results are not always as expected, and some cases remain refractory to this treatment. Therefore, the identification of adjuvant or alternative therapies is required, the modulation of Rho/ROCK pathway is a potential candidate, since the suppression of leukocyte adhesion and neutrophil-induced retinal endothelial cell damage has been shown with its inhibition, which could have beneficial effects in terms of anatomical and visual outcomes. A summary of studies in which ROCK inhibitors are used as a treatment was provided in Table 2.
Since its discovered in the second half of the 90s, the modulation of rho kinase has been considered a potential therapeutic target to explore in a broad range of human disease, being cardiovascular disease one of the main areas in which they have been studied; unfortunately, its inhibition also carries the risk of unacceptable systemic side effects. In this framework, the eye is a possible application area due to the limited number of side effects, being the conjunctival hyperemia the main one.
Of note, the protein kinase family inhibitors have been historically classified according to their mechanism of action based on the binding modes, leading to a type I−V classification [48]. Although, a thorough analysis of the different mechanism of actions goes beyond the scope of this review, is essential to take into consideration that the majority of kinase inhibitors are based on type I mode, in which the inhibitor targets the ATP binding site of the kinase in its active conformation [48–50]. Another practical mode of classification is how they impact the kinase parameters, their difference substrate, and their competititive or non-competetive profile [49]; being a matter of controversy, what is the best classification method.
Regarding the eye, vast majority of rho kinase inhibitors act on both, ROCK-1 and ROCK-2. Fasudil (HA-1077, Fasudil, Asahi Kasei Pharma Corporation, Tokyo, Japan), an isoquinoline derivate and a calcium antagonist, is approved for the treatment of cerebral vasospasms in Japan and China; its fluorinated analogue, with much more power and selectivity. Ripasudil (K-115, Glanatec® Ophthalmic Solution 0.4%, Kowa Company, Ltd., Japan) and netarsudil (AR-13324, Rhopressa® Ophthalmic Solution 0.02%, Aerie Pharmaceuticals, Inc., Irvine, California), an amino-isoquinoline amide, act on both kinases and are the best-known ones. The three of them have been approved for glaucoma treatment because of their ability to reduce intraocular pressure lowering aqueous outflow resistance [19,51,52]. Only Belumosudil (KD-025) is a selective inhibitor of ROCK-2 (phase 2 clinical trials) being considered as a potential treatment for patients with some systemic inflammatory conditions [53]. AMA-0076, GSK-269962A, H-1152, Y-27632, RKI-1447, SAR-407899, SB-772077-B, Wf-536 andY-39983, are also double-acting ROCK inhibitor molecules [19,51].
It is crucial to take into consideration the mode of administration as most of the glaucoma preclinical and clinical studies are based on topical applied ophthalmic solutions, having been demonstrated their safety profile and hypotensive properties but with a high incidence of conjunctival hyperemia and with limited intraocular bioavailability due to the short corneal residence time and their characteristic hydrophilic properties [52,54–57]. This is one reason under the high rate of studies based on intravitreal injections for pathologies involving the posterior segment of the eye, trying to increase the intraocular active-drug concentration [58,59].
(…)
Of note, they were aware of the limitations of the study, as it was a retrospective study with a small number of patients with a short follow-up period and the drug was administered topically, with implies a limited intraocular bioavailability. However, it should be considered for the development of further prospective clinical trials.
As previously explained, netarsudil, appears to be effective in lowering IOP, but with low intraocular bioavailability when administered topically [52]. Interestingly, netarsudil has particular physiochemical and pharmacological properties, as its more lipophilic profile, which makes it a suitable candidate for intravitreal administration. AR-13503, a prodrug of netarsudil with ROCK/PKC inhibition properties, is being investigated for its administration intravitreally in a biodegradable implant [60]. In vivo studies showed a safety profile and an improved bioavailability, being able to deliver adequate levels of AR-13503 to the posterior segment for a 5-6 month-time period [61].
Moreover, alone or in combination with anti-VEGF drugs, AR-13503 has shown an additive efficacy on antiangiogenic properties and RPE barrier functions protecting power on DR in vitro models[60,61]. These characteristics make AR-13503 implant a possible game-changing drug for the treatment of DME. The prodrug is ongoing a phase 1 safety testing clinical trial in DME and Neovascular Age-Related Macular Degeneration patients, which has not finished yet (NCT03835884).
Since the studies in this review are focused on DMO, questions arise as to whether ROCK inhibitors could be of benefit in other types of macular oedema, either as unique therapy or in combination with other therapies. It is interesting to note that the potential use of the intravitreal injection of bevacizumab and fasudil in refractory cases of macular oedema secondary to retinal vein occlusion is under investigation, although the results have not been published yet (NCT: 03391219).”
References
- Daruich, A.; Matet, A.; Moulin, A.; Kowalczuk, L.; Nicolas, M.; Sellam, A.; Rothschild, P.R.; Omri, S.; Gélizé, E.; Jonet, L.; et al. Mechanisms of Macular Edema: Beyond the Surface; Elsevier Ltd, 2018; Vol. 63; ISBN 3314427816.
- Araújo, R.S.; Santos, D.F.; Silva, G.A. The Role of the Retinal Pigment Epithelium and Müller Cells Secretome in Neovascular Retinal Pathologies. Biochimie 2018, 155, 104–108, doi:10.1016/j.biochi.2018.06.019.
- Lee, H.; Jeon, H.L.; Park, S.J.; Shin, J.Y. Effect of Statins, Metformin, Angiotensin-Converting Enzyme Inhibitors, and Angiotensin II Receptor Blockers on Age-Related Macular Degeneration. Yonsei Medical Journal 2019, 60, 679–686, doi:10.3349/ymj.2019.60.7.679.
- Bahrami, B.; Zhu, M.; Hong, T.; Chang, A. Diabetic Macular Oedema: Pathophysiology, Management Challenges and Treatment Resistance. Diabetologia 2016, 59, 1594–1608.
- Rizzolo, L.J. Development and Role of Tight Junctions in the Retinal Pigment Epithelium. International Review of Cytology 2007, 258, 195–234, doi:10.1016/S0074-7696(07)58004-6.
- Omri, S.; Omri, B.; Savoldelli, M.; Jonet, L.; Thillaye-Goldenberg, B.; Thuret, G.; Gain, P.; Jeanny, J.C.; Crisanti, P.; Behar-Cohen, F. The Outer Limiting Membrane (OLM) Revisited: Clinical Implications. Clinical Ophthalmology 2010, 4, 183–195, doi:10.2147/opth.s5901.
- Romero-Aroca, P.; Baget-Bernaldiz, M.; Pareja-Rios, A.; Lopez-Galvez, M.; Navarro-Gil, R.; Verges, R. Diabetic Macular Edema Pathophysiology: Vasogenic versus Inflammatory. Journal of Diabetes Research 2016, 2016.
- Sacconi, R.; Giuffrè, C.; Corbelli, E.; Borrelli, E.; Querques, G.; Bandello, F. Emerging Therapies in the Management of Macular Edema: A Review [Version 1; Peer Review: 2 Approved]. F1000Research 2019, 8.
- Vinores, S.A. Breakdown of the blood-retinal barrier. In Encyclopedia of the Eye; Elsevier, 2010; pp. 216–222 ISBN 9780123742032.
- Miller, J.W.; le Couter, J.; Strauss, E.C.; Ferrara, N. Vascular Endothelial Growth Factor a in Intraocular Vascular Disease. Ophthalmology 2013, 120, 106–114, doi:10.1016/j.ophtha.2012.07.038.
- Sodhi, A.; Ma, T.; Menon, D.; Deshpande, M.; Jee, K.; Dinabandhu, A.; Vancel, J.; Lu, D.; Montaner, S. Angiopoietin-like 4 Binds Neuropilins and Cooperates with VEGF to Induce Diabetic Macular Edema. Journal of Clinical Investigation 2019, 129, 4593–4608, doi:10.1172/JCI120879.
- Fogli, S.; Mogavero, S.; Egan, C.G.; del Re, M.; Danesi, R. Pathophysiology and Pharmacological Targets of VEGF in Diabetic Macular Edema. Pharmacological Research 2016, 103, 149–157, doi:10.1016/j.phrs.2015.11.003.
- Miller, K.; Fortun, J.A. Diabetic Macular Edema: Current Understanding, Pharmacologic Treatment Options, and Developing Therapies. Asia-Pacific Journal of Ophthalmology 2018, 7, 28–35, doi:10.22608/APO.2017529.
- Ruiz-Loredo, A.Y.; López, E.; López-Colomé, A.M. Thrombin Promotes Actin Stress Fiber Formation in RPE through Rho/ROCK-Mediated MLC Phosphorylation. Journal of Cellular Physiology 2011, 226, 414–423, doi:10.1002/jcp.22347.
- Jahani, V.; Kavousi, A.; Mehri, S.; Karimi, G. Rho Kinase, a Potential Target in the Treatment of Metabolic Syndrome. Biomedicine and Pharmacotherapy 2018, 106, 1024–1030.
- Moshirfar, M.; Parker, L.; Birdsong, O.C.; Ronquillo, Y.C.; Hofstedt, D.; Shah, T.J.; Gomez, A.T.; Hoopes, P.C.S. Use of Rho Kinase Inhibitors in Ophthalmology: A Review of the Literature. Medical hypothesis, discovery & innovation ophthalmology journal 2018, 7, 101–111.
- Rao, P.V.; Pattabiraman, P.P.; Kopczynski, C. Role of the Rho GTPase/Rho Kinase Signaling Pathway in Pathogenesis and Treatment of Glaucoma: Bench to Bedside Research. Experimental Eye Research 2017, 158, 23–32.
- Feng, Y.; Lograsso, P. V.; Defert, O.; Li, R. Rho Kinase (ROCK) Inhibitors and Their Therapeutic Potential. Journal of Medicinal Chemistry 2016, 59, 2269–2300.
- Koch, J.C.; Tatenhorst, L.; Roser, A.E.; Saal, K.A.; Tönges, L.; Lingor, P. ROCK Inhibition in Models of Neurodegeneration and Its Potential for Clinical Translation. Pharmacology and Therapeutics 2018, 189, 1–21, doi:10.1016/j.pharmthera.2018.03.008.
- Yamashita, F.; Fukuyama, E.; Mizoguchi, K.; Takizawa, S.; Xu, S.; Kawakata, H. Scale Dependence of Rock Friction at High Work Rate. Nature 2015, 528, 254–257, doi:10.1038/nature16138.
- Shimizu, Y.; Dobashi, K.; Sano, T.; Yamada, M. Rock Activation in Lung of Idiopathic Pulmonary Fibrosis with Oxidative Stress. International Journal of Immunopathology and Pharmacology 2014, 27, 37–44, doi:10.1177/039463201402700106.
- Zhang, C.; Zhang, S.; Zhang, Z.; He, J.; Xu, Y.; Liu, S. ROCK Has a Crucial Role in Regulating Prostate Tumor Growth through Interaction with C-Myc. Oncogene 2014, 33, 5582–5591, doi:10.1038/onc.2013.505.
- Alleaume-Butaux, A.; Nicot, S.; Pietri, M.; Baudry, A.; Dakowski, C.; Tixador, P.; Ardila-Osorio, H.; Haeberlé, A.M.; Bailly, Y.; Peyrin, J.M.; et al. Double-Edge Sword of Sustained ROCK Activation in Prion Diseases through Neuritogenesis Defects and Prion Accumulation. PLoS Pathogens 2015, 11, doi:10.1371/journal.ppat.1005073.
- Halász, É.; Townes-Anderson, E. Rock Inhibitors in Ocular Disease. ADMET and DMPK 2016, 4, 280–301.
- Abbhi, V.; Piplani, P. Rho-Kinase (ROCK) Inhibitors - A Neuroprotective Therapeutic Paradigm with a Focus on Ocular Utility. Current Medicinal Chemistry 2018, 27, 2222–2256, doi:10.2174/0929867325666181031102829.
- Fukiage, C.; Mizutani, K.; Kawamoto, Y.; Azuma, M.; Shearer, T.R. Involvement of Phosphorylation of Myosin Phosphatase by ROCK in Trabecular Meshwork and Ciliary Muscle Contraction. Biochemical and Biophysical Research Communications 2001, 288, 296–300, doi:10.1006/bbrc.2001.5751.
- Sun, H.; Breslin, J.W.; Zhu, J.; Yuan, S.Y.; Wu, M.H. Rho and ROCK Signaling in VEGF-Induced Microvascular Endothelial Hyperpermeability. Microcirculation 2006, 13, 237–247, doi:10.1080/10739680600556944.
- Nourinia, R.; Nakao, S.; Zandi, S.; Safi, S.; Hafezi-Moghadam, A.; Ahmadieh, H. ROCK Inhibitors for the Treatment of Ocular Diseases. British Journal of Ophthalmology 2018, 102.
- Arita, R.; Hata, Y.; Ishibashi, T. ROCK as a Therapeutic Target of Diabetic Retinopathy. Journal of Ophthalmology 2010, 2010, 1–9, doi:10.1155/2010/175163.
- Lu, Q.Y.; Chen, W.; Lu, L.; Zheng, Z.; Xu, X. Involvement of RhoA/ROCK1 Signaling Pathway in Hyperglycemia-Induced Microvascular Endothelial Dysfunction in Diabetic Retinopathy. International Journal of Clinical and Experimental Pathology 2014, 7, 7268–7277.
- Mohammad, G.; AlSharif, H.M.; Siddiquei, M.M.; Ahmad, A.; Alam, K.; El-Asrar, A.M.A. Rho-Associated Protein Kinase-1 Mediates the Regulation of Inflammatory Markers in Diabetic Retina and in Retinal Müller Cells. Annals of Clinical and Laboratory Science 2018, 48, 137–145.
- Yao, J.; Wang, J.; Yao, Y.; Wang, K.; Zhou, Q.; Tang, Y. MiR‑133b Regulates Proliferation and Apoptosis in High‑glucose‑induced Human Retinal Endothelial Cells by Targeting Ras Homolog Family Member A. International Journal of Molecular Medicine 2018, 42, 839–850, doi:10.3892/ijmm.2018.3694.
- Ozturk, B.T.; Bozkurt, B.; Kerimoglu, H.; Okka, M.; Kamis, U.; Gunduz, K. Effect of Serum Cytokines and VEGF Levels on Diabetic Retinopathy and Macular Thickness. Molecular Vision 2009, 15, 1906–1914.
- Nawaz, M.I.; van Raemdonck, K.; Mohammad, G.; Kangave, D.; van Damme, J.; Abu El-Asrar, A.M.; Struyf, S. Autocrine CCL2, CXCL4, CXCL9 and CXCL10 Signal in Retinal Endothelial Cells and Are Enhanced in Diabetic Retinopathy. Experimental Eye Research 2013, 109, 67–76, doi:10.1016/j.exer.2013.01.008.
- Mohammad, G.; AlSharif, H.M.; Siddiquei, M.M.; Ahmad, A.; Alam, K.; El-Asrar, A.M.A. Rho-Associated Protein Kinase-1 Mediates the Regulation of Inflammatory Markers in Diabetic Retina and in Retinal Müller Cells. Annals of Clinical and Laboratory Science 2018, 48, 137–145.
- Yamaguchi, M.; Nakao, S.; Arita, R.; Kaizu, Y.; Arima, M.; Zhou, Y.; Kita, T.; Yoshida, S.; Kimura, K.; Isobe, T.; et al. Vascular Normalization by ROCK Inhibitor: Therapeutic Potential of Ripasudil (K-115) Eye Drop in Retinal Angiogenesis and Hypoxia. Investigative Ophthalmology and Visual Science 2016, 57, 2264–2276, doi:10.1167/iovs.15-17411.
- Bryan, B.A.; Dennstedt, E.; Mitchell, D.C.; Walshe, T.E.; Noma, K.; Loureiro, R.; Saint‐Geniez, M.; Campaigniac, J.; Liao, J.K.; Patricia, D.A. RhoA/ROCK Signaling Is Essential for Multiple Aspects of VEGF‐mediated Angiogenesis. The FASEB Journal 2010, 24, 3186–3195, doi:10.1096/fj.09-145102.
- Zhang, X.H.; Feng, Z.H.; Wang, X.Y. The ROCK Pathway Inhibitor Y-27632 Mitigates Hypoxia and Oxidative Stress-Induced Injury to Retinal Muller Cells. Neural Regeneration Research 2018, 13, 549–555, doi:10.4103/1673-5374.228761.
- Yokota, T.; Utsunomiya, K.; Taniguchi, K.; Gojo, A.; Kurata, H.; Tajima, N. Involvement of the Rho/Rho Kinase Signaling Pathway in Platelet-Derived Growth Factor BB-Induced Vascular Endothelial Growth Factor Expression in Diabetic Rat Retina. Japanese Journal of Ophthalmology 2007, 51, 424–430, doi:10.1007/s10384-007-0471-0.
- Rothschild, P.R.; Salah, S.; Berdugo, M.; Gélizé, E.; Delaunay, K.; Naud, M.C.; Klein, C.; Moulin, A.; Savoldelli, M.; Bergin, C.; et al. ROCK-1 Mediates Diabetes-Induced Retinal Pigment Epithelial and Endothelial Cell Blebbing: Contribution to Diabetic Retinopathy. Scientific Reports 2017, 7, 1–15, doi:10.1038/s41598-017-07329-y.
- Arita, R.; Hata, Y.; Nakao, S.; Kita, T.; Miura, M.; Kawahara, S.; Zandi, S.; Almulki, L.; Tayyari, F.; Shimokawa, H.; et al. Rho Kinase Inhibition by Fasudil Ameliorates Diabetes-Induced Microvascular Damage. Diabetes 2009, 58, 215–226, doi:10.2337/db08-0762.
- Chen, Y.L.; Ren, Y.; Xu, W.; Rosa, R.H.; Kuo, L.; Hein, T.W. Constriction of Retinal Venules to Endothelin-1: Obligatory Roles of ET A Receptors, Extracellular Calcium Entry, and Rho Kinase. Investigative Ophthalmology and Visual Science 2018, 59, 5167–5175, doi:10.1167/iovs.18-25369.
- Rosa, R.H. Divergent Roles of Nitric Oxide and Rho Kinase in Vasomotor Regulation of Human Retinal Arterioles. 2009, doi:10.1167/iovs.09-4391.
- Yamaguchi, M.; Nakao, S.; Arita, R.; Kaizu, Y.; Arima, M.; Zhou, Y.; Kita, T.; Yoshida, S.; Kimura, K.; Isobe, T.; et al. Vascular Normalization by ROCK Inhibitor: Therapeutic Potential of Ripasudil (K-115) Eye Drop in Retinal Angiogenesis and Hypoxia. Investigative Ophthalmology and Visual Science 2016, 57, 2264–2276, doi:10.1167/iovs.15-17411.
- Barreiro, O.; Yáñez-Mó, M.; Serrador, J.M.; Montoya, M.C.; Vicente-Manzanares, M.; Tejedor, R.; Furthmayr, H.; Sánchez-Madrid, F. Dynamic Interaction of VCAM-1 and ICAM-1 with Moesin and Ezrin in a Novel Endothelial Docking Structure for Adherent Leukocytes. Journal of Cell Biology 2002, 157, 1233–1245, doi:10.1083/jcb.200112126.
- Nourinia, R.; Ahmadieh, H.; Shahheidari, M.H.; Zandi, S.; Nakao, S.; Hafezi-Moghadam, A. Intravitreal Fasudil Combined with Bevacizumab for Treatment of Refractory Diabetic Macular Edema; a Pilot Study. Journal of Ophthalmic and Vision Research 2013, 8, 337–340.
- Ahmadieh, H.; Nourinia, R.; Hafezi-Moghadam, A. Intravitreal Fasudil Combined with Bevacizumab for Persistent Diabetic Macular Edema: A Novel Treatment. JAMA Ophthalmology 2013, 131, 923–924, doi:10.1001/jamaophthalmol.2013.143.
- van Linden, O.P.J.; Kooistra, A.J.; Leurs, R.; de Esch, I.J.P.; de Graaf, C. KLIFS: A Knowledge-Based Structural Database to Navigate Kinase-Ligand Interaction Space. Journal of Medicinal Chemistry 2014, 57, 249–277.
- Feng, Y.; Lograsso, P. v.; Defert, O.; Li, R. Rho Kinase (ROCK) Inhibitors and Their Therapeutic Potential. Journal of Medicinal Chemistry 2016, 59, 2269–2300, doi:10.1021/acs.jmedchem.5b00683.
- Zhao, Z.; Wu, H.; Wang, L.; Liu, Y.; Knapp, S.; Liu, Q.; Gray, N.S. Exploration of Type II Binding Mode: A Privileged Approach for Kinase Inhibitor Focused Drug Discovery? ACS Chemical Biology 2014, 9, 1230–1241.
- Tanna, A.P.; Johnson, M. Rho Kinase Inhibitors as a Novel Treatment for Glaucoma and Ocular Hypertension. Ophthalmology 2018, 125, 1741–1756.
- Hoy, S.M. Netarsudil Ophthalmic Solution 0.02%: First Global Approval. Drugs 2018, 78, 389–396, doi:10.1007/s40265-018-0877-7.
- Jagasia, M.; Lazaryan, A.; Bachier, C.R.; Salhotra, A.; Weisdorf, D.J.; Zoghi, B.; Essell, J.; Green, L.; Schueller, O.; Patel, J.; et al. ROCK2 Inhibition With Belumosudil (KD025) for the Treatment of Chronic Graft-Versus-Host Disease. Journal of Clinical Oncology 2021, JCO.20.02754, doi:10.1200/jco.20.02754.
- Garnock-Jones, K.P. Ripasudil: First Global Approval. Drugs 2014, 74, 2211–2215, doi:10.1007/s40265-014-0333-2.
- Nakabayashi, S.; Kawai, M.; Yoshioka, T.; Song, Y.S.; Tani, T.; Yoshida, A.; Nagaoka, T. Effect of Intravitreal Rho Kinase Inhibitor Ripasudil (K-115) on Feline Retinal Microcirculation. Experimental Eye Research 2015, 139, 132–135, doi:10.1016/j.exer.2015.07.008.
- Pakravan, M.; Beni, A.N.; Ghahari, E.; Varshochian, R.; Yazdani, S.; Esfandiari, H.; Ahmadieh, H. The Ocular Hypotensive Efficacy of Topical Fasudil, a Rho-Associated Protein Kinase Inhibitor, in Patients with End-Stage Glaucoma. American Journal of Therapeutics 2016, 24, E676–E680, doi:10.1097/MJT.0000000000000362.
- Mietzner, R.; Breunig, M. Causative Glaucoma Treatment: Promising Targets and Delivery Systems. Drug Discovery Today 2019, 24, 1606–1613.
- Li, M.; Yasumura, D.; Ma, A.A.K.; Matthes, M.T.; Yang, H.; Nielson, G.; Huang, Y.; Szoka, F.C.; LaVail, M.M.; Diamond, M.I. Intravitreal Administration of HA-1077, a ROCK Inhibitor, Improves Retinal Function in a Mouse Model of Huntington Disease. PLoS ONE 2013, 8, doi:10.1371/journal.pone.0056026.
- Mietzner, R.; Kade, C.; Froemel, F.; Pauly, D.; Stamer, W.D.; Ohlmann, A.; Wegener, J.; Fuchshofer, R.; Breunig, M. Fasudil Loaded PLGA Microspheres as Potential Intravitreal Depot Formulation for Glaucoma Therapy. Pharmaceutics 2020, 12, 1–22, doi:10.3390/pharmaceutics12080706.
- Glendenning, A.; Crews, K.; Sturdivant, J.; A deLong, M.; Kopczynski, C.; Lin, C.-W. Sustained Release, Biodegradable PEA Implants for Intravitreal Delivery of the ROCK/PKC Inhibitor AR-13503 | IOVS | ARVO Journals. Invest. Ophthalmol. Vis. Sci. 2018, 59, 5672.
- Ding, J.; Crews, K.; Carbajal, K.; Weksler, M.; Moore, L.; Carlson, E.C.; Lin, C.-W. Ocular Tissue Distribution and Duration of Release of AR-13503 Following Administration of AR-13503 Sustained Release Intravitreal Implant in Rabbits and Miniature Swine | IOVS | ARVO Journals. Invest. Ophthalmol. Vis. Sci. 2019, 60, 5387.

Reviewer 2 Report
This is an interesting review article about the RHO kinase and refractory diabetic macular edema. As Rho/ROCK pathway is highlighted recently and it can be potential target of therapy, the review article is timely. Compared to the well summarized the diabetic retinopathy and DMO part, the part about Rho/ROCK and ROCK inhibitors is hard to read and appears not well summarized. I have several concers as follows- I suggest a Figure illustrating the mechanism of DMO and the position of the Rho/ROCK be presented for the readers' understanding.
- Line 206. It is not at all certain that anti-VEGF is associated with retinal or macular atrophy and the evidence is not sufficient.
- Figure 1 is hard to understand without the legend. There should be more informations on the figure itself.
- The abbreviations should be unscrambled at ther first use. There are many abbreviation that readers might not know the full terms or meaning. (i.e. GK, ROCK...)
- Authors should elaborate more about the ROCK inhibitors such as its history, mechanism of action, classifications, the clinical use, etc.
- Table 1 itself is hard to get the idea of ROCK signalling in DR pathway. A figure explaining the table 1 would be helpful for reader to understand it.
- Page 12 is hard to read. I recommend English editing especially for this page.
- Before showing table 2 about clinical studies, a summary table of preclinical trials should be shown regarding the efficacy and safety of ROCK inhibitors in animals.
- Table 2 is about the clinical studies on humans. The title of the table should be more specific rather than "treatments".
- Table 2. In all studies, the nature of the study such as prospective or retrospective should be described evn in pilot studies.
- Table 2. The outcomes should be described quantitatively with statsitical significance.
- Line 428 NTC -> NCT
- For the intravitreal drug to be effective for the chronic diseases such as DMO, the intraocular PK should be also considered. I hope the authors mention about it regarding ROCK inhibitors.
Author Response
To whom may concern,
First and foremost, thank you for taking the time to assess our manuscript. We addressed all the concerns you raised and appreciated your patience. Our responses to your comments are detailed below, point by point and in the order presented.
Review Report Form 2
- I suggest a Figure illustrating the mechanism of DMO and the position of the Rho/ROCK be presented for the readers' understanding.
- We agree with the reviewer that illustrating the position of the rho/ROCK would be helpful. In fact, Figure 2, was included for that purpose. We apologize if our original figure did not show it clearly. For this reason, we completed the figure adding schematic representations of ROCKs molecular structure, retinal pigmentary epithelium, retinal blood vessels and an OCT DMO image. We hope that it is clearer (Figure 2 and figure caption lines 292-302, page 9). Changes regarding the main manuscript have been highlighted with green colour:
“Figure 2. Rho activation and participation in DMO pathways. (A) In response to hyperglycaemia and in the presence of cytokines (TNF-α) and growing factors, such as VEGF and TGF-β, the activation of Rho by guanine exchange factors (GEF) is produced. GTPase-activating proteins (GAP) stimulate endogenous GTPase activity, facilitating GTP hydrolysis in GDP and inactivating Rho. (B) On the left side a normal-structured external retina is represented and on the right side an anomalous neurovascular junction in DMO. Rho-kinase signalling pathways are upregulated in Müller glial cells in response to hypoxia and oxidative stress and take part in DMO pathogenesis. (C) Representation of vessels from a normal retina (C,1) and a diabetic one (C,2). within inner-retinal vessels, rho pathways are involved in retinal vasoconstriction, endoluminal blebbing, endothelial impairment and leukostasis in response to an increased inflammation state and ICAM-1 and VCAM-1 hyperproduction in DMO. (D) At RPE level, ROCK participates in bleb formation and cellular remodelation and produces polarization and cell adhesion changes in response to hyperglycaemia modifying outer-BRB. In normal RPE cells ROCK-1 is cytoplasmic (D,1) but in diabetic conditions is recruited at the membrane (D,2). (E) Representation of an OCT image showing a cystoid DMO with hyperreflective dots as a result of the mentioned patho-physiological abnormalities. Rho kinase and ROCK pathways”.
- Line 206. It is not at all certain that anti-VEGF is associated with retinal or macular atrophy and the evidence is not sufficient.
- We agree with the observation; indeed, we tried to emphasize that it is not an established fact. We apologize for that misunderstanding, and the new sentence would read as follows (page 6 line 204-211):
“This approach implies repeated intravitreal administration of anti-angiogenic agents for an extended period, which means a well-known burden for patients and the possibility of significant side effects. In addition, some studies have suggested that sustained anti-VEGF suppression could have undesirable consequences, such as capillary dropout, retinal thinning, ganglion-cell damage and a certain predisposition to retinal atrophy [1,2]. Moreover, intravitreal injections could produce an angio-fibrotic switch because of the low VEGF levels resulting in tractional complications because of fibrosis stimulation and epiretinal membrane formation [3]”.
- Figure 1 is hard to understand without the legend. There should be more information on the figure itself.
- We appreciate the suggestion, but the text was removed from the images to make them easier to appreciate and not to cover the findings from the clinical case.
- The abbreviations should be unscrambled at their first use. There are many abbreviation that readers might not know the full terms or meaning. (i.e. GK, ROCK...)
- We have fixed the error carefully reviewing all the abbreviations used in the text, i.e. page 11, line 310 (table caption): “GK—Goto-Kakizaki, ET-1—Endothelin-1; CD—cluster of differentiation; MCP-1—Monocyte Chemoattractant Protein-1” and page 8, line 281: “Rho-associated kinases (ROCK)”.
- Authors should elaborate more about the ROCK inhibitors such as its history, mechanism of action, classifications, the clinical use, etc.
- We agree with that suggestion. For this purpose, we have added a couple of paragraphs extending section 6 (From the bench to the bedside) in page 13, line 377, which reads as follows (“(…)” symbol was used to shorten the non-modified text from the original manuscript):
“6. From the bench to the bedside
Anti-VEGF therapy in DMO treatment has changed the management of this condition, improving the control of the disease. However, anatomical and visual results are not always as expected, and some cases remain refractory to this treatment. Therefore, the identification of adjuvant or alternative therapies is required, the modulation of Rho/ROCK pathway is a potential candidate, since the suppression of leukocyte adhesion and neutrophil-induced retinal endothelial cell damage has been shown with its inhibition, which could have beneficial effects in terms of anatomical and visual outcomes. A summary of studies in which ROCK inhibitors are used as a treatment was provided in Table 2.
Since its discovered in the second half of the 90s, the modulation of rho kinase has been considered a potential therapeutic target to explore in a broad range of human disease, being cardiovascular disease one of the main areas in which they have been studied; unfortunately, its inhibition also carries the risk of unacceptable systemic side effects. In this framework, the eye is a possible application area due to the limited number of side effects, being the conjunctival hyperemia the main one.
Of note, the protein kinase family inhibitors have been historically classified according to their mechanism of action based on the binding modes, leading to a type I−V classification [4]. Although, a thorough analysis of the different mechanism of actions goes beyond the scope of this review, is essential to take into consideration that the majority of kinase inhibitors are based on type I mode, in which the inhibitor targets the ATP binding site of the kinase in its active conformation [4–6]. Another practical mode of classification is how they impact the kinase parameters, their difference substrate, and their competititive or non-competetive profile (XX); being a matter of controversy, what is the best classification method.
Regarding the eye, vast majority of rho kinase inhibitors act on both, ROCK-1 and ROCK-2. Fasudil (HA-1077, Fasudil, Asahi Kasei Pharma Corporation, Tokyo, Japan), an isoquinoline derivate and a calcium antagonist, is approved for the treatment of cerebral vasospasms in Japan and China; its fluorinated analogue, with much more power and selectivity. Ripasudil (K-115, Glanatec® Ophthalmic Solution 0.4%, Kowa Company, Ltd., Japan) and netarsudil (AR-13324, Rhopressa® Ophthalmic Solution 0.02%, Aerie Pharmaceuticals, Inc., Irvine, California), an amino-isoquinoline amide, act on both kinases and are the best-known ones. The three of them have been approved for glaucoma treatment because of their ability to reduce intraocular pressure lowering aqueous outflow resistance [7–9]. Only Belumosudil (KD-025) is a selective inhibitor of ROCK-2 (phase 2 clinical trials) being considered as a potential treatment for patients with some systemic inflammatory conditions [10]. AMA-0076, GSK-269962A, H-1152, Y-27632, RKI-1447, SAR-407899, SB-772077-B, Wf-536 andY-39983, are also double-acting ROCK inhibitor molecules [7,8].
It is crucial to take into consideration the mode of administration as most of the glaucoma preclinical and clinical studies are based on topical applied ophthalmic solutions, having been demonstrated their safety profile and hypotensive properties but with a high incidence of conjunctival hyperemia and with limited intraocular bioavailability due to the short corneal residence time and their characteristic hydrophilic properties [9,11–14]. This is one reason under the high rate of studies based on intravitreal injections for pathologies involving the posterior segment of the eye, trying to increase the intraocular active-drug concentration [15,16].”
- Moreover, that section was reviewed and redesigned with the aim of making it clearer and more precise and table 2 was relocated in a better position, just after Table 1 in section 5 page 11, line 310.
- Table 1 itself is hard to get the idea of ROCK signalling in DR pathway. A figure explaining the table 1 would be helpful for reader to understand it.
- We apologize if our original table 1 did not result easy to understand, we have modified the table and changed the order of the rows to make Table 1 easier to understand. Page 11, Table 1:
Table 1. ROCK signalling involvement in diabetic retinopathy pathway.
|
Author, year |
Type of Study |
Model |
Outcomes |
|
|
Hyperglycaemia |
||||
|
(Lu et al. 2014) |
In vitro |
Choroid-retinal endothelial cell line |
High level of glucose increased RhoA activity |
|
|
(Mohammad et al. 2018) |
In vitro |
Human retinal Müller glial cells |
High glucose-induced upregulation of ROCK-1 in human retinal Müller glial cells |
|
|
(Yao et al. 2018) |
In vitro |
High-glucose-induced human retinal endothelial cells |
Rho/ROCK signalling pathway is involved in the hyperglycaemia-induced microvascular endothelial dysfunction |
|
|
Inflammation and vascularisation |
||||
|
(Mohammad et al. 2018) |
In vitro |
Human retinal Müller glial cells |
High glucose-induced upregulation of p-ERK ½, p-NF-κB, iNOS, VEGF and MCP-1/ CCL2 |
|
|
(Yamaguchi et al. 2016) |
In vitro |
Human retinal microvascular endothelial cells |
Increase in ROCK activity by VEGF stimulation |
|
|
(Bryan et al. 2010) |
Ex vivo/In vitro |
Ex vivo retinal explants Bovine retinal endothelial cells |
Rho/ROCK pathway mediates different VEGF-mediated angiogenesis processes |
|
|
RPE |
||||
|
(Rothschild et al. 2017)
|
In vivo
|
GK rat model |
Diabetic conditions modified the subcellular distribution of ROCK-1, which was located along the membrane, while ROCK-2 localisation was undisturbed |
|
|
Human retina tissue of diabetic patients |
Diabetic conditions modified the subcellular distribution of ROCK-1, which was located along the membrane, while ROCK-2 localisation was undisturbed |
|||
|
Vasoconstriction |
||||
|
(Rothschild et al. 2017) |
In vivo |
GK rat model |
Increased expression of ROCK-1 in mural cells and in the vessel wall of constricted areas |
|
|
(Chen et al. 2018) |
|
Porcine models |
ROCK signalling is involved in the venular constriction due to ET-1 |
|
|
(Rosa 2009) |
In vitro |
Human retina |
ET-1 provokes vasoconstriction of retinal arterioles via ROCK-signalling pathway |
|
|
(Nagaoka et al., n.d.) |
In vitro |
Porcine retinal arterioles |
Rho/ROCK pathway takes part in the simvastatin-induced NO-mediated dilatation of retinal arterioles |
|
|
Endothelial impairment |
||||
|
(Rothschild et al. 2017) |
In vivo
|
GK rat model |
ROCK-1 expression in endothelial cell blebs. |
|
|
Human retina tissue of diabetic patients |
||||
|
(Lu et al. 2014) |
|
|
Under hyperglycemic conditions, ROCK-1 contributes to endothelial dysfunction, through the damage of tight junctions |
|
|
(Arita et al. 2009) |
In vitro |
Diabetic rat model |
Increased levels of activated Rho in diabetic rat retinas |
|
|
Leukostasis |
||||
|
(Arita et al. 2009)
|
|
Diabetic rat model
|
CD11b and CD18 fluorescence in neutrophils of diabetic rats were higher, while CD11a fluorescence intensity did not differ |
|
|
ICAM expression was significantly higher in diabetic retinas |
||||
GK—Goto-Kakizaki, ET-1—Endothelin-1; CD—cluster of differentiation; MCP-1—Monocyte Chemoattractant Protein-1
- In addition, we added the suggested figure and extended the information in section 5. We hope that it might help to clarify this section. Figure 3 was included within section 5 as follows:
“Figure 3. ROCKs involvement in the mechanisms underlying DMO according to the results of preclinical studies. Together with inflammation, elevated glucose, which is a thorn in DR, constitutes a strong stimulus for Rho pathways activation and ROCK-1 upregulation. Moreover, VEGF overproduction because of DR, also increases ROCK activity, particularly those pathways in charge of VEGF-mediated angiogenic processes, getting caught in a vicious circle. At RPE level, ROCK-1 cytoplasmic proteins translocate to cell membrane and participate in actin cytoskeleton and tight cell junctions remodelation. At a vascular level, the calibre of the retinal vessels is also regulated by ROCKs related pathways. In addition, they contribute to endothelial impairment by ROCK-1 blebs formation and tight junction modifications, as happened at RPE. Finally, these kinases stimulate ICAM expression, which facilitate leukocyte adhesion and contributes to leukostasis. Thus, ROCKs take part in leukostasis, inflammation and vascularisation processes and participate in endothelial and RPE impairment, perpetuating DMO pathogenesis.”
- Page 12 is hard to read. I recommend English editing especially for this page.
- Grammar and syntax were revised by your own English edition service. Here I am enclosing the bill we paid for doing that (English-editing-certificate). Nevertheless, the changes introduced throughout the paper to answer the reviewers’ corrections were made personally.
- Moreover, we have made a revision and redesign of section 5, and we hope this page will be easier to read now. In addition, some initial information was added at the beginning about ROCKs and their mechanism of action, figure 2 was modified, and the information on page 12 was divided into subsections to make it easier to read:
“5. Rho kinase and ROCK pathways
The Rho subfamily belongs to a member of the small molecule G protein in the Ras superfamily and has GTPase activity. It acts as a molecular switch (from inactive state (bound to GDP) to activated state (bound to GTP), exerting various biological effects by binding to its downstream target effector molecule (Figure 2). Among them, Rho-associated kinases (ROCK) are the most important effector molecules of the serine-threonine family that participate in the downstream signaling of Rho GTP-binding proteins [17–19]. There are two homologous isomers in the cell: ROCK1 (ROCK-I, ROKβ) and ROCK2 (ROCK-II, ROKα). Both are involved in the regulation of cell morphology, polarity, and cytoskeletal remodeling by regulating actin and cell migration, e.g., ROCKs phosphorylate myosin light chain (MLC) and LIM kinases to change actin cytoskeleton [7,20,21].
(Figure 2)
The ROCK molecular structure consists of three parts, the N-terminal kinase region, that catalyzes the phosphorylation/dephosphorylation of a series of downstream substrates; a coiled-coil region, in which the Rho-binding domain (RBD) regulates the activation signal of Rho transduction; and a C-terminal region terminal, which contains a pleckstrin homology (PH) and cysteine rich domain (CRD). This region contains a self-inhibiting region which interacts with the kinase domain to inhibit ROCK activity [22–25].
The ROCK1 and ROCK2 genes are located in the 18q11 and 2p24 chromosome regions. In addition, ROCK2 has a splicing mutant (mROCK2), called the small ROCK, which appears to be a product of partial gene duplication.
ROCK2 and mROCK2 are mainly localized to the cytoplasm, but ROCK2 is also localized to the plasma membrane through the C-terminal region. ROCK2 is highly expressed in brain and heart. However, ROCK1 can be concentrated in the plasma membrane of endothelium or in the center of the microtubule tissue of the moving cells, indicating its involvement in cell migration. ROCK1, is highly expressed in lung, liver, spleen, and kidney [22–25]. Both ROCK isomers are also present in ocular tissues and aberrant regulation of ROCK levels plays a role in the pathogenesis of corneal wound healing, glaucoma, diabetic retinopathy, and AMD [26–28].
5.1. Rock signalin pathway
ROCKs can be activated by the GTP-bound form of Rho and the activated form then phosphorylates downstream targets [26]. Binding of Rho-GTP opens the loop formation of the enzyme, and the activated form then phosphorylates downstream targets (Figure 2). Then, the activated ROCK can directly phosphorylate MLC, LIM Kinases, intermediate filament proteins, among other substrates, and in most cases, substrates are phosphorylated by their respective ROCK proteins. Taken together, Rho pathway activation leads to a concerted series of events resulting in increased actin-myosin contractility and cytoskeletal change. The C-terminus of the ROCK protein is a self-inhibiting region of kinase activity, including the RBD region and the PH region. In the inactive state, the RBD and PH regions of the C-terminus of the ROCK protein interact with the kinase domain to form a self-inhibiting loop (Figure 2) [22–26].
5.2. Relationship with diabetic retinopathy
Recently, RhoA and ROCK has been implicated in the pathogenesis of DR. The long-term glucose homeostasis imbalance, which leads to an accumulation of advanced glycation end products in the vessels. Under these inflammatory and hyperglycaemic diabetic conditions, guanine exchange factors (GEF) activate Rho, and subsequently activating ROCKs [7]. This abnormal expression of ROCK protein plays an important role in DMO pathogenesis, acting on retinal microvessels, leading to structural changes including vascular rigidity promoting leucocyte adhesion to the microvasculature by intercellular adhesion molecule-1 (ICAM-1) and contributing to VEGF-elicited microvascular hyperpermeability [29] (Figure 2). In addition, the activation of the Rho/ROCK signaling pathway regulates the NF-κB signaling pathway, which upregulates inflammatory genes, fibronectin and transcription factor AP-1 activation and accumulation of glomerular matrix proteins. Upregulation of Rho/ROCK pathway also results in the dephosphorylation of endothelial nitric oxide synthase (eNOS), which induces endothelial cell apoptosis and vasoconstriction [26–28,30].
Considering that increased activity of the Rho/ROCK pathway in diabetic patients contributes to exacerbates retinal vessel permeability, macular oedema, vascular occlusion and retinal ischaemia, several in vitro and in vivo studies have been conducted to assess the effect of Rho/ROCK pathway inhibition on diabetic retinopathy and DMO (Table 1 and Table 2).
(Table 1 and Table 2)
5.3. Involvement of Rho/ROCK pathway in the pathogenesis of DR
As mentioned before, the pathogenesis of DMO is multifactorial and complex involving several mechanisms in which Rho/ROCK pathway is actively engaged. Recent studies have revealed that Rho and its target protein ROCK are implicated in important pathological pathways in DR and DMO such as hyperglycaemia, RPE disturbances, vasoconstriction, endothelial impairment, leukostasis and inflammation, and vascularization (Table 1 and Figure 3) [26–28,30,31].
(Figure 3)
5.3.1. Hyperglycaemia
Hyperglycaemia is one of the most important stimuli for ROCK-1 activation, and the correlation between diabetes and microvascular endothelial cell dysfunction has been demonstrated in various studies before. e.g., on choroid-retinal endothelial cell lines, it has been shown by a meaningful increase in phospho-myosin phosphatase target protein-1 (p-MYPT-1) as a result of this overactivity in response to elevated glucose [32]. Rho activity is also involved in the pathogenesis of renal and aortic complications during diabetic states [33,34]. In addition, hyperglycaemia has been shown to increase the expression of monocyte chemoattractant protein-1 (MCP-1/CCL2) and VEGF by vascular endothelial cells and in the vitreous samples from patients with DR [35,36]. It has been reported that the inhibition of RhoA/ROCK1 may attenuate the hypertonicity of endothelial cell caused by high glucose microenvironment, partly block inflammation and prevented considerably the apoptosis of endothelial cells aroused by high glucose [27,28,31,32].
5.3.2. Inflammation and vascularization
The levels of the proinflammatory mediators are increased in the ocular microenvironment of patients with DR, suggesting that persistent inflammation is critical for DR initiation and progression.
Several studies have documented that diabetes enhances the production of inflammatory mediators such as p-ERK ½, p-NF-κB, iNOS, VEGF and MCP-1/ CCL2), and how Rho-kinases play a crucial role in the inflammatory signalling, proliferation, fibrosis, and apoptosis through mitogen-activated protein kinase (MAPK) p38MAP kinase and NF-κB activation. Activation of NF-κB induces the upregulation of leukocyte adhesion molecules and the production of proinflammatory cytokines and angiogenic factors [35].
As mentioned, inflammatory factors play a key role in DMO pathogenesis, and their own signalling pathways and expression are influenced by Rho kinases. In vitro assays has also been conducted to prove this relationship using human retinal Müller glial cells and microvascular endothelial cells, ex vivo retinal explants, bovine retinal endothelial cells and rat retinal cells [37–41]. Results of recent research indicate that ROCK-1 activation induces focal vascular constrictions, endoluminal blebbing, retinal hypoxia, and remodelling of RPE cells contributing to outer barrier breakdown [33,42]. The blebbing-induced closure reversed by ROCK inhibitors, could open a window for intervention in case of macular ischemia.
VEGF stimulates ROCK activity and rho/ROCK pathways, particularly those mediating VEGF-dependent angiogenic processes on proliferative and non-proliferative DR, i.e. Rho/ROCK pathways regulate VEGF expression induced by platelet-derived growth factor (PDGF)-BB in retinal vessel pericytes [38,39,41]. Rho kinase signaling is also involved in the injury of inner retinal cells such as Müller cells in the case of hypoxia and oxidative stress [40].
5.3.3. RPE disturbances
Diabetic hyperglycaemia, TNF-α, and TGF-β activate and increase the ROCK pathway at endothelial and RPE cells and hyalocytes [32,43]. ROCK-1 proteins are usually cytoplasmic proteins in rat and human RPEs and endothelial cells, but in diabetic induced models, it has been demonstrated to translocate to the membrane accompanying ROCK-1 positive cell membrane bleb formations. By contrast, ROCK-2 localization remained undisturbed with staining at the cytoplasmic membrane [42]. At the RPE level, this translocation implies a remodelling of the actin cytoskeleton and a modification and opening of tight cell junctions, so crucial for RPE functions and participating in the outer BRB breakdown [33,42,44]. In addition, relocation and inactivation of ROCK-1 and an improvement of RPE barrier function after fasudil (HA-1077) injection have been reported in vitro assays supporting Rho-pathway involvement in DMO pathogenesis [42].
5.3.4. Vasoconstriction
Vasoconstriction is an important mechanism for retinal regulation and for proper functioning, which is damaged in DMO. Rat, porcine and human retinal models and in vitro and in vivo assays have shown Rho participation in this damage by multiple mechanisms. ROCK signalling is involved in retinal arteriolar and venular constriction due to vascular endothelin-1 (ET-1), a potent vasoconstrictor secreted by vascular endothelial cells [45,46]. In fact, expression of ROCK-1 was increased in mural cells, and there exist ROCK-1 positive blebs on the vessel wall of constricted areas from diabetic rats, narrowing retinal vessels lumen [42]. It has been shown promising results in hypoxia-induced retinal neovascularization animal models just using topical ROCK inhibitors. This study showed that the topical ROCK inhibitor inhibited neovascularisation and could enhance normal retinal vascularisation, opening a new horizon for treatment of DR without affecting normal retinal vasculature or reducing the neuroretinal damage [47].
Its contribution also has been demonstrated in simvastatin-induced nitric-oxide-mediated dilatation of retinal arterioles [48]. The eNOS enzyme and its phosphorylation appear to be downregulated in diabetic retinal endothelial cells, and fasudil treatment has also proven to have a positive effect in reversing the decreased expression of this enzyme, but the production of nitric oxide (NO) is required to be maintained for fasudil to work [43].
5.3.5. Endothelial impairmet
ROCK-1 translocation and blebbing and an increased level of activated Rho in endothelial cells were also found in diabetic rat retinas producing endothelial impairment and the DMO [42,43]. Rat and human models have shown that under hyperglycaemic conditions and ROCK-1 activation, which occurred in the RPE, occludin, ZO-1 and claudin-5 protein levels are reduced, destabilising tight junctions in endothelial cells and causing inner-BRB disruption [32,49].
5.3.6. Leukostasis
CD11b/CD18 is a beta-2 integrin, a leukocyte-specific transmembrane protein participating in neutrophils adhesion, communication and migration, and they bind other adhesion molecules such as VCAM-1, ICAM-1 or certain complement components [50]. These molecules have shown to be upregulated in retinal vessels from rat diabetic models since an augmented expression of ICAM-1, an augmented fluorescence of beta-2 integrins and a higher number of neutrophils bound to harmed endothelial cells with respect to controls were found in in vitro assays [43,50]. Rho pathway activation stimulates ICAM overexpression and facilitates leukocyte adhesion and leukostasis by producing ICAM clustering and activating anchoring proteins at endothelial cells by phosphorylation [42,43]. The adhesion and stasis of leukocytes have been shown to produce diabetic microvascular damage and endothelial dysfunction contributing to DMO pathogenesis [51]. Intravitreal fasudil, ripasudil (K-115) and AMA0428, ROCK inhibitors suppressed and reduced leukocyte migration, inhibited ICAM-1 expression in endothelial cells and decreased vascular leakage and neurodegeneration of diabetic retinal vessels, thereby improving retinal thickness and DMO [43,52–54].
The eNOS enzyme and its phosphorylation appear to be downregulated in diabetic retinal endothelial cells, and fasudil treatment has also proven to have a positive effect in reversing the decreased expression of this enzyme, but the production of nitric oxide (NO) is required to be maintained for fasudil to work [22–27,30].
- Before showing table 2 about clinical studies, a summary table of preclinical trials should be shown regarding the efficacy and safety of ROCK inhibitors in animals.
- We agree with the reviewer that a summary table of preclinical trials regarding these topics would be helpful. This was the reason why we added to table 1 (Page 11), which is a summary of the findings from those trials. Nevertheless, we added some more information in section 6 about the efficacy and safety of ROCK inhibitors.
- The following text was included in section 6, page 13, line 386:
“Since its discovered in the second half of the 90s, the modulation of rho kinase has been considered a potential therapeutic target to explore in a broad range of human disease, being cardiovascular disease one of the main areas in which they have been studied; unfortunately, its inhibition also carries the risk of unacceptable systemic side effects. In this framework, the eye is a possible application area due to the limited number of side effects, being the conjunctival hyperemia the main one.
Of note, the protein kinase family inhibitors have been historically classified according to their mechanism of action based on the binding modes, leading to a type I−V classification [4]. Although, a thorough analysis of the different mechanism of actions goes beyond the scope of this review, is essential to take into consideration that the majority of kinase inhibitors are based on type I mode, in which the inhibitor targets the ATP binding site of the kinase in its active conformation [4–6]. Another practical mode of classification is how they impact the kinase parameters, their difference substrate, and their competititive or non-competetive profile [5] ; being a matter of controversy, what is the best classification method.
Regarding the eye, vast majority of rho kinase inhibitors act on both, ROCK-1 and ROCK-2. Fasudil (HA-1077, Fasudil, Asahi Kasei Pharma Corporation, Tokyo, Japan), an isoquinoline derivate and a calcium antagonist, is approved for the treatment of cerebral vasospasms in Japan and China; its fluorinated analogue, with much more power and selectivity. Ripasudil (K-115, Glanatec® Ophthalmic Solution 0.4%, Kowa Company, Ltd., Japan) and netarsudil (AR-13324, Rhopressa® Ophthalmic Solution 0.02%, Aerie Pharmaceuticals, Inc., Irvine, California), an amino-isoquinoline amide, act on both kinases and are the best-known ones. The three of them have been approved for glaucoma treatment because of their ability to reduce intraocular pressure lowering aqueous outflow resistance [7–9]. Only Belumosudil (KD-025) is a selective inhibitor of ROCK-2 (phase 2 clinical trials) being considered as a potential treatment for patients with some systemic inflammatory conditions [10]. AMA-0076, GSK-269962A, H-1152, Y-27632, RKI-1447, SAR-407899, SB-772077-B, Wf-536 andY-39983, are also double-acting ROCK inhibitor molecules [7,8].
It is crucial to take into consideration the mode of administration as most of the glaucoma preclinical and clinical studies are based on topical applied ophthalmic solutions, having been demonstrated their safety profile and hypotensive properties but with a high incidence of conjunctival hyperemia and with limited intraocular bioavailability due to the short corneal residence time and their characteristic hydrophilic properties [9,11–14]. This is one reason under the high rate of studies based on intravitreal injections for pathologies involving the posterior segment of the eye, trying to increase the intraocular active-drug concentration [15,16]”.
- The following text was included in section 6, page 15 right after line 422:
“As previously explained, netarsudil, appears to be effective in lowering IOP, but with low intraocular bioavailability when administered topically [9]. Interestingly, netarsudil has particular physiochemical and pharmacological properties, as its more lipophilic profile, which makes it a suitable candidate for intravitreal administration. AR-13503, a prodrug of netarsudil with ROCK/PKC inhibition properties, is being investigated for its administration intravitreally in a biodegradable implant [55]. In vivo studies showed a safety profile and an improved bioavailability, being able to deliver adequate levels of AR-13503 to the posterior segment for a 5-6 month-time period [56].
Moreover, alone or in combination with anti-VEGF drugs, AR-13503 has shown an additive efficacy on antiangiogenic properties and RPE barrier functions protecting power on DR in vitro models[55,56]. These characteristics make AR-13503 implant a possible game-changing drug for the treatment of DME. The prodrug is ongoing a phase 1 safety testing clinical trial in DME and Neovascular Age-Related Macular Degeneration patients, which has not finished yet (NCT03835884).”
- Table 2 is about the clinical studies on humans. The title of the table should be more specific rather than "treatments".
- We have made the change. The new title reads as follows “Table 2. ROCK inhibitors as a therapeutical approach in patients with DMO. Summary of clinical studies”.
- Table 2. In all studies, the nature of the study such as prospective or retrospective should be described even in pilot studies.
- We have fixed the error and added the nature of the study in pilot ones. Therefore, both of them are prospective pilot studies (Table 2).
- Table 2. The outcomes should be described quantitatively with statistical significance.
- We agree and have updated table 2, adding quantitative parameters on average and, if any, their statistical significance (Table 2).
- As we already mentioned, table 2 was relocated right after Table 1 in section 5. All the suggestions in point 9-11 were conducted, and the result of Table 2 is the following one:
Table 2. ROCK inhibitors as a therapeutical approachin patients with DMO. Summary of clinical studies.
|
Author, year |
Type of study |
No. of patients (eyes) |
Inclusion criteria |
Therapy evaluated |
Parameters assessed |
Outcomes |
|
|
Case |
Control |
||||||
|
(Ahmadieh et al. 2013) |
Prospective pilot study |
5 (5) |
- |
Persistent DMO, previously treated with: · Macular laser photocoagulation · Intravitreal bevacizumab injections |
Intravitreal fasudil injection (0.025mg/0.05mL) combined with Intravitreal bevacizumab injection (1.25mg/0,05mL) |
BCVA CMT |
· No toxic retinal effect was found · BCVA improved from 0.82 to 0.34 logMAR (p = 0.04) and CMT improved from 409 to 314µm ( p = 0.04) avg. at week 6 |
|
(Nourinia et al. 2013) |
Prospective pilot study |
15 (15) |
- |
Refractory DMO, previously treated with: · Intravitreal bevacizumab injections |
Intravitreal fasudil injection (0.025mg/0.05mL) combined with Intravitreal bevacizumab injection (1.25mg/0,05mL) |
BCVA CMT |
· BCVA improved from 0.84 to 0.49 log-MAR (p = 0.003) and CMT improved from 448 to 347 µm (p = 0.001) avg. at week 4 |
|
(Ahmadieh et al. 2019) |
Prospective randomised clinical trial |
22 (22) |
22 (22) |
Severe DMO |
Intravitreal fasudil injection (50 µM/L) combined with Intravitreal bevacizumab injection (1.25mg/0,05mL) |
BCVA CMT |
· BCVA changed in 12 ETDRS letters more at month 6 in combined group vs. bevacizumab group (p<0.001). CMT was also significantly more pronounced at month 6 in combined group (p<0.001) |
|
(Minami et al. 2019) |
Retrospective study |
10 (12) |
10 (14) |
Patients under ripasudil treatment for PAG with DMO |
Ripasudil eyedrops |
BCVA FT IOP |
· Decreased IOP (2.7 mmHg less; p = 0.008) and FT(44 µm less; p = 0.01) after one month of treatment · No significant changes were found in the BCVA |
BCVA— Best Corrected Visual Acuity; CMT— Central Macular Thickness; PAG— Primary Angle Glaucoma; IOP— Intraocular Pressure; FT— Foveal Thickness; DMO— Diabetic Macular Oedema; avg. — on average
- Line 428 NTC -> NCT
- We have fixed the mistake in page 15 line 428: “It is interesting to note that the potential use of the intravitreal injection of bevacizumab and fasudil in refractory cases of macular oedema secondary to retinal vein occlusion is under investigation, although the results have not been published yet (NCT: 03391219)”.
- For the intravitreal drug to be effective for the chronic diseases such as DMO, the intraocular PK should be also considered. I hope the authors mention about it regarding ROCK inhibitors.
- We thank the reviewer for pointing this out since it made us include it when revising section 6. In the new version of this section, the intraocular PK of ROCK inhibitors was discussed. This information substituted pages 13-5, lines 385-428:
“A summary of studies in which ROCK inhibitors are used as a treatment was provided in Table 2.
Since its discovered in the second half of the 90s, the modulation of rho kinase has been considered a potential therapeutic target to explore in a broad range of human disease, being cardiovascular disease one of the main areas in which they have been studied; unfortunately, its inhibition also carries the risk of unacceptable systemic side effects. In this framework, the eye is a possible application area due to the limited number of side effects, being the conjunctival hyperemia the main one.
Of note, the protein kinase family inhibitors have been historically classified according to their mechanism of action based on the binding modes, leading to a type I−V classification [4]. Although, a thorough analysis of the different mechanism of actions goes beyond the scope of this review, is essential to take into consideration that the majority of kinase inhibitors are based on type I mode, in which the inhibitor targets the ATP binding site of the kinase in its active conformation [4–6]. Another practical mode of classification is how they impact the kinase parameters, their difference substrate, and their competititive or non-competetive profile (XX); being a matter of controversy, what is the best classification method.
Regarding the eye, vast majority of rho kinase inhibitors act on both, ROCK-1 and ROCK-2. Fasudil (HA-1077, Fasudil, Asahi Kasei Pharma Corporation, Tokyo, Japan), an isoquinoline derivate and a calcium antagonist, is approved for the treatment of cerebral vasospasms in Japan and China; its fluorinated analogue, with much more power and selectivity. Ripasudil (K-115, Glanatec® Ophthalmic Solution 0.4%, Kowa Company, Ltd., Japan) and netarsudil (AR-13324, Rhopressa® Ophthalmic Solution 0.02%, Aerie Pharmaceuticals, Inc., Irvine, California), an amino-isoquinoline amide, act on both kinases and are the best-known ones. The three of them have been approved for glaucoma treatment because of their ability to reduce intraocular pressure lowering aqueous outflow resistance [7–9]. Only Belumosudil (KD-025) is a selective inhibitor of ROCK-2 (phase 2 clinical trials) being considered as a potential treatment for patients with some systemic inflammatory conditions [10]. AMA-0076, GSK-269962A, H-1152, Y-27632, RKI-1447, SAR-407899, SB-772077-B, Wf-536 andY-39983, are also double-acting ROCK inhibitor molecules [7,8].
It is crucial to take into consideration the mode of administration as most of the glaucoma preclinical and clinical studies are based on topical applied ophthalmic solutions, having been demonstrated their safety profile and hypotensive properties but with a high incidence of conjunctival hyperemia and with limited intraocular bioavailability due to the short corneal residence time and their characteristic hydrophilic properties [9,11–14]. This is one reason under the high rate of studies based on intravitreal injections for pathologies involving the posterior segment of the eye, trying to increase the intraocular active-drug concentration [15,16].
(…)
Of note, they were aware of the limitations of the study, as it was a retrospective study with a small number of patients with a short follow-up period and the drug was administered topically, with implies a limited intraocular bioavailability. However, it should be considered for the development of further prospective clinical trials.
As previously explained, netarsudil, appears to be effective in lowering IOP, but with low intraocular bioavailability when administered topically [9]. Interestingly, netarsudil has particular physiochemical and pharmacological properties, as its more lipophilic profile, which makes it a suitable candidate for intravitreal administration. AR-13503, a prodrug of netarsudil with ROCK/PKC inhibition properties, is being investigated for its administration intravitreally in a biodegradable implant [55]. In vivo studies showed a safety profile and an improved bioavailability, being able to deliver adequate levels of AR-13503 to the posterior segment for a 5-6 month-time period [56].
Moreover, alone or in combination with anti-VEGF drugs, AR-13503 has shown an additive efficacy on antiangiogenic properties and RPE barrier functions protecting power on DR in vitro models[55,56]. These characteristics make AR-13503 implant a possible game-changing drug for the treatment of DME. The prodrug is ongoing a phase 1 safety testing clinical trial in DME and Neovascular Age-Related Macular Degeneration patients, which has not finished yet (NCT03835884).
Since the studies in this review are focused on DMO, questions arise as to whether ROCK inhibitors could be of benefit in other types of macular oedema, either as unique therapy or in combination with other therapies. It is interesting to note that the potential use of the intravitreal injection of bevacizumab and fasudil in refractory cases of macular oedema secondary to retinal vein occlusion is under investigation, although the results have not been published yet (NCT: 03391219).”
References
- Karst, S.G.; Schuster, M.; Mitsch, C.; Meyer, E.L.; Kundi, M.; Scholda, C.; Schmidt‐Erfurth, U.M. Atrophy of the Central Neuroretina in Patients Treated for Diabetic Macular Edema. Acta Ophthalmologica 2019, 97, e1054–e1061, doi:10.1111/aos.14173.
- Willmann, G.; Nepomuceno, A.B.; Messias, K.; Barroso, L.; Scott, I.U.; Messias, A.; Jorge, R. Foveal Thickness Reduction after Anti-Vascular Endothelial Growth Factor Treatment in Chronic Diabetic Macular Edema. International Journal of Ophthalmology 2017, 10, 760–764, doi:10.18240/ijo.2017.05.17.
- Romano, M.R.; Allegrini, D.; Della Guardia, C.; Schiemer, S.; Baronissi, I.; Ferrara, M.; Cennamo, G. Vitreous and Intraretinal Macular Changes in Diabetic Macular Edema with and without Tractional Components. Graefe’s Archive for Clinical and Experimental Ophthalmology 2019, 257, 1–8.
- van Linden, O.P.J.; Kooistra, A.J.; Leurs, R.; de Esch, I.J.P.; de Graaf, C. KLIFS: A Knowledge-Based Structural Database to Navigate Kinase-Ligand Interaction Space. Journal of Medicinal Chemistry 2014, 57, 249–277.
- Feng, Y.; Lograsso, P. v.; Defert, O.; Li, R. Rho Kinase (ROCK) Inhibitors and Their Therapeutic Potential. Journal of Medicinal Chemistry 2016, 59, 2269–2300, doi:10.1021/acs.jmedchem.5b00683.
- Zhao, Z.; Wu, H.; Wang, L.; Liu, Y.; Knapp, S.; Liu, Q.; Gray, N.S. Exploration of Type II Binding Mode: A Privileged Approach for Kinase Inhibitor Focused Drug Discovery? ACS Chemical Biology 2014, 9, 1230–1241.
- Koch, J.C.; Tatenhorst, L.; Roser, A.E.; Saal, K.A.; Tönges, L.; Lingor, P. ROCK Inhibition in Models of Neurodegeneration and Its Potential for Clinical Translation. Pharmacology and Therapeutics 2018, 189, 1–21, doi:10.1016/j.pharmthera.2018.03.008.
- Tanna, A.P.; Johnson, M. Rho Kinase Inhibitors as a Novel Treatment for Glaucoma and Ocular Hypertension. Ophthalmology 2018, 125, 1741–1756.
- Hoy, S.M. Netarsudil Ophthalmic Solution 0.02%: First Global Approval. Drugs 2018, 78, 389–396, doi:10.1007/s40265-018-0877-7.
- Jagasia, M.; Lazaryan, A.; Bachier, C.R.; Salhotra, A.; Weisdorf, D.J.; Zoghi, B.; Essell, J.; Green, L.; Schueller, O.; Patel, J.; et al. ROCK2 Inhibition With Belumosudil (KD025) for the Treatment of Chronic Graft-Versus-Host Disease. Journal of Clinical Oncology 2021, JCO.20.02754, doi:10.1200/jco.20.02754.
- Garnock-Jones, K.P. Ripasudil: First Global Approval. Drugs 2014, 74, 2211–2215, doi:10.1007/s40265-014-0333-2.
- Nakabayashi, S.; Kawai, M.; Yoshioka, T.; Song, Y.S.; Tani, T.; Yoshida, A.; Nagaoka, T. Effect of Intravitreal Rho Kinase Inhibitor Ripasudil (K-115) on Feline Retinal Microcirculation. Experimental Eye Research 2015, 139, 132–135, doi:10.1016/j.exer.2015.07.008.
- Pakravan, M.; Beni, A.N.; Ghahari, E.; Varshochian, R.; Yazdani, S.; Esfandiari, H.; Ahmadieh, H. The Ocular Hypotensive Efficacy of Topical Fasudil, a Rho-Associated Protein Kinase Inhibitor, in Patients with End-Stage Glaucoma. American Journal of Therapeutics 2016, 24, E676–E680, doi:10.1097/MJT.0000000000000362.
- Mietzner, R.; Breunig, M. Causative Glaucoma Treatment: Promising Targets and Delivery Systems. Drug Discovery Today 2019, 24, 1606–1613.
- Li, M.; Yasumura, D.; Ma, A.A.K.; Matthes, M.T.; Yang, H.; Nielson, G.; Huang, Y.; Szoka, F.C.; LaVail, M.M.; Diamond, M.I. Intravitreal Administration of HA-1077, a ROCK Inhibitor, Improves Retinal Function in a Mouse Model of Huntington Disease. PLoS ONE 2013, 8, doi:10.1371/journal.pone.0056026.
- Mietzner, R.; Kade, C.; Froemel, F.; Pauly, D.; Stamer, W.D.; Ohlmann, A.; Wegener, J.; Fuchshofer, R.; Breunig, M. Fasudil Loaded PLGA Microspheres as Potential Intravitreal Depot Formulation for Glaucoma Therapy. Pharmaceutics 2020, 12, 1–22, doi:10.3390/pharmaceutics12080706.
- Ruiz-Loredo, A.Y.; López, E.; López-Colomé, A.M. Thrombin Promotes Actin Stress Fiber Formation in RPE through Rho/ROCK-Mediated MLC Phosphorylation. Journal of Cellular Physiology 2011, 226, 414–423, doi:10.1002/jcp.22347.
- Jahani, V.; Kavousi, A.; Mehri, S.; Karimi, G. Rho Kinase, a Potential Target in the Treatment of Metabolic Syndrome. Biomedicine and Pharmacotherapy 2018, 106, 1024–1030.
- Moshirfar, M.; Parker, L.; Birdsong, O.C.; Ronquillo, Y.C.; Hofstedt, D.; Shah, T.J.; Gomez, A.T.; Hoopes, P.C.S. Use of Rho Kinase Inhibitors in Ophthalmology: A Review of the Literature. Medical hypothesis, discovery & innovation ophthalmology journal 2018, 7, 101–111.
- Rao, P.V.; Pattabiraman, P.P.; Kopczynski, C. Role of the Rho GTPase/Rho Kinase Signaling Pathway in Pathogenesis and Treatment of Glaucoma: Bench to Bedside Research. Experimental Eye Research 2017, 158, 23–32.
- Feng, Y.; Lograsso, P. V.; Defert, O.; Li, R. Rho Kinase (ROCK) Inhibitors and Their Therapeutic Potential. Journal of Medicinal Chemistry 2016, 59, 2269–2300.
- Yamashita, F.; Fukuyama, E.; Mizoguchi, K.; Takizawa, S.; Xu, S.; Kawakata, H. Scale Dependence of Rock Friction at High Work Rate. Nature 2015, 528, 254–257, doi:10.1038/nature16138.
- Shimizu, Y.; Dobashi, K.; Sano, T.; Yamada, M. Rock Activation in Lung of Idiopathic Pulmonary Fibrosis with Oxidative Stress. International Journal of Immunopathology and Pharmacology 2014, 27, 37–44, doi:10.1177/039463201402700106.
- Zhang, C.; Zhang, S.; Zhang, Z.; He, J.; Xu, Y.; Liu, S. ROCK Has a Crucial Role in Regulating Prostate Tumor Growth through Interaction with C-Myc. Oncogene 2014, 33, 5582–5591, doi:10.1038/onc.2013.505.
- Alleaume-Butaux, A.; Nicot, S.; Pietri, M.; Baudry, A.; Dakowski, C.; Tixador, P.; Ardila-Osorio, H.; Haeberlé, A.M.; Bailly, Y.; Peyrin, J.M.; et al. Double-Edge Sword of Sustained ROCK Activation in Prion Diseases through Neuritogenesis Defects and Prion Accumulation. PLoS Pathogens 2015, 11, doi:10.1371/journal.ppat.1005073.
- Halász, É.; Townes-Anderson, E. Rock Inhibitors in Ocular Disease. ADMET and DMPK 2016, 4, 280–301.
- Abbhi, V.; Piplani, P. Rho-Kinase (ROCK) Inhibitors - A Neuroprotective Therapeutic Paradigm with a Focus on Ocular Utility. Current Medicinal Chemistry 2018, 27, 2222–2256, doi:10.2174/0929867325666181031102829.
- Fukiage, C.; Mizutani, K.; Kawamoto, Y.; Azuma, M.; Shearer, T.R. Involvement of Phosphorylation of Myosin Phosphatase by ROCK in Trabecular Meshwork and Ciliary Muscle Contraction. Biochemical and Biophysical Research Communications 2001, 288, 296–300, doi:10.1006/bbrc.2001.5751.
- Sun, H.; Breslin, J.W.; Zhu, J.; Yuan, S.Y.; Wu, M.H. Rho and ROCK Signaling in VEGF-Induced Microvascular Endothelial Hyperpermeability. Microcirculation 2006, 13, 237–247, doi:10.1080/10739680600556944.
- Nourinia, R.; Nakao, S.; Zandi, S.; Safi, S.; Hafezi-Moghadam, A.; Ahmadieh, H. ROCK Inhibitors for the Treatment of Ocular Diseases. British Journal of Ophthalmology 2018, 102.
- Arita, R.; Hata, Y.; Ishibashi, T. ROCK as a Therapeutic Target of Diabetic Retinopathy. Journal of Ophthalmology 2010, 2010, 1–9, doi:10.1155/2010/175163.
- Lu, Q.Y.; Chen, W.; Lu, L.; Zheng, Z.; Xu, X. Involvement of RhoA/ROCK1 Signaling Pathway in Hyperglycemia-Induced Microvascular Endothelial Dysfunction in Diabetic Retinopathy. International Journal of Clinical and Experimental Pathology 2014, 7, 7268–7277.
- Mohammad, G.; AlSharif, H.M.; Siddiquei, M.M.; Ahmad, A.; Alam, K.; El-Asrar, A.M.A. Rho-Associated Protein Kinase-1 Mediates the Regulation of Inflammatory Markers in Diabetic Retina and in Retinal Müller Cells. Annals of Clinical and Laboratory Science 2018, 48, 137–145.
- Yao, J.; Wang, J.; Yao, Y.; Wang, K.; Zhou, Q.; Tang, Y. MiR‑133b Regulates Proliferation and Apoptosis in High‑glucose‑induced Human Retinal Endothelial Cells by Targeting Ras Homolog Family Member A. International Journal of Molecular Medicine 2018, 42, 839–850, doi:10.3892/ijmm.2018.3694.
- Ozturk, B.T.; Bozkurt, B.; Kerimoglu, H.; Okka, M.; Kamis, U.; Gunduz, K. Effect of Serum Cytokines and VEGF Levels on Diabetic Retinopathy and Macular Thickness. Molecular Vision 2009, 15, 1906–1914.
- Nawaz, M.I.; van Raemdonck, K.; Mohammad, G.; Kangave, D.; van Damme, J.; Abu El-Asrar, A.M.; Struyf, S. Autocrine CCL2, CXCL4, CXCL9 and CXCL10 Signal in Retinal Endothelial Cells and Are Enhanced in Diabetic Retinopathy. Experimental Eye Research 2013, 109, 67–76, doi:10.1016/j.exer.2013.01.008.
- Mohammad, G.; AlSharif, H.M.; Siddiquei, M.M.; Ahmad, A.; Alam, K.; El-Asrar, A.M.A. Rho-Associated Protein Kinase-1 Mediates the Regulation of Inflammatory Markers in Diabetic Retina and in Retinal Müller Cells. Annals of Clinical and Laboratory Science 2018, 48, 137–145.
- Yamaguchi, M.; Nakao, S.; Arita, R.; Kaizu, Y.; Arima, M.; Zhou, Y.; Kita, T.; Yoshida, S.; Kimura, K.; Isobe, T.; et al. Vascular Normalization by ROCK Inhibitor: Therapeutic Potential of Ripasudil (K-115) Eye Drop in Retinal Angiogenesis and Hypoxia. Investigative Ophthalmology and Visual Science 2016, 57, 2264–2276, doi:10.1167/iovs.15-17411.
- Bryan, B.A.; Dennstedt, E.; Mitchell, D.C.; Walshe, T.E.; Noma, K.; Loureiro, R.; Saint‐Geniez, M.; Campaigniac, J.; Liao, J.K.; Patricia, D.A. RhoA/ROCK Signaling Is Essential for Multiple Aspects of VEGF‐mediated Angiogenesis. The FASEB Journal 2010, 24, 3186–3195, doi:10.1096/fj.09-145102.
- Zhang, X.H.; Feng, Z.H.; Wang, X.Y. The ROCK Pathway Inhibitor Y-27632 Mitigates Hypoxia and Oxidative Stress-Induced Injury to Retinal Muller Cells. Neural Regeneration Research 2018, 13, 549–555, doi:10.4103/1673-5374.228761.
- Yokota, T.; Utsunomiya, K.; Taniguchi, K.; Gojo, A.; Kurata, H.; Tajima, N. Involvement of the Rho/Rho Kinase Signaling Pathway in Platelet-Derived Growth Factor BB-Induced Vascular Endothelial Growth Factor Expression in Diabetic Rat Retina. Japanese Journal of Ophthalmology 2007, 51, 424–430, doi:10.1007/s10384-007-0471-0.
- Rothschild, P.R.; Salah, S.; Berdugo, M.; Gélizé, E.; Delaunay, K.; Naud, M.C.; Klein, C.; Moulin, A.; Savoldelli, M.; Bergin, C.; et al. ROCK-1 Mediates Diabetes-Induced Retinal Pigment Epithelial and Endothelial Cell Blebbing: Contribution to Diabetic Retinopathy. Scientific Reports 2017, 7, 1–15, doi:10.1038/s41598-017-07329-y.
- Arita, R.; Hata, Y.; Nakao, S.; Kita, T.; Miura, M.; Kawahara, S.; Zandi, S.; Almulki, L.; Tayyari, F.; Shimokawa, H.; et al. Rho Kinase Inhibition by Fasudil Ameliorates Diabetes-Induced Microvascular Damage. Diabetes 2009, 58, 215–226, doi:10.2337/db08-0762.
- Rizzolo, L.J. Development and Role of Tight Junctions in the Retinal Pigment Epithelium. International Review of Cytology 2007, 258, 195–234, doi:10.1016/S0074-7696(07)58004-6.
- Chen, Y.L.; Ren, Y.; Xu, W.; Rosa, R.H.; Kuo, L.; Hein, T.W. Constriction of Retinal Venules to Endothelin-1: Obligatory Roles of ET A Receptors, Extracellular Calcium Entry, and Rho Kinase. Investigative Ophthalmology and Visual Science 2018, 59, 5167–5175, doi:10.1167/iovs.18-25369.
- Rosa, R.H. Divergent Roles of Nitric Oxide and Rho Kinase in Vasomotor Regulation of Human Retinal Arterioles. 2009, doi:10.1167/iovs.09-4391.
- Yamaguchi, M.; Nakao, S.; Arita, R.; Kaizu, Y.; Arima, M.; Zhou, Y.; Kita, T.; Yoshida, S.; Kimura, K.; Isobe, T.; et al. Vascular Normalization by ROCK Inhibitor: Therapeutic Potential of Ripasudil (K-115) Eye Drop in Retinal Angiogenesis and Hypoxia. Investigative Ophthalmology and Visual Science 2016, 57, 2264–2276, doi:10.1167/iovs.15-17411.
- Nagaoka, T.; Hein, T.W.; Yoshida, A.; Kuo, L. Simvastatin Elicits Dilation of Isolated Porcine Retinal Arterioles: Role of Nitric Oxide and Mevalonate-Rho Kinase Pathways., doi:10.1167/iovs.06-0856.
- Arima, M.; Nakao, S.; Yamaguchi, M.; Feng, H.; Fujii, Y.; Ishibashi, T.; Stitt, A.; Sonoda, K. Claudin-5 Redistribution Induced by Inflammation Leads to Anti-. 1–79.
- Fagerholm, S.C.; Guenther, C.; Asens, M.L.; Savinko, T.; Uotila, L.M. Beta2-Integins and Interacting Proteins in Leukocyte Trafficking, Immune Supression, and Immunodeficiency Disease. Frontiers in Immunology 2019, 10, 1–10, doi:10.3389/fimmu.2019.00254.
- Joussen, A.M.; Murata, T.; Tsujikawa, A.; Kirchhof, B.; Bursell, S.E.; Adamis, A.P. Leukocyte-Mediated Endothelial Cell Injury and Death in the Diabetic Retina. American Journal of Pathology 2001, 158, 147–152, doi:10.1016/S0002-9440(10)63952-1.
- Barreiro, O.; Yáñez-Mó, M.; Serrador, J.M.; Montoya, M.C.; Vicente-Manzanares, M.; Tejedor, R.; Furthmayr, H.; Sánchez-Madrid, F. Dynamic Interaction of VCAM-1 and ICAM-1 with Moesin and Ezrin in a Novel Endothelial Docking Structure for Adherent Leukocytes. Journal of Cell Biology 2002, 157, 1233–1245, doi:10.1083/jcb.200112126.
- Nourinia, R.; Ahmadieh, H.; Shahheidari, M.H.; Zandi, S.; Nakao, S.; Hafezi-Moghadam, A. Intravitreal Fasudil Combined with Bevacizumab for Treatment of Refractory Diabetic Macular Edema; a Pilot Study. Journal of Ophthalmic and Vision Research 2013, 8, 337–340.
- Ahmadieh, H.; Nourinia, R.; Hafezi-Moghadam, A. Intravitreal Fasudil Combined with Bevacizumab for Persistent Diabetic Macular Edema: A Novel Treatment. JAMA Ophthalmology 2013, 131, 923–924, doi:10.1001/jamaophthalmol.2013.143.
- Glendenning, A.; Crews, K.; Sturdivant, J.; A deLong, M.; Kopczynski, C.; Lin, C.-W. Sustained Release, Biodegradable PEA Implants for Intravitreal Delivery of the ROCK/PKC Inhibitor AR-13503 | IOVS | ARVO Journals. Invest. Ophthalmol. Vis. Sci. 2018, 59, 5672.
- Ding, J.; Crews, K.; Carbajal, K.; Weksler, M.; Moore, L.; Carlson, E.C.; Lin, C.-W. Ocular Tissue Distribution and Duration of Release of AR-13503 Following Administration of AR-13503 Sustained Release Intravitreal Implant in Rabbits and Miniature Swine | IOVS | ARVO Journals. Invest. Ophthalmol. Vis. Sci. 2019, 60, 5387.
Supplementary files
Supplementary file 1. English-editing-certificate

Reviewer 3 Report
The review is narrative, maybe a systematic design should be more appropriate.
No other Rho-kinase has been mentioned, such netarsudil, please add them to the search terms.
The manuscript focuses on DMO treatment, not on Rho Kinase in DMO, in my opinion, should be completely redesigned, focusing on the role of the Rho/ROCK signaling pathway in the pathogenesis of DMO and the studies published, not only reporting them but also deeply discussing their results.
Paragraphs 1,2,3, and 4 are only the introduction and should have a small part in the manuscript length.
Author Response
To whom may concern,
First and foremost, thank you for taking the time to assess our manuscript. We addressed all the concerns you raised and appreciate your patience. Our responses to your comments are detailed below, point by point and in the order presented. Line and page numbers used to locate all the revisions have been made according to the original manuscript uploaded.
Review Report Form 3
- The review is narrative; maybe a systematic design should be more appropriate.
- We disagree with your assertion. Our manuscript is not a narrative review. Despite we have not performed a Meta-analysis, our review was conducted following the guidelines for a systematic search in the literature. Please see below the flow diagram for our systematic review which included searches of databases and registers (Flow-diagram-PRISMA2020).
- No other Rho-kinase has been mentioned, such netarsudil. Please add them to the search terms.
- We have fixed the error. Netarsudil term was included in our review and so, as suggested, it has been included in search terms and figures (changes regarding the main manuscript have been highlighted with green colour):
- Page 1 lines 26-28: “A systematic literature search was performed, covering the years 1991 to 2021, using the following keywords: “rho-Associated Kinas-es”, “Diabetic Retinopathy”, “Macular Edema”, “Ripasudil” “Fasudil” and “Netarudil”.”
- Page 4, lines 77-80: “Potentially relevant articles were sought by using the search terms in combination as Medical Subject Headings (MeSH) terms and text words: “rho-Associated Kinases”, “Diabetic Retinopathy”, “Diabetic Macular Edema”, “Ripasudil”, “Fasudil” and “Netarsudil”“.
- The manuscript focuses on DMO treatment, not on Rho Kinase in DMO, in my opinion, it should be completely redesigned, focusing on the role of the Rho/ROCK signaling pathway in the pathogenesis of DMO studies published, not only reporting them but also deeply discussing their results.
- We agreed with your comment and decided to modify, reorganise and complete sections 5 and 6, the sections on ROCK signalling pathways and inhibitors topics.
- Point 5 was extended and reorganised into subsections. Figures and tables were modified, and the information of ROCK signalling pathway’s role in DMO supplemented. Figure 3 was also added. The following text complement section 5 from the original manuscript (from page 8 line 280 to page 13 line 376). “(…)” symbol was used to shorten the non-modified text from the original manuscript:
“5. Rho kinase and ROCK pathways
The Rho subfamily belongs to a member of the small molecule G protein in the Ras superfamily and has GTPase activity. It acts as a molecular switch (from inactive state (bound to GDP) to activated state (bound to GTP), exerting various biological effects by binding to its downstream target effector molecule (Figure 2). Among them, Rho-associated kinases (ROCK) are the most important effector molecules of the serine-threonine family that participate in the downstream signaling of Rho GTP-binding proteins [1–3]. There are two homologous isomers in the cell: ROCK1 (ROCK-I, ROKβ) and ROCK2 (ROCK-II, ROKα). Both are involved in the regulation of cell morphology, polarity, and cytoskeletal remodeling by regulating actin and cell migration, e.g., ROCKs phosphorylate myosin light chain (MLC) and LIM kinases to change actin cytoskeleton [4–6].
Figure 2. Rho activation and participation in DMO pathways. (A) In response to hyperglycaemia and in the presence of cytokines (TNF-α) and growing factors, such as VEGF and TGF-β, the activation of Rho by guanine exchange factors (GEF) is produced. GTPase-activating proteins (GAP) stimulate endogenous GTPase activity, facilitating GTP hydrolysis in GDP and inactivating Rho. (B) On the left side a normal-structured external retina is represented and on the right side an anomalous neurovascular junction in DMO. Rho-kinase signalling pathways are upregulated in Müller glial cells in response to hypoxia and oxidative stress and take part in DMO pathogenesis. (C) Representation of vessels from a normal retina (C,1) and a diabetic one (C,2). within inner-retinal vessels, rho pathways are involved in retinal vasoconstriction, endoluminal blebbing, endothelial impairment and leukostasis in response to an increased inflammation state and ICAM-1 and VCAM-1 hyperproduction in DMO. (D) At RPE level, ROCK participates in bleb formation and cellular remodelation and produces polarization and cell adhesion changes in response to hyperglycaemia modifying outer-BRB. In normal RPE cells ROCK-1 is cytoplasmic (D,1) but in diabetic conditions is recruited at the membrane (D,2). (E) Representation of an OCT image showing a cystoid DMO with hyperreflective dots as a result of the mentioned patho-physiological abnormalities. Rho kinase and ROCK pathways.
The ROCK molecular structure consists of three parts, the N-terminal kinase region, that catalyzes the phosphorylation/dephosphorylation of a series of downstream substrates; a coiled-coil region, in which the Rho-binding domain (RBD) regulates the activation signal of Rho transduction; and a C-terminal region terminal, which contains a pleckstrin homology (PH) and cysteine rich domain (CRD). This region contains a self-inhibiting region which interacts with the kinase domain to inhibit ROCK activity [7–10].
The ROCK1 and ROCK2 genes are located in the 18q11 and 2p24 chromosome regions. In addition, ROCK2 has a splicing mutant (mROCK2), called the small ROCK, which appears to be a product of partial gene duplication.
ROCK2 and mROCK2 are mainly localized to the cytoplasm, but ROCK2 is also localized to the plasma membrane through the C-terminal region. ROCK2 is highly expressed in brain and heart. However, ROCK1 can be concentrated in the plasma membrane of endothelium or in the center of the microtubule tissue of the moving cells, indicating its involvement in cell migration. ROCK1, is highly expressed in lung, liver, spleen, and kidney [7–10]. Both ROCK isomers are also present in ocular tissues and aberrant regulation of ROCK levels plays a role in the pathogenesis of corneal wound healing, glaucoma, diabetic retinopathy, and AMD [11–13].
5.1. Rock signalin pathway
ROCKs can be activated by the GTP-bound form of Rho and the activated form then phosphorylates downstream targets [11]. Binding of Rho-GTP opens the loop formation of the enzyme, and the activated form then phosphorylates downstream targets (Figure 2). Then, the activated ROCK can directly phosphorylate MLC, LIM Kinases, intermediate filament proteins, among other substrates, and in most cases, substrates are phosphorylated by their respective ROCK proteins. Taken together, Rho pathway activation leads to a concerted series of events resulting in increased actin-myosin contractility and cytoskeletal change. The C-terminus of the ROCK protein is a self-inhibiting region of kinase activity, including the RBD region and the PH region. In the inactive state, the RBD and PH regions of the C-terminus of the ROCK protein interact with the kinase domain to form a self-inhibiting loop (Figure 2) [7–11].
5.2. Relationship with diabetic retinopathy
Recently, RhoA and ROCK has been implicated in the pathogenesis of DR. The long-term glucose homeostasis imbalance, which leads to an accumulation of advanced glycation end products in the vessels. Under these inflammatory and hyperglycaemic diabetic conditions, guanine exchange factors (GEF) activate Rho, and subsequently activating ROCKs [6]. This abnormal expression of ROCK protein plays an important role in DMO pathogenesis, acting on retinal microvessels, leading to structural changes including vascular rigidity promoting leucocyte adhesion to the microvasculature by intercellular adhesion molecule-1 (ICAM-1) and contributing to VEGF-elicited microvascular hyperpermeability [14] (Figure 2). In addition, the activation of the Rho/ROCK signaling pathway regulates the NF-κB signaling pathway, which upregulates inflammatory genes, fibronectin and transcription factor AP-1 activation and accumulation of glomerular matrix proteins. Upregulation of Rho/ROCK pathway also results in the dephosphorylation of endothelial nitric oxide synthase (eNOS), which induces endothelial cell apoptosis and vasoconstriction [11–13,15].
Considering that increased activity of the Rho/ROCK pathway in diabetic patients contributes to exacerbates retinal vessel permeability, macular oedema, vascular occlusion and retinal ischaemia, several in vitro and in vivo studies have been conducted to assess the effect of Rho/ROCK pathway inhibition on diabetic retinopathy and DMO (Table 1 and Table 2).
Table 1. ROCK signalling involvement in diabetic retinopathy pathway.
|
Author, year |
Type of Study |
Model |
Outcomes |
|
|
Hyperglycaemia |
||||
|
(Lu et al. 2014) |
In vitro |
Choroid-retinal endothelial cell line |
High level of glucose increased RhoA activity |
|
|
(Mohammad et al. 2018) |
In vitro |
Human retinal Müller glial cells |
High glucose-induced upregulation of ROCK-1 in human retinal Müller glial cells |
|
|
(Yao et al. 2018) |
In vitro |
High-glucose-induced human retinal endothelial cells |
Rho/ROCK signalling pathway is involved in the hyperglycaemia-induced microvascular endothelial dysfunction |
|
|
Inflammation and vascularisation |
||||
|
(Mohammad et al. 2018) |
In vitro |
Human retinal Müller glial cells |
High glucose-induced upregulation of p-ERK ½, p-NF-κB, iNOS, VEGF and MCP-1/ CCL2 |
|
|
(Yamaguchi et al. 2016) |
In vitro |
Human retinal microvascular endothelial cells |
Increase in ROCK activity by VEGF stimulation |
|
|
(Bryan et al. 2010) |
Ex vivo/In vitro |
Ex vivo retinal explants Bovine retinal endothelial cells |
Rho/ROCK pathway mediates different VEGF-mediated angiogenesis processes |
|
|
RPE |
||||
|
(Rothschild et al. 2017)
|
In vivo
|
GK rat model |
Diabetic conditions modified the subcellular distribution of ROCK-1, which was located along the membrane, while ROCK-2 localisation was undisturbed |
|
|
Human retina tissue of diabetic patients |
Diabetic conditions modified the subcellular distribution of ROCK-1, which was located along the membrane, while ROCK-2 localisation was undisturbed |
|||
|
Vasoconstriction |
||||
|
(Rothschild et al. 2017) |
In vivo |
GK rat model |
Increased expression of ROCK-1 in mural cells and in the vessel wall of constricted areas |
|
|
(Chen et al. 2018) |
|
Porcine models |
ROCK signalling is involved in the venular constriction due to ET-1 |
|
|
(Rosa 2009) |
In vitro |
Human retina |
ET-1 provokes vasoconstriction of retinal arterioles via ROCK-signalling pathway |
|
|
(Nagaoka et al., n.d.) |
In vitro |
Porcine retinal arterioles |
Rho/ROCK pathway takes part in the simvastatin-induced NO-mediated dilatation of retinal arterioles |
|
|
Endothelial impairment |
||||
|
(Rothschild et al. 2017) |
In vivo
|
GK rat model |
ROCK-1 expression in endothelial cell blebs. |
|
|
Human retina tissue of diabetic patients |
||||
|
(Lu et al. 2014) |
|
|
Under hyperglycemic conditions, ROCK-1 contributes to endothelial dysfunction, through the damage of tight junctions |
|
|
(Arita et al. 2009) |
In vitro |
Diabetic rat model |
Increased levels of activated Rho in diabetic rat retinas |
|
|
Leukostasis |
||||
|
(Arita et al. 2009)
|
|
Diabetic rat model
|
CD11b and CD18 fluorescence in neutrophils of diabetic rats were higher, while CD11a fluorescence intensity did not differ |
|
|
ICAM expression was significantly higher in diabetic retinas |
||||
GK—Goto-Kakizaki, ET-1—Endothelin-1; CD—cluster of differentiation; MCP-1—Monocyte Chemoattractant Protein-1
Table 2. ROCK inhibitors as a therapeutical approachin patients with DMO. Summary of clinical studies.
|
Author, year |
Type of study |
No. of patients (eyes) |
Inclusion criteria |
Therapy evaluated |
Parameters assessed |
Outcomes |
|
|
Case |
Control |
||||||
|
(Ahmadieh et al. 2013) |
Prospective pilot study |
5 (5) |
- |
Persistent DMO, previously treated with: · Macular laser photocoagulation · Intravitreal bevacizumab injections |
Intravitreal fasudil injection (0.025mg/0.05mL) combined with Intravitreal bevacizumab injection (1.25mg/0,05mL) |
BCVA CMT |
· No toxic retinal effect was found · BCVA improved from 0.82 to 0.34 logMAR (p = 0.04) and CMT improved from 409 to 314µm ( p = 0.04) avg. at week 6 |
|
(Nourinia et al. 2013) |
Prospective pilot study |
15 (15) |
- |
Refractory DMO, previously treated with: · Intravitreal bevacizumab injections |
Intravitreal fasudil injection (0.025mg/0.05mL) combined with Intravitreal bevacizumab injection (1.25mg/0,05mL) |
BCVA CMT |
· BCVA improved from 0.84 to 0.49 log-MAR (p = 0.003) and CMT improved from 448 to 347 µm (p = 0.001) avg. at week 4 |
|
(Ahmadieh et al. 2019) |
Prospective randomised clinical trial |
22 (22) |
22 (22) |
Severe DMO |
Intravitreal fasudil injection (50 µM/L) combined with Intravitreal bevacizumab injection (1.25mg/0,05mL) |
BCVA CMT |
· BCVA changed in 12 ETDRS letters more at month 6 in combined group vs. bevacizumab group (p<0.001). CMT was also significantly more pronounced at month 6 in combined group (p<0.001) |
|
(Minami et al. 2019) |
Retrospective study |
10 (12) |
10 (14) |
Patients under ripasudil treatment for PAG with DMO |
Ripasudil eyedrops |
BCVA FT IOP |
· Decreased IOP (2.7 mmHg less; p = 0.008) and FT(44 µm less; p = 0.01) after one month of treatment · No significant changes were found in the BCVA |
BCVA— Best Corrected Visual Acuity; CMT— Central Macular Thickness; PAG— Primary Angle Glaucoma; IOP— Intraocular Pressure; FT— Foveal Thickness; DMO— Diabetic Macular Oedema; avg. — on average
5.3. Involvement of Rho/ROCK pathway in the pathogenesis of DR
As mentioned before, the pathogenesis of DMO is multifactorial and complex involving several mechanisms in which Rho/ROCK pathway is actively engaged. Recent studies have revealed that Rho and its target protein ROCK are implicated in important pathological pathways in DR and DMO such as hyperglycaemia, RPE disturbances, vasoconstriction, endothelial impairment, leukostasis and inflammation, and vascularization (Table 1 and Figure 3) [11–13,15,16].
Figure 3. ROCKs involvement in the mechanisms underlying DMO according to the results of preclinical studies. Together with inflammation, elevated glucose, which is a thorn in DR, constitutes a strong stimulus for Rho pathways activation and ROCK-1 upregulation. Moreover, VEGF overproduction because of DR, also increases ROCK activity, particularly those pathways in charge of VEGF-mediated angiogenic processes, getting caught in a vicious circle. At RPE level, ROCK-1 cytoplasmic proteins translocate to cell membrane and participate in actin cytoskeleton and tight cell junctions remodelation. At a vascular level, the calibre of the retinal vessels is also regulated by ROCKs related pathways. In addition, they contribute to endothelial impairment by ROCK-1 blebs formation and tight junction modifications, as happened at RPE. Finally, these kinases stimulate ICAM expression, which facilitate leukocyte adhesion and contributes to leukostasis. Thus, ROCKs take part in leukostasis, inflammation and vascularisation processes and participate in endothelial and RPE impairment, perpetuating DMO pathogenesis.
5.3.1. Hyperglycaemia
Hyperglycaemia is one of the most important stimuli for ROCK-1 activation, and the correlation between diabetes and microvascular endothelial cell dysfunction has been demonstrated in various studies before. e.g., on choroid-retinal endothelial cell lines, it has been shown by a meaningful increase in phospho-myosin phosphatase target protein-1 (p-MYPT-1) as a result of this overactivity in response to elevated glucose [17]. Rho activity is also involved in the pathogenesis of renal and aortic complications during diabetic states [18,19]. In addition, hyperglycaemia has been shown to increase the expression of monocyte chemoattractant protein-1 (MCP-1/CCL2) and VEGF by vascular endothelial cells and in the vitreous samples from patients with DR [20,21]. It has been reported that the inhibition of RhoA/ROCK1 may attenuate the hypertonicity of endothelial cell caused by high glucose microenvironment, partly block inflammation and prevented considerably the apoptosis of endothelial cells aroused by high glucose [12,13,16,17].
5.3.2. Inflammation and vascularization
The levels of the proinflammatory mediators are increased in the ocular microenvironment of patients with DR, suggesting that persistent inflammation is critical for DR initiation and progression.
Several studies have documented that diabetes enhances the production of inflammatory mediators such as p-ERK ½, p-NF-κB, iNOS, VEGF and MCP-1/ CCL2), and how Rho-kinases play a crucial role in the inflammatory signalling, proliferation, fibrosis, and apoptosis through mitogen-activated protein kinase (MAPK) p38MAP kinase and NF-κB activation. Activation of NF-κB induces the upregulation of leukocyte adhesion molecules and the production of proinflammatory cytokines and angiogenic factors [20].
As mentioned, inflammatory factors play a key role in DMO pathogenesis, and their own signalling pathways and expression are influenced by Rho kinases. In vitro assays has also been conducted to prove this relationship using human retinal Müller glial cells and microvascular endothelial cells, ex vivo retinal explants, bovine retinal endothelial cells and rat retinal cells [22–26]. Results of recent research indicate that ROCK-1 activation induces focal vascular constrictions, endoluminal blebbing, retinal hypoxia, and remodelling of RPE cells contributing to outer barrier breakdown [18,27]. The blebbing-induced closure reversed by ROCK inhibitors, could open a window for intervention in case of macular ischemia.
VEGF stimulates ROCK activity and rho/ROCK pathways, particularly those mediating VEGF-dependent angiogenic processes on proliferative and non-proliferative DR, i.e. Rho/ROCK pathways regulate VEGF expression induced by platelet-derived growth factor (PDGF)-BB in retinal vessel pericytes [23,24,26]. Rho kinase signaling is also involved in the injury of inner retinal cells such as Müller cells in the case of hypoxia and oxidative stress [25].
5.3.3. RPE disturbances
Diabetic hyperglycaemia, TNF-α, and TGF-β activate and increase the ROCK pathway at endothelial and RPE cells and hyalocytes [17,28]. ROCK-1 proteins are usually cytoplasmic proteins in rat and human RPEs and endothelial cells, but in diabetic induced models, it has been demonstrated to translocate to the membrane accompanying ROCK-1 positive cell membrane bleb formations…
(…)
5.3.4. Vasoconstriction
Vasoconstriction is an important mechanism for retinal regulation and for proper functioning, which is damaged in DMO. Rat, porcine and human retinal models and in vitro and in vivo assays have shown Rho participation in this damage by multiple mechanisms. ROCK signalling is involved in retinal arteriolar and venular constriction due to vascular endothelin-1 (ET-1), a potent vasoconstrictor secreted by vascular endothelial cells [29,30]. In fact, expression of ROCK-1 was increased in mural cells, and there exist ROCK-1 positive blebs on the vessel wall of constricted areas from diabetic rats, narrowing retinal vessels lumen [27]. It has been shown promising results in hypoxia-induced retinal neovascularization animal models just using topical ROCK inhibitors. This study showed that the topical ROCK inhibitor inhibited neovascularisation and could enhance normal retinal vascularisation, opening a new horizon for treatment of DR without affecting normal retinal vasculature or reducing the neuroretinal damage [31].
(…)
5.3.5. Endothelial impairmet
ROCK-1 translocation and blebbing and an increased level of activated Rho in endothelial cells were also found in diabetic rat retinas producing endothelial impairment and the DMO…
(…)
5.3.6. Leukostasis
CD11b/CD18 is a beta-2 integrin, a leukocyte-specific transmembrane protein participating in neutrophils adhesion, communication and migration…
(…)
Intravitreal fasudil, ripasudil (K-115) and AMA0428, ROCK inhibitors suppressed and reduced leukocyte migration, inhibited ICAM-1 expression in endothelial cells and decreased vascular leakage and neurodegeneration of diabetic retinal vessels, thereby improving retinal thickness and DMO [28,32–34].
The eNOS enzyme and its phosphorylation appear to be downregulated in diabetic retinal endothelial cells, and fasudil treatment has also proven to have a positive effect in reversing the decreased expression of this enzyme, but the production of nitric oxide (NO) is required to be maintained for fasudil to work [7–12,15].
- Section 6 was also supplemented by adding information about ROCK inhibitors history, classification and types and recent advancements on netarsudil topic. Table 2 was relocated in section 5. The present text is intended to complement section 6 from the original manuscript from pages 13- 15 and lines 377-428:
“6. From the bench to the bedside
Anti-VEGF therapy in DMO treatment has changed the management of this condition, improving the control of the disease. However, anatomical and visual results are not always as expected, and some cases remain refractory to this treatment. Therefore, the identification of adjuvant or alternative therapies is required, the modulation of Rho/ROCK pathway is a potential candidate, since the suppression of leukocyte adhesion and neutrophil-induced retinal endothelial cell damage has been shown with its inhibition, which could have beneficial effects in terms of anatomical and visual outcomes. A summary of studies in which ROCK inhibitors are used as a treatment was provided in Table 2.
Since its discovered in the second half of the 90s, the modulation of rho kinase has been considered a potential therapeutic target to explore in a broad range of human disease, being cardiovascular disease one of the main areas in which they have been studied; unfortunately, its inhibition also carries the risk of unacceptable systemic side effects. In this framework, the eye is a possible application area due to the limited number of side effects, being the conjunctival hyperemia the main one.
Of note, the protein kinase family inhibitors have been historically classified according to their mechanism of action based on the binding modes, leading to a type I−V classification [35]. Although, a thorough analysis of the different mechanism of actions goes beyond the scope of this review, is essential to take into consideration that the majority of kinase inhibitors are based on type I mode, in which the inhibitor targets the ATP binding site of the kinase in its active conformation [35–37]. Another practical mode of classification is how they impact the kinase parameters, their difference substrate, and their competititive or non-competetive profile [36]; being a matter of controversy, what is the best classification method.
Regarding the eye, vast majority of rho kinase inhibitors act on both, ROCK-1 and ROCK-2. Fasudil (HA-1077, Fasudil, Asahi Kasei Pharma Corporation, Tokyo, Japan), an isoquinoline derivate and a calcium antagonist, is approved for the treatment of cerebral vasospasms in Japan and China; its fluorinated analogue, with much more power and selectivity. Ripasudil (K-115, Glanatec® Ophthalmic Solution 0.4%, Kowa Company, Ltd., Japan) and netarsudil (AR-13324, Rhopressa® Ophthalmic Solution 0.02%, Aerie Pharmaceuticals, Inc., Irvine, California), an amino-isoquinoline amide, act on both kinases and are the best-known ones. The three of them have been approved for glaucoma treatment because of their ability to reduce intraocular pressure lowering aqueous outflow resistance [6,38,39]. Only Belumosudil (KD-025) is a selective inhibitor of ROCK-2 (phase 2 clinical trials) being considered as a potential treatment for patients with some systemic inflammatory conditions [40]. AMA-0076, GSK-269962A, H-1152, Y-27632, RKI-1447, SAR-407899, SB-772077-B, Wf-536 andY-39983, are also double-acting ROCK inhibitor molecules [6,38].
It is crucial to take into consideration the mode of administration as most of the glaucoma preclinical and clinical studies are based on topical applied ophthalmic solutions, having been demonstrated their safety profile and hypotensive properties but with a high incidence of conjunctival hyperemia and with limited intraocular bioavailability due to the short corneal residence time and their characteristic hydrophilic properties [39,41–44]. This is one reason under the high rate of studies based on intravitreal injections for pathologies involving the posterior segment of the eye, trying to increase the intraocular active-drug concentration [45,46].
(…)
Of note, they were aware of the limitations of the study, as it was a retrospective study with a small number of patients with a short follow-up period and the drug was administered topically, with implies a limited intraocular bioavailability. However, it should be considered for the development of further prospective clinical trials.
As previously explained, netarsudil, appears to be effective in lowering IOP, but with low intraocular bioavailability when administered topically [39]. Interestingly, netarsudil has particular physiochemical and pharmacological properties, as its more lipophilic profile, which makes it a suitable candidate for intravitreal administration. AR-13503, a prodrug of netarsudil with ROCK/PKC inhibition properties, is being investigated for its administration intravitreally in a biodegradable implant [47]. In vivo studies showed a safety profile and an improved bioavailability, being able to deliver adequate levels of AR-13503 to the posterior segment for a 5-6 month-time period [48].
Moreover, alone or in combination with anti-VEGF drugs, AR-13503 has shown an additive efficacy on antiangiogenic properties and RPE barrier functions protecting power on DR in vitro models[47,48]. These characteristics make AR-13503 implant a possible game-changing drug for the treatment of DME. The prodrug is ongoing a phase 1 safety testing clinical trial in DME and Neovascular Age-Related Macular Degeneration patients, which has not finished yet (NCT03835884).
Since the studies in this review are focused on DMO, questions arise as to whether ROCK inhibitors could be of benefit in other types of macular oedema, either as unique therapy or in combination with other therapies. It is interesting to note that the potential use of the intravitreal injection of bevacizumab and fasudil in refractory cases of macular oedema secondary to retinal vein occlusion is under investigation, although the results have not been published yet (NCT: 03391219).”
- The results of each study have been discussed as presented. Moreover, a discussion section was added at the end of point 6, as can be read in the text above. We hope that after making these modifications, the purpose of our manuscript is more straightforward and better focused.
- Paragraphs 1,2,3, and 4 are only the introduction and should have a small part in the manuscript length.
- We appreciate the reviewer suggestion. However, our group believes it is useful to contextualize the review and introduce the material to a less experienced audience .
References
- Ruiz-Loredo, A.Y.; López, E.; López-Colomé, A.M. Thrombin Promotes Actin Stress Fiber Formation in RPE through Rho/ROCK-Mediated MLC Phosphorylation. Journal of Cellular Physiology 2011, 226, 414–423, doi:10.1002/jcp.22347.
- Jahani, V.; Kavousi, A.; Mehri, S.; Karimi, G. Rho Kinase, a Potential Target in the Treatment of Metabolic Syndrome. Biomedicine and Pharmacotherapy 2018, 106, 1024–1030.
- Moshirfar, M.; Parker, L.; Birdsong, O.C.; Ronquillo, Y.C.; Hofstedt, D.; Shah, T.J.; Gomez, A.T.; Hoopes, P.C.S. Use of Rho Kinase Inhibitors in Ophthalmology: A Review of the Literature. Medical hypothesis, discovery & innovation ophthalmology journal 2018, 7, 101–111.
- Rao, P.V.; Pattabiraman, P.P.; Kopczynski, C. Role of the Rho GTPase/Rho Kinase Signaling Pathway in Pathogenesis and Treatment of Glaucoma: Bench to Bedside Research. Experimental Eye Research 2017, 158, 23–32.
- Feng, Y.; Lograsso, P. V.; Defert, O.; Li, R. Rho Kinase (ROCK) Inhibitors and Their Therapeutic Potential. Journal of Medicinal Chemistry 2016, 59, 2269–2300.
- Koch, J.C.; Tatenhorst, L.; Roser, A.E.; Saal, K.A.; Tönges, L.; Lingor, P. ROCK Inhibition in Models of Neurodegeneration and Its Potential for Clinical Translation. Pharmacology and Therapeutics 2018, 189, 1–21, doi:10.1016/j.pharmthera.2018.03.008.
- Yamashita, F.; Fukuyama, E.; Mizoguchi, K.; Takizawa, S.; Xu, S.; Kawakata, H. Scale Dependence of Rock Friction at High Work Rate. Nature 2015, 528, 254–257, doi:10.1038/nature16138.
- Shimizu, Y.; Dobashi, K.; Sano, T.; Yamada, M. Rock Activation in Lung of Idiopathic Pulmonary Fibrosis with Oxidative Stress. International Journal of Immunopathology and Pharmacology 2014, 27, 37–44, doi:10.1177/039463201402700106.
- Zhang, C.; Zhang, S.; Zhang, Z.; He, J.; Xu, Y.; Liu, S. ROCK Has a Crucial Role in Regulating Prostate Tumor Growth through Interaction with C-Myc. Oncogene 2014, 33, 5582–5591, doi:10.1038/onc.2013.505.
- Alleaume-Butaux, A.; Nicot, S.; Pietri, M.; Baudry, A.; Dakowski, C.; Tixador, P.; Ardila-Osorio, H.; Haeberlé, A.M.; Bailly, Y.; Peyrin, J.M.; et al. Double-Edge Sword of Sustained ROCK Activation in Prion Diseases through Neuritogenesis Defects and Prion Accumulation. PLoS Pathogens 2015, 11, doi:10.1371/journal.ppat.1005073.
- Halász, É.; Townes-Anderson, E. Rock Inhibitors in Ocular Disease. ADMET and DMPK 2016, 4, 280–301.
- Abbhi, V.; Piplani, P. Rho-Kinase (ROCK) Inhibitors - A Neuroprotective Therapeutic Paradigm with a Focus on Ocular Utility. Current Medicinal Chemistry 2018, 27, 2222–2256, doi:10.2174/0929867325666181031102829.
- Fukiage, C.; Mizutani, K.; Kawamoto, Y.; Azuma, M.; Shearer, T.R. Involvement of Phosphorylation of Myosin Phosphatase by ROCK in Trabecular Meshwork and Ciliary Muscle Contraction. Biochemical and Biophysical Research Communications 2001, 288, 296–300, doi:10.1006/bbrc.2001.5751.
- Sun, H.; Breslin, J.W.; Zhu, J.; Yuan, S.Y.; Wu, M.H. Rho and ROCK Signaling in VEGF-Induced Microvascular Endothelial Hyperpermeability. Microcirculation 2006, 13, 237–247, doi:10.1080/10739680600556944.
- Nourinia, R.; Nakao, S.; Zandi, S.; Safi, S.; Hafezi-Moghadam, A.; Ahmadieh, H. ROCK Inhibitors for the Treatment of Ocular Diseases. British Journal of Ophthalmology 2018, 102.
- Arita, R.; Hata, Y.; Ishibashi, T. ROCK as a Therapeutic Target of Diabetic Retinopathy. Journal of Ophthalmology 2010, 2010, 1–9, doi:10.1155/2010/175163.
- Lu, Q.Y.; Chen, W.; Lu, L.; Zheng, Z.; Xu, X. Involvement of RhoA/ROCK1 Signaling Pathway in Hyperglycemia-Induced Microvascular Endothelial Dysfunction in Diabetic Retinopathy. International Journal of Clinical and Experimental Pathology 2014, 7, 7268–7277.
- Mohammad, G.; AlSharif, H.M.; Siddiquei, M.M.; Ahmad, A.; Alam, K.; El-Asrar, A.M.A. Rho-Associated Protein Kinase-1 Mediates the Regulation of Inflammatory Markers in Diabetic Retina and in Retinal Müller Cells. Annals of Clinical and Laboratory Science 2018, 48, 137–145.
- Yao, J.; Wang, J.; Yao, Y.; Wang, K.; Zhou, Q.; Tang, Y. MiR‑133b Regulates Proliferation and Apoptosis in High‑glucose‑induced Human Retinal Endothelial Cells by Targeting Ras Homolog Family Member A. International Journal of Molecular Medicine 2018, 42, 839–850, doi:10.3892/ijmm.2018.3694.
- Ozturk, B.T.; Bozkurt, B.; Kerimoglu, H.; Okka, M.; Kamis, U.; Gunduz, K. Effect of Serum Cytokines and VEGF Levels on Diabetic Retinopathy and Macular Thickness. Molecular Vision 2009, 15, 1906–1914.
- Nawaz, M.I.; van Raemdonck, K.; Mohammad, G.; Kangave, D.; van Damme, J.; Abu El-Asrar, A.M.; Struyf, S. Autocrine CCL2, CXCL4, CXCL9 and CXCL10 Signal in Retinal Endothelial Cells and Are Enhanced in Diabetic Retinopathy. Experimental Eye Research 2013, 109, 67–76, doi:10.1016/j.exer.2013.01.008.
- Mohammad, G.; AlSharif, H.M.; Siddiquei, M.M.; Ahmad, A.; Alam, K.; El-Asrar, A.M.A. Rho-Associated Protein Kinase-1 Mediates the Regulation of Inflammatory Markers in Diabetic Retina and in Retinal Müller Cells. Annals of Clinical and Laboratory Science 2018, 48, 137–145.
- Yamaguchi, M.; Nakao, S.; Arita, R.; Kaizu, Y.; Arima, M.; Zhou, Y.; Kita, T.; Yoshida, S.; Kimura, K.; Isobe, T.; et al. Vascular Normalization by ROCK Inhibitor: Therapeutic Potential of Ripasudil (K-115) Eye Drop in Retinal Angiogenesis and Hypoxia. Investigative Ophthalmology and Visual Science 2016, 57, 2264–2276, doi:10.1167/iovs.15-17411.
- Bryan, B.A.; Dennstedt, E.; Mitchell, D.C.; Walshe, T.E.; Noma, K.; Loureiro, R.; Saint‐Geniez, M.; Campaigniac, J.; Liao, J.K.; Patricia, D.A. RhoA/ROCK Signaling Is Essential for Multiple Aspects of VEGF‐mediated Angiogenesis. The FASEB Journal 2010, 24, 3186–3195, doi:10.1096/fj.09-145102.
- Zhang, X.H.; Feng, Z.H.; Wang, X.Y. The ROCK Pathway Inhibitor Y-27632 Mitigates Hypoxia and Oxidative Stress-Induced Injury to Retinal Muller Cells. Neural Regeneration Research 2018, 13, 549–555, doi:10.4103/1673-5374.228761.
- Yokota, T.; Utsunomiya, K.; Taniguchi, K.; Gojo, A.; Kurata, H.; Tajima, N. Involvement of the Rho/Rho Kinase Signaling Pathway in Platelet-Derived Growth Factor BB-Induced Vascular Endothelial Growth Factor Expression in Diabetic Rat Retina. Japanese Journal of Ophthalmology 2007, 51, 424–430, doi:10.1007/s10384-007-0471-0.
- Rothschild, P.R.; Salah, S.; Berdugo, M.; Gélizé, E.; Delaunay, K.; Naud, M.C.; Klein, C.; Moulin, A.; Savoldelli, M.; Bergin, C.; et al. ROCK-1 Mediates Diabetes-Induced Retinal Pigment Epithelial and Endothelial Cell Blebbing: Contribution to Diabetic Retinopathy. Scientific Reports 2017, 7, 1–15, doi:10.1038/s41598-017-07329-y.
- Arita, R.; Hata, Y.; Nakao, S.; Kita, T.; Miura, M.; Kawahara, S.; Zandi, S.; Almulki, L.; Tayyari, F.; Shimokawa, H.; et al. Rho Kinase Inhibition by Fasudil Ameliorates Diabetes-Induced Microvascular Damage. Diabetes 2009, 58, 215–226, doi:10.2337/db08-0762.
- Chen, Y.L.; Ren, Y.; Xu, W.; Rosa, R.H.; Kuo, L.; Hein, T.W. Constriction of Retinal Venules to Endothelin-1: Obligatory Roles of ET A Receptors, Extracellular Calcium Entry, and Rho Kinase. Investigative Ophthalmology and Visual Science 2018, 59, 5167–5175, doi:10.1167/iovs.18-25369.
- Rosa, R.H. Divergent Roles of Nitric Oxide and Rho Kinase in Vasomotor Regulation of Human Retinal Arterioles. 2009, doi:10.1167/iovs.09-4391.
- Yamaguchi, M.; Nakao, S.; Arita, R.; Kaizu, Y.; Arima, M.; Zhou, Y.; Kita, T.; Yoshida, S.; Kimura, K.; Isobe, T.; et al. Vascular Normalization by ROCK Inhibitor: Therapeutic Potential of Ripasudil (K-115) Eye Drop in Retinal Angiogenesis and Hypoxia. Investigative Ophthalmology and Visual Science 2016, 57, 2264–2276, doi:10.1167/iovs.15-17411.
- Barreiro, O.; Yáñez-Mó, M.; Serrador, J.M.; Montoya, M.C.; Vicente-Manzanares, M.; Tejedor, R.; Furthmayr, H.; Sánchez-Madrid, F. Dynamic Interaction of VCAM-1 and ICAM-1 with Moesin and Ezrin in a Novel Endothelial Docking Structure for Adherent Leukocytes. Journal of Cell Biology 2002, 157, 1233–1245, doi:10.1083/jcb.200112126.
- Nourinia, R.; Ahmadieh, H.; Shahheidari, M.H.; Zandi, S.; Nakao, S.; Hafezi-Moghadam, A. Intravitreal Fasudil Combined with Bevacizumab for Treatment of Refractory Diabetic Macular Edema; a Pilot Study. Journal of Ophthalmic and Vision Research 2013, 8, 337–340.
- Ahmadieh, H.; Nourinia, R.; Hafezi-Moghadam, A. Intravitreal Fasudil Combined with Bevacizumab for Persistent Diabetic Macular Edema: A Novel Treatment. JAMA Ophthalmology 2013, 131, 923–924, doi:10.1001/jamaophthalmol.2013.143.
- van Linden, O.P.J.; Kooistra, A.J.; Leurs, R.; de Esch, I.J.P.; de Graaf, C. KLIFS: A Knowledge-Based Structural Database to Navigate Kinase-Ligand Interaction Space. Journal of Medicinal Chemistry 2014, 57, 249–277.
- Feng, Y.; Lograsso, P. v.; Defert, O.; Li, R. Rho Kinase (ROCK) Inhibitors and Their Therapeutic Potential. Journal of Medicinal Chemistry 2016, 59, 2269–2300, doi:10.1021/acs.jmedchem.5b00683.
- Zhao, Z.; Wu, H.; Wang, L.; Liu, Y.; Knapp, S.; Liu, Q.; Gray, N.S. Exploration of Type II Binding Mode: A Privileged Approach for Kinase Inhibitor Focused Drug Discovery? ACS Chemical Biology 2014, 9, 1230–1241.
- Tanna, A.P.; Johnson, M. Rho Kinase Inhibitors as a Novel Treatment for Glaucoma and Ocular Hypertension. Ophthalmology 2018, 125, 1741–1756.
- Hoy, S.M. Netarsudil Ophthalmic Solution 0.02%: First Global Approval. Drugs 2018, 78, 389–396, doi:10.1007/s40265-018-0877-7.
- Jagasia, M.; Lazaryan, A.; Bachier, C.R.; Salhotra, A.; Weisdorf, D.J.; Zoghi, B.; Essell, J.; Green, L.; Schueller, O.; Patel, J.; et al. ROCK2 Inhibition With Belumosudil (KD025) for the Treatment of Chronic Graft-Versus-Host Disease. Journal of Clinical Oncology 2021, JCO.20.02754, doi:10.1200/jco.20.02754.
- Garnock-Jones, K.P. Ripasudil: First Global Approval. Drugs 2014, 74, 2211–2215, doi:10.1007/s40265-014-0333-2.
- Nakabayashi, S.; Kawai, M.; Yoshioka, T.; Song, Y.S.; Tani, T.; Yoshida, A.; Nagaoka, T. Effect of Intravitreal Rho Kinase Inhibitor Ripasudil (K-115) on Feline Retinal Microcirculation. Experimental Eye Research 2015, 139, 132–135, doi:10.1016/j.exer.2015.07.008.
- Pakravan, M.; Beni, A.N.; Ghahari, E.; Varshochian, R.; Yazdani, S.; Esfandiari, H.; Ahmadieh, H. The Ocular Hypotensive Efficacy of Topical Fasudil, a Rho-Associated Protein Kinase Inhibitor, in Patients with End-Stage Glaucoma. American Journal of Therapeutics 2016, 24, E676–E680, doi:10.1097/MJT.0000000000000362.
- Mietzner, R.; Breunig, M. Causative Glaucoma Treatment: Promising Targets and Delivery Systems. Drug Discovery Today 2019, 24, 1606–1613.
- Li, M.; Yasumura, D.; Ma, A.A.K.; Matthes, M.T.; Yang, H.; Nielson, G.; Huang, Y.; Szoka, F.C.; LaVail, M.M.; Diamond, M.I. Intravitreal Administration of HA-1077, a ROCK Inhibitor, Improves Retinal Function in a Mouse Model of Huntington Disease. PLoS ONE 2013, 8, doi:10.1371/journal.pone.0056026.
- Mietzner, R.; Kade, C.; Froemel, F.; Pauly, D.; Stamer, W.D.; Ohlmann, A.; Wegener, J.; Fuchshofer, R.; Breunig, M. Fasudil Loaded PLGA Microspheres as Potential Intravitreal Depot Formulation for Glaucoma Therapy. Pharmaceutics 2020, 12, 1–22, doi:10.3390/pharmaceutics12080706.
- Glendenning, A.; Crews, K.; Sturdivant, J.; A deLong, M.; Kopczynski, C.; Lin, C.-W. Sustained Release, Biodegradable PEA Implants for Intravitreal Delivery of the ROCK/PKC Inhibitor AR-13503 | IOVS | ARVO Journals. Invest. Ophthalmol. Vis. Sci. 2018, 59, 5672.
- Ding, J.; Crews, K.; Carbajal, K.; Weksler, M.; Moore, L.; Carlson, E.C.; Lin, C.-W. Ocular Tissue Distribution and Duration of Release of AR-13503 Following Administration of AR-13503 Sustained Release Intravitreal Implant in Rabbits and Miniature Swine | IOVS | ARVO Journals. Invest. Ophthalmol. Vis. Sci. 2019, 60, 5387.
Supplementary Files
Supplementary file. Flow-diagram-PRISMA2020

Round 2
Reviewer 3 Report
Great Job, you improved the manuscript as requested; however, the image quality is very low, please improve it.
Moreover, you said that the manuscript is a systematic review, but the Prisma flowchart should be available in the text, and every article included in the review should be enlisted and classified according to different guidelines, as you can see in a manuscript published by my group:
- 10.3390/jcm9061682
-
https:// doi.org/10.3390/app11020724
Author Response
To whom may concern,
Once again, thank you for taking your time to assess our manuscript. We addressed all the concerns you raised and appreciate your patience. Our responses to your comments are detailed below, point by point and in the order presented.
Review Report Form. Round 2.
Great Job, you improved the manuscript as requested; however, the image quality is very low, please improve it.
- We highly appreciate this comment. Therefore, we have adjusted the quality of the images taking into consideration the instructions provided for authors from the cells journal MDPI webpage. Of note, in the subsection “Preparing Figures, Schemes and Tables” it is established that images must be provided in a single zip archive with a minimum of 1000 pixels width/height or a resolution of 300 dpi or higher. We have achieved resolutions of 6211 x 10301 px, 7081 x 8730 px and 750 x 3986 px for figures 1, 2 and 3 respectively.
Moreover, you said that the manuscript is a systematic review, but the Prisma flowchart should be available in the text, and every article included in the review should be enlisted and classified according to different guidelines, as you can see in a manuscript published by my group:https://doi.org/10.3390/jcm9061682; https://doi.org/10.3390/app11020724
- We appreciate your comment. Regarding the Prisma flowchart, we would like to maintain it as a supplementary file. Even though, if you believe otherwise, it could be included in the text.
- With respect to the list of articles, we have enlisted and classified every article included in the review according to MixedMethodsAppraisalTool (MMAT), designed for the appraisal of the stage of systematic mixed studies reviews, that allows appraising the methodological quality of five categories to studies: qualitative research, randomized controlled trials, non-randomized studies, quantitative descriptive studies, and mixed methods studies. We want to reference this table in the text, in the materials and methods section, if possible.
“2. Materials and Methods
A comprehensive review of the literature was performed through MEDLINE, PubMed, Web of Science, Scopus, and Embase electronic databases, covering the years 1991-2021. Potentially relevant articles were sought by using the search terms in combination as Medical Subject Headings (MeSH) terms and text words: “rho-Associated Kinases”, “Diabetic Retinopathy”, “Diabetic Macular Edema”, “Ripasudil”, “Fasudil” and “Netarsudil”. We also studied reviews, comments, and disquisitions on the pathology.In addition, we scanned the reference lists of the retrieved publications to identify additional relevant articles. The search was supplemented using the MedLine option ‘Related Articles’. No language restrictions were applied. The abstracts for each article were studied to ensure relevance and significance to the review.The identification of studies was illustrated using the PRISMA 2020 flow diagram for new systematic reviews (Supplementary file 1). Two reviewers independently rated the quality of each study by assessing its methodology with the Mixed Methods Appraisal Tool (MMAT), designed for the appraisal of the stage of systematic mixed studies reviews (Supplementary file 2) [1]”.
- Hong QN, Pluye P, Fàbregues S, Bartlett G, Boardman F, Cargo M, Dagenais P, Gagnon M-P, Griffiths F, Nicolau B, O’Cathain A, Rousseau M-C, Vedel I. Mixed Methods Appraisal Tool (MMAT), version 2018. Registration of Copyright (#1148552), Canadian Intellectual Property Office, Industry Canada.